# TIERED GOSSIP LEARNING COMMUNICATION-FRUGAL AND SCALABLE COLLABORATIVE LEARNING

## ABSTRACT

Modern edge deployments require collaborative training schemes that avoid both the single-server bottleneck of federated learning and the high communication burden of peer-to-peer (P2P) systems. We propose Tiered Gossip Learning (TGL), a two-layer push–gossip–pull protocol that combines the fault tolerance of P2P training with the efficiency of hierarchical aggregation. In each round, device-level leaves push their models to a randomly selected set of relays; relays gossip among themselves; and each leaf then pulls and averages models from another random subset of relays. Unlike other hierarchical schemes, TGL is fully coordinator-free, with communication and aggregation decentralized across nodes. It matches baseline accuracy with up to two-thirds fewer model exchanges, and surpasses it when exchanges are equal, across diverse datasets including CIFAR-10, FEM-NIST, and AG-News. We provide convergence guarantees for TGL under standard smoothness, bounded variance and heterogeneity assumptions, and show how its layered structure enables explicit control of consensus-distance bounds. Thus, TGL brings together the strengths of FL and P2P design, enabling robust, low-cost mixing enabling large scale collaborative learning.

## 1 INTRODUCTION

Collaborative training is increasingly vital for machine learning applications spanning edge devices, sensor networks, and distributed organizations. These settings demand decentralized solutions due to limited resources, privacy constraints, and heterogeneity in local (non-iid) data. As data ownership becomes increasingly localized, retaining data at the source has become essential. Federated Learning (FL) (McMahan et al., 2017; Yang et al., 2019; Kairouz et al., 2021) emerged as a practical response to these constraints, enabling clients to share a global model but train locally and periodically send model updates to a central server for aggregation. While FL is attractive for its simplicity and ease of client participation, it struggles to scale. Congestion at the server increases with more clients, slowing down training (Lian et al., 2017) and introducing a single point of failure. A central challenge, therefore, is to support *large-scale participation* without overwhelming any single node.

Hierarchical Federated Learning (HFL) (Liu et al., 2020; Abad et al., 2020) extends this idea by introducing intermediate edge servers beneath a single root server to distribute the aggregation load. However, HFL inherits, and often amplifies the limitations of FL: each added server introduces an additional point of failure, where any server failure disconnects its associated clients. Moreover, as the number of edge servers increases, the root server becomes a communication bottleneck, similar to the original FL setting. Centralized coordination offers simpler orchestration at the cost of fault tolerance and long-term scalability, motivating the need for decentralized alternatives. Although privacy and security concerns in centralized systems are beyond this paper's scope, such considerations have also contributed to the growing interest in fully decentralized paradigms.

Peer-to-Peer Learning (P2PL) (Lian et al., 2017; Koloskova et al., 2020; Kong et al., 2021) eliminates central servers entirely, distributing communication uniformly across all participating nodes. This enhances fault-tolerance and removes single points of failure, but introduces a new trade-off: maintaining strong model mixing requires higher node degrees. The quality of this mixing is governed by the spectral gap of the gossip matrix, which

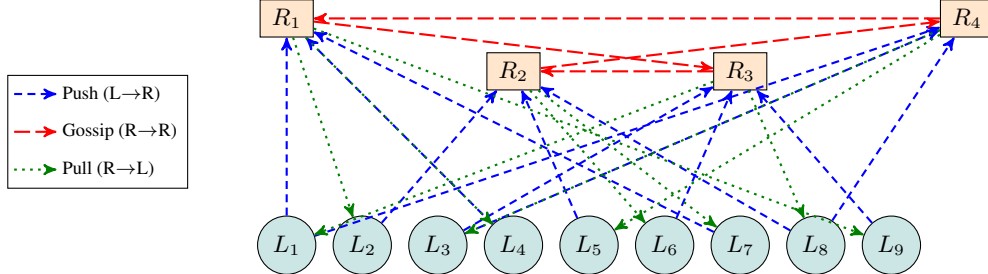

Figure 2: A snapshot of the Tiered Gossip Learning (TGL) network with 9 leaves and 4 relays using 25 directed edges, where connections dynamically change in each round, illustrating the three-stage communication process. In **Stage 1 (Leaf-to-Relay Push)**, relays aggregate models from randomly sampled leaves (shown as blue dashed lines). In **Stage 2 (Relay Gossip)**, each relay exchanges models with other relay (red dashed lines) and averages received models. In **Stage 3 (Relay-to-Leaf Pull)**, each leaf retrieves a model from randomly selected relay (green dotted lines). TGL's two-tier hierarchical design—featuring a decentralized relay layer atop a leaf layer with random dynamic connections enables scalable and fault-tolerant training while significantly reducing communication cost, a key reason behind its superior empirical performance.

reflects how effectively information flows through the network and controls the speed at which models reach consensus. A larger spectral gap implies faster convergence under bounded heterogeneity, but preserving it becomes increasingly expensive as the network scales. As shown in Figure 1, when the number of nodes $n$ is small, even a modest node degree $k$ yields a sufficiently large spectral gap. However, *as $n$ increases, flat P2P systems must raise $k$ to preserve mixing quality*, leading to higher per-node and total communication. Prior work on exponential graphs (Ying et al., 2021) partially addresses this by scaling degree as $\mathcal{O}(\log n)$, but the communication burden still grows system-wide, limiting scalability.

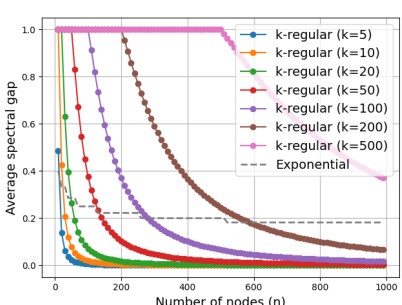

Figure 1: Spectral gap vs. number of nodes $n$ for $k$-random regular and exponential graphs. Maintaining adequate mixing quality as $n$ increases requires scaling up the node degree $k$.

Each prior approach concentrates the burden of scalability differently: FL on a central server, HFL on edge and root servers, and P2PL uniformly across all nodes. However, this uniform burden in P2PL is often impractical, as nodes are forced to scale their communication as the network grows. A key insight is that hierarchical designs with asymmetric load sharing are essential for practical scalability. While HFL attempts this, its centralized structure introduces more points of failure. This motivates a *decentralized top layer* with a small set of high-capacity nodes forming a robust communication backbone.

We address this challenge with Tiered Gossip Learning (TGL), a hybrid design illustrated in Figure 2 that incorporates the key principles of *hierarchy, asymmetric load sharing, and a decentralized top layer with random, dynamic connections.* Leaves interact only with a small, randomly chosen subset of relays; relays gossip among themselves; and leaves then pull updates from another random relay subset. This distributes aggregation across decentralized relays, avoiding single-server bottlenecks while enabling more leaves to participate under fixed per-leaf degree. Spectrally, the two-layer process multiplies the push, gossip, and pull mixing matrices; although each matrix can be sparse, their product yields a denser effective mixing matrix that improves consensus without necessarily increasing leaf degrees. TGL thus operationalizes these core principles to enable scalable, fault-tolerant, and communication-efficient collaborative learning.

We summarize our key contributions as follows:

1. We introduce *Tiered Gossip Learning* (TGL), a two-layer push–gossip–pull protocol that synthesizes structural decentralization with hierarchical efficiency. TGL leverages a tiered

architecture with asymmetric load sharing to achieve strong model mixing at low communication volume, while ensuring that no central coordination is required and all aggregation and communication remain decentralized. This design enhances fault tolerance and enables scalable participation without overloading any single node, addressing key limitations of prior hierarchical and decentralized approaches.

2. We prove convergence of TGL under standard assumptions of smoothness, bounded variance, and bounded heterogeneity. We also derive stage-wise and overall bounds on the expected consensus distance, expressed in terms of network parameters including node count and degree, and highlight how the tiered mixing structure improves global consensus with low node degrees and enables independent control of consensus at each layer.

3. We evaluate TGL on vision tasks: FEMNIST with a CNN and CIFAR-10 with a ResNet, and on a language task, AG News with a TinyTransformer, under non-iid data. We compare performance across varying per-round model exchange budgets. TGL consistently achieves better or comparable accuracy while using only **one-third the communication volume** than prior decentralized baselines, including EL-Local, Exponential, Base-$(k+1)$, and Erdős–Rényi topologies. For instance, on CIFAR-10 with 100 nodes, TGL attains the same accuracy with only 400 edges that the best baseline requires 1200 edges to reach.

## 2 BACKGROUND AND RELATED WORK

**Related Work.** The idea of fully decentralized collaborative learning was popularized by Lian et al. (2017), who showed that Peer-to-Peer Learning (P2PL) using Decentralized SGD (DSGD) can surpass Federated Learning (FL) in wall-clock time by eliminating the central server that easily gets congested. This sparked extensive research on decentralized optimization (Assran et al., 2019; Koloskova et al., 2020), with subsequent work focusing on accelerating convergence via algorithmic refinements (Yu et al., 2019; Yuan et al., 2021; Chen et al., 2021).

A key challenge in collaborative learning is the presence of non-iid data across clients. Under bounded heterogeneity, prior work has focused on strengthening consensus: either by reducing sensitivity to the communication graph (Tang et al., 2018; Li et al., 2019; Kong et al., 2021), or by designing sparse yet well-mixing topologies such as expanders and logarithmic-degree graphs (Nedić et al., 2018; Chow et al., 2016; Ying et al., 2021; Takezawa et al., 2023; Wang et al., 2019; Song et al., 2022). More recently, *Epidemic Learning* (EL) (De Vos et al., 2023) showed that dynamic random graphs can improve mixing more efficiently than fixed topologies. EL also highlighted the importance of relaxing the requirement for doubly stochastic mixing: while its Oracle variant (ELO) relies on a coordinator to construct such matrices, the Local variant (ELL) achieves full decentralization by operating with only row-stochastic weights. Our work continues in this direction, aiming to improve mixing efficiency under constrained communication without any centralized component.

When heterogeneity is large or unbounded, an alternative line of work has explored clustering strategies, where nodes preferentially interact with peers holding similar data or model updates (Ghosh et al., 2020; Sattler et al., 2020). Such methods mitigate the degradation in local performance that arises from enforcing a single global consensus, but generally require orchestration by a central scheduler, and thus remain structurally closer to FL. Some P2PL approaches (Onoszko et al., 2021; Li et al., 2022) have also incorporated clustering mechanisms for adaptive peer discovery.

A complementary orthogonal direction reduces communication cost by limiting client participation per round. Methods such as Teleportation (Takezawa & Stich, 2025) and Plexus (de Vos et al., 2023) lower communication cost by activating only a subset of nodes per iteration, but at the cost of discarding many updates. Other methods reduce the per-message size through compressed communication (Koloskova et al., 2019) or quantization (Chen et al., 2024). In contrast, our work targets scaling to larger node participation under bounded heterogeneity by improving mixing efficiency at limited communication volume, without relying on sampling or compression, and by relaxing the requirement for doubly stochastic matrices to eliminate central orchestration.

**Background** Consider a distributed learning setup with $n_l$ nodes, each holding a private dataset $\mathcal{D}_i$. The nodes collaboratively train a global model by periodically exchanging updates. Each node initializes the same model $\mathbf{x}_0$ and follows a shared training routine. The local objective for node $i$ is: $f^{(i)}(\mathbf{x}) := \mathbb{E}_{\xi \sim \mathcal{D}_i}[f(\mathbf{x}, \xi)]$, $\xi$ being a mini-batch sampled from $\mathcal{D}_i$. The global objective $F$

minimizes the average local objectives across all $n_l$ nodes:

$$\min_{\mathbf{x}} F(\mathbf{x}) = \min_{\mathbf{x}} \frac{1}{n_l} \sum_{i=1}^{n_l} f^{(i)}(\mathbf{x}). \tag{1}$$

Let $X_t \in \mathbb{R}^{n_l \times d}$ denote the global model matrix, whose rows are the node models at round $t$: $X_t = [\mathbf{x}_t^{(1)}, \ldots, \mathbf{x}_t^{(n_l)}]^\top$. This matrix is not available to any individual node but serves as a convenient global representation for analysis. Each node locally updates $\mathbf{x}_t^{(i)} \to \mathbf{x}_{t'}^{(i)}$ for $t < t' < t+1$. The global matrix then updates via mixing, expressed as:

$$X_{t+1} = W_t X_{t'}, \tag{2}$$

where $W_t \in \mathbb{R}^{n_l \times n_l}$ is the mixing matrix. Every row of $W_t$ represents the aggregation weights assigned by a node to all other nodes.

**Mixing Matrix.** Each row $i$ of $W_t$ specifies the aggregation weights assigned by node $i$ to all others. Equivalently, column $i$ represents the weights assigned to node $i$, i.e., its contribution to mixing. Some works assume $W_t$ is doubly stochastic, where both rows and columns sum to one. This assumption simplifies analysis by ensuring equal contribution from all nodes, but enforcing it typically requires centralized coordination, as highlighted by the Oracle variant of Epidemic Learning (EL) (De Vos et al., 2023). Denser matrices typically yield larger spectral gaps and faster mixing, but at the cost of high node degrees and communication cost. In our design, detailed in Section 3, we instead compose multiple sparse matrices across mixing stages to attain strong effective mixing even under sparse connectivity through their product.

**Consensus Distance.** In decentralized settings, where only partial averaging occurs, local models may remain different after each round. To quantify this divergence, we use the *consensus distance* (CD), defined in prior works Kong et al. (2021) as the average squared Euclidean distance of each node's model from the global mean:

$$\text{CD}_t = \frac{1}{n_l} \sum_{i=1}^{n_l} \left\| \mathbf{x}_t^{(i)} - \bar{\mathbf{x}}_t \right\|^2, \tag{3}$$

where $\bar{\mathbf{x}}_t = \frac{1}{n_l} \sum_{i=1}^{n_l} \mathbf{x}_t^{(i)}$. To measure mixing efficiency, we define the *consensus distance ratio* (CDR) as:

$$\text{CDR} = \frac{\text{CD}_{t+1}}{\text{CD}_{t'}}, \tag{4}$$

for $t < t' < t+1$, where $t'$ denotes the pre-gossip stage and $t+1$ the post-gossip stage. A lower CDR indicates stronger mixing. When CD approaches zero, nodes approach exact averaging; how close CDR gets to zero depends on the spectral properties of $W_t$.

**Spectral Gap.** The spectral gap of a mixing matrix $W$ is defined as $1 - \lambda_2$, where $\lambda_2$ is the second-largest eigenvalue in magnitude ($\lambda_1 = 1$ for a row stochastic matrix). This gap measures how quickly models converge to their average. A larger spectral gap implies faster information diffusion and better mixing (Ying et al., 2021; Lovász, 1993; Chung, 1997) under bounded heterogeneity.

**Number of Edges as a Proxy for Communication Cost.** In our setting, model updates are fixed-size messages, so each directed edge corresponds to exactly one exchange. We define one directed edge as one unit of communication cost. A node's degree reflects its individual budget, while the total number of directed edges captures system-wide bandwidth and aggregation effort. Since computation per node also scales with its in-degree, the edge count serves as a unified measure of both communication and aggregation cost. This abstraction provides a simple metric using which we design and evaluate scalable collaborative learning protocols to reduce the communication cost.

## 3 DESIGN OF TIERED GOSSIP LEARNING (TGL)

TGL, as illustrated in Figure 2 is composed of two kinds of nodes: *relays*, which act as server-like intermediaries, and *leaves*, which hold private data and perform local training. A small number of

relays ($n_r$) support a larger population of leaves ($n_l$), mixing and redistributing their models without any direct leaf-to-leaf communication. Leaves may be constrained, with limited connections to relays, while relays are assumed more capable and can sustain larger fan-out to many leaves. This intentional asymmetry offloads heavier communication to the relay layer.

Figure 1 shows that strong mixing can be achieved with lower communication burden when fewer nodes are involved, motivating a modest choice of $n_r$ based on the relay fanout budget. With unbounded relay budget, a single relay suffices and TGL reduces to FL, while tighter budgets can be accommodated by using multiple relays to share the load.

The relays themselves form a fully decentralized gossip layer with no central coordinator. Moreover, all connections in TGL are random and dynamic across rounds. This improves fault tolerance: if a relay fails, the system continues to collaborate through the remaining $n_r - 1$ relays and resampled connections, unlike FL or HFL. In this way, TGL synthesizes the design principles of hierarchy, asymmetric load sharing, decentralized relay layer, and random dynamic connectivity to enable effective mixing, fault-tolerance, and practical scalability without necessarily burdening constrained nodes as the system grows. We parameterize the protocol below.

Between consecutive rounds $t$ and $t+1$, we define three synchronization steps: $t+\frac{1}{4}, t+\frac{2}{4}, t+\frac{3}{4}$, which correspond to the three mixing stages described below. This synchronous formulation, also used in prior work such as EL, simplifies analysis. Extensions to asynchronous operation are also possible and are discussed in Appendix C.

Each leaf $i$ performs local training from $\mathbf{x}_t^{(i)}$ to $\mathbf{x}_{t+\frac{1}{4}}^{(i)}$ via $T_{\text{loc}}$ steps of SGD on its private dataset, where each step uses one mini-batch. The updated models are then mixed in three fixed hops described below.

**Stage 1: Leaf-to-Relay Push.** Each relay $k$ independently selects a random subset of $b_{lr}$ leaves (also its in-degree or budget), denoted by $\mathcal{L}_k$, which respond by pushing their current models to relay $k$, which aggregates them as:

$$\mathbf{x}_{t+\frac{2}{4}}^{(k)} = \frac{1}{b_{lr}} \sum_{i \in \mathcal{L}_k} \mathbf{x}_{t+\frac{1}{4}}^{(i)}, \quad X_{t+\frac{2}{4}} = W_{lr} X_{t+\frac{1}{4}},$$

where $W_{lr} \in \mathbb{R}^{n_r \times n_l}$ is the leaf-to-relay mixing matrix formed from the sampled leaf sets and $\mathbf{x}_{t+\frac{2}{4}}^{(k)}$ is the aggregated model at relay $k$. Sampling is independent across relays and across rounds, ensuring fair selection of leaves over time, with each leaf having an expected out-degree of $n_r b_{lr}/n_l$. Only sampled leaves perform local training in that round.

---

**Algorithm 1** Tiered Gossip Learning (TGL)

---

**Input:** $n_l$ (leaves), $n_r$ (relays), $T$ (global rounds), $T_{\text{loc}}$ (local SGD steps), $\eta$ (learning rate), $b_{lr}, b_{rr}, b_{rl}$: push, gossip, pull budgets

**Init:** $\mathbf{x}_0^{(i)} \leftarrow \mathbf{x}_0, \ \forall i \in [n_l]$

**for** $t = 1$ to $T$ **do**
    **Step 0: Local Training**
    $\mathbf{x}_{t+\frac{1}{4}}^{(i)} \leftarrow \text{SGD}(\mathbf{x}_t^{(i)}, \mathcal{D}_i, T_{\text{loc}}, \eta)$
    **Step 1: Push**
    Each relay $k$ samples $\mathcal{L}_k, |\mathcal{L}_k| = b_{lr}$
    $\mathbf{x}_{t+\frac{2}{4}}^{(k)} \leftarrow \frac{1}{b_{lr}} \sum_{i \in \mathcal{L}_k} \mathbf{x}_{t+\frac{1}{4}}^{(i)}$
    **Step 2: Gossip**
    Each relay $k$ sends to $b_{rr}$ relays and receives from a set of $\mathcal{R}_k$ relays.
    $\mathbf{x}_{t+\frac{3}{4}}^{(k)} \leftarrow \frac{1}{|\mathcal{R}_k|+1}(\mathbf{x}_{t+\frac{2}{4}}^{(k)} + \sum_{m \in \mathcal{R}_k} \mathbf{x}_{t+\frac{2}{4}}^{(m)})$
    **Step 3: Pull**
    Each leaf $i$ samples $\mathcal{S}_i, |\mathcal{S}_i| = b_{rl}$
    $\mathbf{x}_{t+1}^{(i)} \leftarrow \frac{1}{b_{rl}} \sum_{k \in \mathcal{S}_i} \mathbf{x}_{t+\frac{3}{4}}^{(k)}$
**end for**
**Output:** $\{\mathbf{x}_T^{(i)}\}_{i=1}^{n_l}$

---

**Stage 2: Relay-to-Relay Gossip.** Relays then engage in decentralized gossip, each sending its aggregated model to $b_{rr}$ (its out-degree) randomly selected relays. While the out-degree is fixed, we allow the in-degree to vary across rounds, with each relay receiving models from a set $\mathcal{R}_k$ of peers. Over time, the expected in-degree also equals $b_{rr}$, but this relaxation avoids the need for central coordination. Each relay then updates its state by averaging its own model with those it receives:

$$\mathbf{x}_{t+\frac{3}{4}}^{(k)} = \frac{1}{|\mathcal{R}_k|+1}\Big(\mathbf{x}_{t+\frac{1}{2}}^{(k)} + \sum_{m \in \mathcal{R}_k} \mathbf{x}_{t+\frac{1}{2}}^{(m)}\Big), \quad X_{t+\frac{3}{4}} = W_{rr} X_{t+\frac{1}{2}},$$

$W_{rr} \in \mathbb{R}^{n_r \times n_r}$ is the relay-relay mixing matrix and $\mathbf{x}_{t+\frac{3}{4}}^{(k)}$ is the updated model at the relay $k$.

**Stage 3: Relay-to-Leaf Pull.** Each leaf $i$ then pulls updates from a randomly selected subset $\mathcal{S}_i$ of $b_{rl} \geq 1$ relays and averages the received models:

$$\mathbf{x}_{t+1}^{(i)} = \frac{1}{b_{rl}} \sum_{k \in \mathcal{S}_i} \mathbf{x}_{t+\frac{3}{4}}^{(k)}, \quad X_{t+1} = W_{rl} X_{t+\frac{3}{4}},$$

where $W_{rl} \in \mathbb{R}^{n_l \times n_r}$ is the relay-to-leaf mixing matrix and $\mathbf{x}_{t+1}^{(i)}$ is the updated model at leaf $i$ for the next round. Since models have already been mixed in the relay layer, even a small pull budget $b_{rl}$ (as low as 1) can suffice, making it easy for leaves to join and for collaboration to scale. However, when $b_{rl} = 1$, each leaf relies on a single relay, similar to receiving from a single server in FL, which may raise privacy concerns. Larger $b_{rl}$ values mitigate this risk.

Combining all three stages yields the end-to-end transformation:

$$X_{t+1} = W_{rl} W_{rr} W_{lr} X_{t+\frac{1}{4}} \equiv W_{TGL} X_{t+\frac{1}{4}},$$

where $W_{TGL} \in \mathbb{R}^{n_l \times n_l}$ is the overall mixing matrix for TGL as formed by the connections in a given round. Although the individual matrices may be sparse due to constrained budgets, $W_{TGL}$, being the product of three matrices is typically dense, enabling effective mixing across leaves. This dense composition, enabled by the hierarchical structure, is key to TGL's improved performance over all flat P2PL baselines.

Each mixing stage in TGL has its own budget parameter $(b_{lr}, b_{rr}, b_{rl})$, enabling independent control. For example, if leaves are constrained but relays underutilized, increasing $b_{rr}$ densifies $W_{rr}$, improving mixing without raising the leaf budget. This decoupled control is a key advantage of TGL over P2PL systems, where unified roles prevent such separation. Together, the tuple $(n_r, b_{lr}, b_{rr}, b_{rl})$ defines the core design parameters of TGL. In the next section, we analyze how these parameters shape stage-wise consensus and overall convergence.

TGL further benefits from several structural properties. Bounding the in-degrees in Stage 1 (relay receive) and Stage 3 (leaf receive) ensures that no receiver is overwhelmed, either due to too many senders (as in relays receiving from many leaves) or limited capacity (as in leaves pulling from relays). Stage 2, which involves only the small relay set, is less sensitive to such constraints.

The use of dynamic, random sampling provides two practical advantages. First, it naturally tolerates node failures by aggregating only messages that arrive. Second, it enables seamless scaling: when a new node is added, others simply acknowledge it in their sampling pool. A new leaf, for instance, can begin by pulling a recent model in Stage 3 and then participate normally from the following round. Finally, the total number of directed edges per round, $n_r b_{lr} + n_r b_{rr} + n_l b_{rl}$, serves as a direct proxy for communication and computation cost, which we use to normalize comparisons across baselines in our evaluation.

# 4 THEORETICAL ANALYSIS

In this section, we analyze the convergence behavior of TGL under standard assumptions used in stochastic first-order methods. Specifically, we assume for local functions $f$ and global function $F$:

**Assumption 4.1** (Smoothness). For each $i \in [n_l]$, the local function $f^{(i)} : \mathbb{R}^d \to \mathbb{R}$ is differentiable, and there exists a constant $L < \infty$ such that for all $x, y \in \mathbb{R}^d$: $\|\nabla f^{(i)}(y) - \nabla f^{(i)}(x)\| \leq L \|y - x\|$.

**Assumption 4.2** (Bounded Stochastic Noise). There exists a constant $\sigma < \infty$ such that for all $i \in [n_l]$ and $x \in \mathbb{R}^d$: $\mathbb{E}_{\xi \sim D^{(i)}}\big[\|\nabla f(x, \xi) - \nabla f^{(i)}(x)\|^2\big] \leq \sigma^2$, where $\sigma$ captures variance introduced by stochastic gradients due to batch sampling.

**Assumption 4.3** (Bounded Heterogeneity). There exists a constant $\mathcal{H} < \infty$ such that for all $x \in \mathbb{R}^d$: $\frac{1}{n_l} \sum_{i \in [n_l]} \|\nabla f^{(i)}(x) - \nabla F(x)\|^2 \leq \mathcal{H}^2$, where $\mathcal{H}$ quantifies the heterogeneity arising from the non-iid data distribution.

**Theorem 4.4.** *Consider Algorithm 1 under the above assumptions. Let the initial optimization gap be:* $\Delta_0 := F(x_0) - \min_{x \in \mathbb{R}^d} F(x)$. *Then, for any $T \geq 1$, with $n_l \geq 2$ leaves with pull budget $b_{rl} \geq 1$, and $n_r \geq 2$ relays with push and gossip budgets $b_{lr} \geq 1, b_{rr} \geq 1$, selecting the step size as:*

$$\gamma \in \Theta\left(\min\left\{\sqrt{\frac{n_l \Delta_0}{TL((1+\beta')\sigma^2 + \beta'\mathcal{H}^2)}}, \sqrt[3]{\frac{\Delta_0}{TL^2 \beta_{TGL}(\sigma^2 + \mathcal{H}^2)}}, \frac{1}{L}\right\}\right), \text{we obtain:}$$

$$\frac{1}{n_l T} \sum_{t=0}^{T-1} \sum_{i=1}^{n_l} \mathbb{E}\left[\left\|\nabla F(x_t^{(i)})\right\|^2\right] \in \mathcal{O}\left(\sqrt{\frac{L\Delta_0}{Tn_l}((1+\beta')\sigma^2 + \beta'\mathcal{H}^2)} + \sqrt[3]{\frac{L^2\beta_{TGL}\Delta_0^2(\sigma^2+\mathcal{H}^2)}{T^2}} + \frac{L\Delta_0}{T}\right).$$

(5)

*where*

$$\beta_{TGL} := \beta_{lr}\beta_{rr}\beta_{rl}, \quad \beta' := \frac{1}{2}\left[\beta_{TGL} + \frac{n_l}{n_r}\beta_{lr}(1+\beta_{rr})\right], \quad \beta_{lr} := \frac{1}{b_{lr}}\left(1 - \frac{b_{lr}-1}{n_l-1}\right),$$

$$\beta_{rr} := \frac{1}{b_{rr}}\left(1 - \left(1 - \frac{b_{rr}}{n_r-1}\right)^{n_r}\right) - \frac{1}{n_r-1}, \quad \beta_{rl} := \frac{1}{b_{rl}}\left(1 - \frac{b_{rl}-1}{n_r-1}\right)$$

(6)

These $\beta$-terms appear in Lemma A.2 as stage-wise contraction factors, bounding the expected consensus distance after each stage of mixing:

$$\frac{\mathbb{E}[\text{CD}_{t+\frac{2}{4}}]}{\mathbb{E}[\text{CD}_{t+\frac{1}{4}}]} \le \beta_{lr}, \quad \frac{\mathbb{E}[\text{CD}_{t+\frac{3}{4}}]}{\mathbb{E}[\text{CD}_{t+\frac{2}{4}}]} \le \beta_{rr}, \quad \frac{\mathbb{E}[\text{CD}_{t+1}]}{\mathbb{E}[\text{CD}_{t+\frac{3}{4}}]} \le \beta_{rl} \implies \frac{\mathbb{E}[\text{CD}_{t+1}]}{\mathbb{E}[\text{CD}_{t+\frac{1}{4}}]} \le \beta_{\text{TGL}}.$$

As shown in Equation 6, each budget parameter $b_{lr}, b_{rr}, b_{rl}$ directly controls its corresponding $\beta$-value: increasing the budget reduces the contraction factor, yielding faster consensus. Since $\beta_{\text{TGL}}$ is multiplicative, improving any single stage strengthens the overall contraction rate.

We structure the proof through three key lemmas: average preservation in Lemma A.1, bounding stage-wise consensus contraction in Lemma A.2, and bounding the deviation of the average model in Lemma A.3. These results are combined in Lemmas A.4 and A.5 to yield the full convergence proof in Appendix D, and additional analysis in Appendix E, with a complete overview in Appendix A.

## 5 EVALUATION

**Setup** We evaluate TGL on three machine learning tasks with non-iid splits: CIFAR-10 (image), FEMNIST (image), and AGNews (text). Models, node counts, and training settings are in Table 1.

Table 1: Summary of datasets, model architectures (with total parameters and number of output classes), and training configurations. Each configuration is denoted as (learning rate, local steps, batch size, total rounds).

| Dataset | Model (Params, Classes) | Data Distribution | Training Config |
|---|---|---|---|
| CIFAR-10 Krizhevsky & Hinton (2009) | ResNet-20 (0.27M, 10) | Dirichlet $\alpha = 0.1$ | $(0.05, 3, 128, 500)$ |
| FEMNIST Caldas et al. (2018) | CNN (6.6M, 62) | Writer-level non-iid | $(0.02, 3, 32, 500)$ |
| AG News Zhang et al. (2015) | Tiny Transformer (12.9M, 4) | Dirichlet $\alpha = 0.1$ | $(0.04, 4, 64, 500)$ |

We use a 4:1 train–test split for FEMNIST and AG News, and 50k:10k split for CIFAR-10, partitioning the training sets in a non-iid manner across 175/350 nodes for FEMNIST, 100/200 nodes for AG News and CIFAR-10 in our experiments.

**Baselines** We compare TGL against several representative schemes that aim to improve mixing efficiency in decentralized learning. ELL (De Vos et al., 2023) uses fixed out-degree $k$ with random, dynamic connections. Erdos-Renyi (Erdos & Rényi, 1984) network provides a classic randomized baseline where edges are sampled based on fixed probability. Exponential graphs (Ying et al., 2021) assign node degrees that grow logarithmically with network size. Base-$(k+1)$ graphs (Takezawa et al., 2023) extend ring topologies by adding structured shortcuts. FedAvg (McMahan et al., 2017) serves as an FL-based reference. Since ELL performs best among the baselines, we highlight TGL vs. ELL in the main results and defer others to the appendix.

**Metrics** We report test accuracy distributions across all nodes after 500 rounds using candlestick plots (25–75 percentile as body; rest as whiskers). We also track the consensus distance before and after mixing in each round. To capture mixing efficiency, we compute the ratio (CDR) of these two values (after/before) and plot its negative base-10 logarithm. In this metric, larger values correspond to stronger consensus.

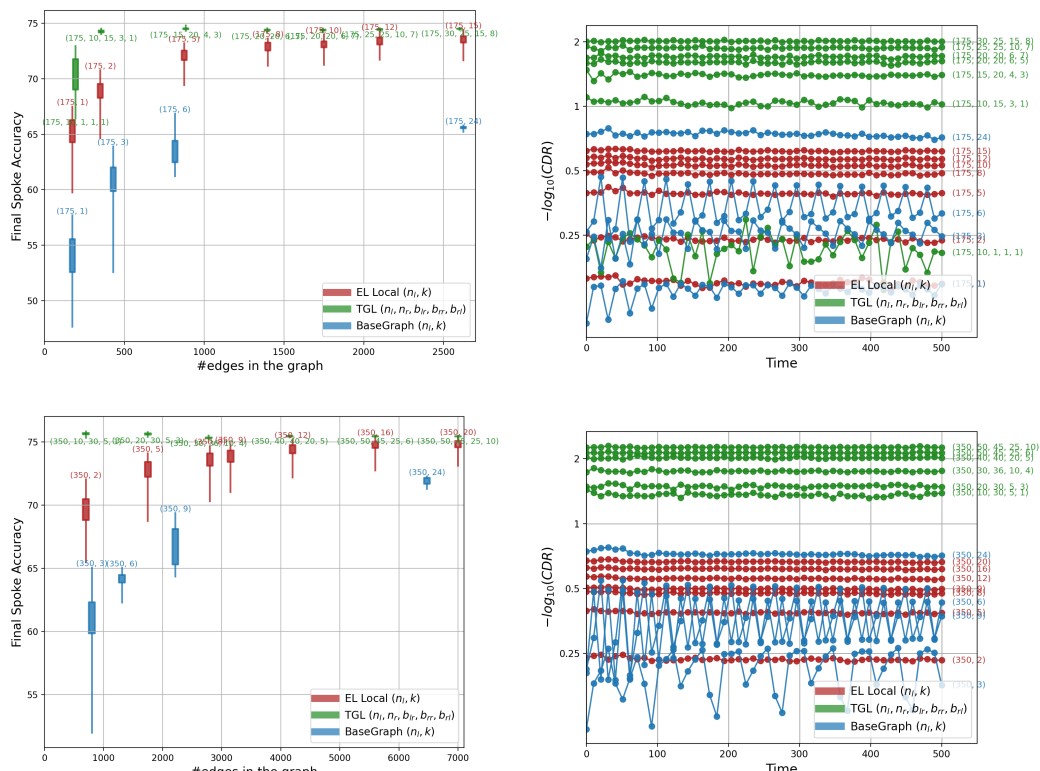

Figure 3: **TGL (green) vs. ELL (red) and BaseGraph (blue) on FEMNIST ($n_l = 175, 350$). Top:** $n_l = 175$. **Left:** Final accuracy vs. edge count: TGL reaches the same accuracy with 400 edges that ELL requires 1200 to match. **Right:** Mixing efficiency across 500 rounds, with TGL consistently above the baselines at matching edge counts. **Bottom:** $n_l = 350$. **Left:** Even at its lowest setting (700 edges), TGL surpasses ELL's best performance at 7000 edges. **Right:** Mixing efficiency again highlights TGL's stronger consensus and scalability.

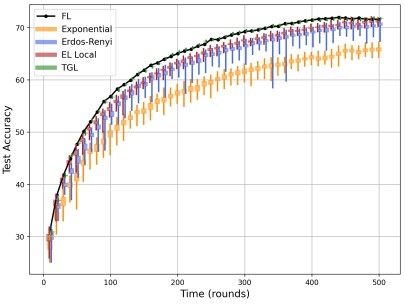

Figure 4: Acc-time plot for CIFAR-10 with 100 nodes. TGL matches FL and outperforms Exponential, Erdos-Renyi, and ELL.

**Results** Figure 3 presents representative results. On the left, the x-axis shows the total communication cost (number of model exchanges per round), and the y-axis shows final test accuracy as candlesticks. TGL consistently outperforms ELL and BaseGraph, achieving comparable or better accuracy at much lower cost. For instance, TGL reaches 74% accuracy with under 400 edges—something ELL needs over 1200 edges for, while Erdős–Rényi tracks ELL closely and BaseGraph lags behind. Each candle is annotated with its configuration tuple. Note that BaseGraph only permits specific edge counts, leading to sparse evaluation points and shows weak performance. For clarity in the plots, we defer full Erdős–Rényi results to Appendix B in Figures 9, 10, 11.

On the right, the plots show the corresponding CDR curves ($-\log_{10}$ of the consensus distance ratio). Except for the lowest-budget configuration $(1, 1, 1)$, which we include only for exposition, all practical TGL configurations yield distinctly higher mixing than the baselines. This strong mixing directly explains the accuracy improvements seen on the left. Even at low leaf degrees (1–3), TGL matches or exceeds the mixing quality of P2P methods that require degree 10 or more, highlighting the efficiency gained from the relay layer. Similar trends are ob-

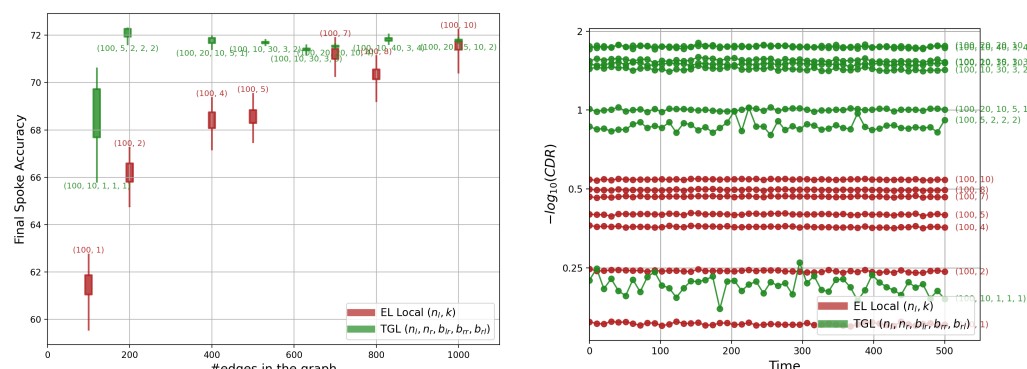

Figure 5: **TGL (green) vs. ELL (red) on CIFAR-10** ($n_l = 100$). Final accuracy (represented as candles) vs. total edges (budget) on the left and mixing efficiency ($-\log(\text{CDR})$) on the right. TGL preserves the high accuracy and better CDR at low budget trend.

served on CIFAR-10 and AG News (Figures 5, 7, 8). For CIFAR-10, BaseGraph is omitted because its weak accuracy compresses the axis scale and obscures differences between TGL and ELL.

TGL not only achieves higher final accuracy, it also maintains this advantage throughout training. Figure 4 shows the accuracy trajectory for CIFAR-10 with 100 nodes at a communication cost of 700 edges. This configuration was chosen for fair comparison with the Exponential Graph baseline, which, for a given number of nodes, deterministically fixes the number of edges. Under this setting, ELL (with $k = 7$) outperforms Exponential Graph, consistent with prior work, while Erdős–Rényi (simulated at 700 edges) again closely follows ELL. TGL, configured as $(100, 20, 15, 10, 2)$ to match 700 edges, tracks closely with Federated Learning (FL), which is included as a performance upper bound. Unlike other methods, FL operates at the lowest cost (100 edges, equal to the number of nodes), and serves to show the maximum accuracy achievable.

Finally, the high mixing observed in the CDR plots explains TGL's accuracy gains. To complete the picture, we complement this with a mathematical simulation: 1000 rounds of randomly generated graphs for TGL, ELL, and other baselines at varying edge counts, measuring the average spectral gap. As shown in Figure 6 (Appendix B), TGL's effective matrices quickly become dense, yielding spectral gaps that approach 1 much faster than the baselines. This links the observed mixing advantage to the spectral properties of TGL, tying the empirical gains back to those same foundations.

## 6 DISCUSSION

TGL demonstrates that a carefully designed tiered topology can combine the fault tolerance of P2P training with the efficiency of hierarchical aggregation. By distributing aggregation across relays, TGL achieves strong consensus with far fewer model exchanges than flat P2P, while remaining coordinator-free. The design is not without trade-offs. Relays introduce a modest management overhead, and each round involves three mixing stages, which may add latency. However, this cost is often offset in practice: the relay layer is small, and its fast gossip allows TGL to reach consensus more quickly than multi-round P2P protocols. A further consideration is privacy: when leaves pull from a single relay, their updates may be easier to infer, though higher pull degrees can mitigate this.

Two properties make TGL especially promising for future extensions. First, it performs well even under sparse connectivity and low exchange counts, suggesting compatibility with adaptive participation schemes. Second, its receiver-driven sampling, with fixed in-degrees in Stages 1 and 3, prevents malicious nodes from targeting specific relays or leaves, improving robustness under failures or attacks.

In summary, TGL shows that tiered gossip can synthesize the strengths of FL and P2P systems, avoiding server bottlenecks, reducing total communication cost, and scaling gracefully, while opening avenues for more adaptive, privacy-aware, and robust collaborative learning.

## ETHICS STATEMENT

This work was conducted in full compliance with the ICLR Code of Ethics. All datasets used in our experiments (CIFAR-10, FEMNIST, and AG News) are publicly available and widely used in the research community, and we cite their original sources appropriately. No private or sensitive data was collected or used. Our study focuses on algorithmic design and evaluation, and we do not foresee any direct negative societal impacts.

## REPRODUCIBILITY STATEMENT

We are committed to ensuring reproducibility of our results. All algorithmic details are provided in Section 3, including a step-by-step description of TGL. Table 1 specifies the datasets, models, and training configurations used. Appendix B gives further details on model architectures and data preprocessing. Table 2 lists all experimental configurations along with the corresponding edge counts, enabling replication of any reported experiment. We will make the codebase publicly available upon publication.

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

## A    APPENDIX - PROOF SKETCH

In this proof, we aim to show convergence properties of the Tiered Gossip Learning (TGL) algorithm as described in Algorithm 1. Specifically, we establish an upper bound on the average gradient norm across nodes and time, characterizing convergence in terms of key parameters: step size $\gamma$, number of iterations $T$, budgets $b_{lr}, b_{rr}, b_{rl}$, number of leaves and relays $n_l, n_r$, and the mixing coefficients $\beta_{TGL}, \beta'$.

The central claim (Theorem 4.4) asserts that, given smoothness (Assumption 4.1), bounded stochastic noise (Assumption 4.2), and bounded heterogeneity (Assumption 4.3), we can choose an appropriate step size $\gamma$ to ensure the average squared gradient norm diminishes at the rate:

$$\mathcal{O}\left(\sqrt{\frac{L\Delta_0}{Tn_l}((1+\beta')\sigma^2 + \beta'\mathcal{H}^2)} + \sqrt[3]{\frac{L^2\beta_{TGL}\Delta_0^2(\sigma^2 + \mathcal{H}^2)}{T^2}} + \frac{L\Delta_0}{T}\right).$$

To achieve this, we decompose the analysis into several intermediate steps encapsulated by individual lemmas:

**Lemma A.1**    This lemma establishes that the average model across all nodes remains invariant in expectation throughout the different stages of a single round of mixing in TGL. Specifically,

$$\mathbb{E}[\overline{x}_{t+1}] = \mathbb{E}[\overline{x}_{t+\frac{3}{4}}] = \mathbb{E}[\overline{x}_{t+\frac{2}{4}}] = \mathbb{E}[\overline{x}_{t+\frac{1}{4}}].$$

This property ensures unbiasedness of the aggregation mechanism at each step.

**Lemma A.2**  This lemma quantifies how the expected consensus distance among the leaf nodes evolves across one full round of TGL mixing. Specifically, it shows that the pairwise distance between leaf models after all three stages (pull, gossip, push) is contractive in expectation, relative to the distance immediately after local updates (before stage 1 mixing). The result is obtained by bounding the contraction at each stage and combining them:

$$
\frac{1}{n_l^2} \sum_{\substack{i,j\in[n_l]\\i\neq j}} \mathbb{E}\left[\left\|x_{t+1}^{(i)} - x_{t+1}^{(j)}\right\|^2\right] \;\leq\; \beta_{TGL} \cdot \frac{1}{n_l^2} \sum_{\substack{i,j\in[n_l]\\i\neq j}} \mathbb{E}\left[\left\|x_{t+\frac{1}{4}}^{(i)} - x_{t+\frac{1}{4}}^{(j)}\right\|^2\right],
$$

where $\beta_{TGL} = \beta_{lr} \cdot \beta_{rr} \cdot \beta_{rl}$. This result captures the compounded effect of communication budgets $b_{lr}, b_{rr}, b_{rl}$ at each mixing stage on consensus formation among leaf models.

**Lemma A.3**  This lemma establishes how the expected deviation in the average model changes during each of the three communication stages in a TGL round: (1) leaf-to-relay push, (2) relay gossip, and (3) relay-to-leaf pull. For each communication stage indexed by $v \in \{1,2,3\}$, the deviation between successive stage-wise averages is bounded in expectation by the consensus error at the current stage, scaled by the corresponding $\beta$ constant:

$$
\mathbb{E}\left[\left\|\bar{x}_{t+\frac{v+1}{4}} - \bar{x}_{t+\frac{v}{4}}\right\|^2\right] \;\leq\; \beta_v \sum_{i\in[n_v]} \mathbb{E}\left[\left\|x_{t+\frac{v}{4}}^{(i)} - \bar{x}_{t+\frac{v}{4}}\right\|^2\right],
$$

where

- For $v = 1$ (leaf-to-relay push): $n_v = n_l$, $\beta_v = \frac{\beta_{lr}}{n_l n_r}$
- For $v = 2$ (relay gossip): $n_v = n_r$, $\beta_v = \frac{\beta_{rr}}{n_r^2}$
- For $v = 3$ (relay-to-leaf pull): $n_v = n_r$, $\beta_v = \frac{\beta_{rl}}{n_r n_l}$

This result captures how local deviations propagate during mixing and is essential in bounding the shift in average models during each stage of the TGL protocol.

**Lemma A.4**  This lemma builds upon Lemma A.2 to bound both the expected consensus distance among leaf models and the variance in their gradients:

**(a) Consensus Distance Bound:**

$$
\frac{1}{n_l^2} \sum_{i,j\in[n_l]} \mathbb{E}\left[\left\|x_t^{(i)} - x_t^{(j)}\right\|^2\right] \leq 20 \cdot \frac{1 + 3\beta_{TGL}}{(1 - \beta_{TGL})^2} \cdot \beta_{TGL} \cdot \gamma^2(\sigma^2 + \mathcal{H}^2).
$$

**(b) Gradient Variance Bound:**

$$
\frac{1}{n_l^2} \sum_{i,j\in[n_l]} \mathbb{E}\left[\left\|g_t^{(i)} - g_t^{(j)}\right\|^2\right] \leq 15(\sigma^2 + \mathcal{H}^2).
$$

**Lemma A.5**  Using the invariance result from Lemma A.1, this lemma provides an upper bound on the squared gradient norm of the global objective evaluated at the mean model $\bar{x}_t$:

$$
\mathbb{E}\left[\left\|\nabla F(\bar{x}_t)\right\|^2\right] \leq \frac{2}{\gamma}\mathbb{E}\left[F(\bar{x}_t) - F(\bar{x}_{t+1})\right] + \frac{L}{2n_l^2} \sum_{i,j\in[n_l]} \mathbb{E}\left[\left\|x_t^{(i)} - x_t^{(j)}\right\|^2\right] + \frac{4L\gamma\sigma^2}{n_l}
$$

$$
+ \frac{4L}{\gamma}\mathbb{E}\left[\left\|\bar{x}_{t+1} - \bar{x}_{t+\frac{3}{4}}\right\|^2 + \left\|\bar{x}_{t+\frac{3}{4}} - \bar{x}_{t+\frac{2}{4}}\right\|^2 + \left\|\bar{x}_{t+\frac{2}{4}} - \bar{x}_{t+\frac{1}{4}}\right\|^2\right].
$$

By applying these lemmas, we systematically derive in Appendix D the final convergence bound on the average squared gradient norm across all leaf nodes and training rounds: $\frac{1}{n_l T}\sum_{t=0}^{T-1}\sum_{i=1}^{n_l}\mathbb{E}\left[\left\|\nabla F(x_t^{(i)})\right\|^2\right]$.

## A.1 All Lemmas and Remarks

This section presents the key lemmas and remarks that are instrumental in establishing the convergence guarantees and characterizing the behavior of the TGL algorithm.

**Lemma A.1** (Average Preservation in Expectation). *The expected average model across all nodes remains unchanged throughout the entire mixing process. That is,*

$$\mathbb{E}[\overline{x}_{t+1}] = \mathbb{E}[\overline{x}_{t+\frac{3}{4}}] = \mathbb{E}[\overline{x}_{t+\frac{2}{4}}] = \mathbb{E}[\overline{x}_{t+\frac{1}{4}}].$$

Here, $\overline{x}$ denotes the average model across all leaf nodes, and the time indices $t + \frac{1}{4}, t + \frac{2}{4}, t + \frac{3}{4}$, and $t + 1$ correspond to the moments after local updates, stage 1 (leaf-to-relay) mixing, stage 2 (relay gossip), and stage 3 (relay-to-leaf) mixing, respectively.

**Lemma A.2.** *For each stage of mixing in TGL, the consensus distance is recursively bounded as follows:*

*(a) Leaf-to-Relay Push:*

$$\frac{1}{n_r^2} \sum_{\substack{i,j \in [n_r] \\ i \neq j}} \mathbb{E}\left[\left\|x_{t+\frac{2}{4}}^{(i)} - x_{t+\frac{2}{4}}^{(j)}\right\|^2\right] \leq \frac{\beta_{lr}}{n_l^2} \sum_{\substack{i,j \in [n_l] \\ i \neq j}} \mathbb{E}\left[\left\|x_{t+\frac{1}{4}}^{(i)} - x_{t+\frac{1}{4}}^{(j)}\right\|^2\right],$$

*where*

$$\beta_{lr} = \frac{1}{b_{lr}}\left(1 - \frac{b_{lr} - 1}{n_l - 1}\right).$$

*(b) Relay Gossip:*

$$\frac{1}{n_r^2} \sum_{\substack{i,j \in [n_r] \\ i \neq j}} \mathbb{E}\left[\left\|x_{t+\frac{3}{4}}^{(i)} - x_{t+\frac{3}{4}}^{(j)}\right\|^2\right] \leq \frac{\beta_{rr}}{n_r^2} \sum_{\substack{i,j \in [n_r] \\ i \neq j}} \mathbb{E}\left[\left\|x_{t+\frac{2}{4}}^{(i)} - x_{t+\frac{2}{4}}^{(j)}\right\|^2\right],$$

*where*

$$\beta_{rr} = \frac{1}{b_{rr}}\left(1 - \left(1 - \frac{b_{rr}}{n_r - 1}\right)^{n_r}\right) - \frac{1}{n_r - 1}.$$

*(c) Relay-to-Leaf Pull:*

$$\frac{1}{n_l^2} \sum_{\substack{i,j \in [n_l] \\ i \neq j}} \mathbb{E}\left[\left\|x_{t+1}^{(i)} - x_{t+1}^{(j)}\right\|^2\right] \leq \frac{\beta_{rl}}{n_r^2} \sum_{\substack{i,j \in [n_r] \\ i \neq j}} \mathbb{E}\left[\left\|x_{t+\frac{3}{4}}^{(i)} - x_{t+\frac{3}{4}}^{(j)}\right\|^2\right],$$

*where*

$$\beta_{rl} = \frac{1}{b_{rl}}\left(1 - \frac{b_{rl} - 1}{n_r - 1}\right).$$

*(d) Final Consensus Bound: Combining the above stages, the consensus distance at time $t + 1$ is bounded in terms of the distance after local updates:*

$$\frac{1}{n_l^2} \sum_{\substack{i,j \in [n_l] \\ i \neq j}} \mathbb{E}\left[\left\|x_{t+1}^{(i)} - x_{t+1}^{(j)}\right\|^2\right] \leq \beta_{TGL} \cdot \frac{1}{n_l^2} \sum_{\substack{i,j \in [n_l] \\ i \neq j}} \mathbb{E}\left[\left\|x_{t+\frac{1}{4}}^{(i)} - x_{t+\frac{1}{4}}^{(j)}\right\|^2\right],$$

*where*

$$\beta_{TGL} = \beta_{lr} \cdot \beta_{rr} \cdot \beta_{rl}.$$

**Lemma A.3.** *For each stage of aggregation in TGL, the expected deviation of the average model is bounded as follows:*

*(a) Leaf-to-Relay Push:*

$$\mathbb{E}\left[\left\|\bar{x}_{t+\frac{2}{4}} - \bar{x}_{t+\frac{1}{4}}\right\|^2\right] = \frac{\beta_{lr}}{n_l n_r} \sum_{i \in [n_l]} \mathbb{E}\left[\left\|x_{t+\frac{1}{4}}^{(i)} - \bar{x}_{t+\frac{1}{4}}\right\|^2\right].$$

*(b) Relay Gossip:*

$$\mathbb{E}\left[\left\|\bar{x}_{t+\frac{3}{4}} - \bar{x}_{t+\frac{2}{4}}\right\|^2\right] \leq \frac{\beta_{rr}}{n_r^2} \sum_{i \in [n_r]} \mathbb{E}\left[\left\|x_{t+\frac{2}{4}}^{(i)} - \bar{x}_{t+\frac{2}{4}}\right\|^2\right].$$

*(c) Relay-to-Leaf Pull:*

$$\mathbb{E}\left[\left\|\bar{x}_{t+1} - \bar{x}_{t+\frac{3}{4}}\right\|^2\right] = \frac{\beta_{rl}}{n_l n_r} \sum_{i \in [n_r]} \mathbb{E}\left[\left\|x_{t+\frac{3}{4}}^{(i)} - \bar{x}_{t+\frac{3}{4}}\right\|^2\right].$$

**Lemma A.4.** *The expected consensus distance and gradient variance across leaf nodes are bounded as follows:*

*(a) Consensus Distance Bound:*

$$\frac{1}{n_l^2} \sum_{i,j \in [n_l]} \mathbb{E}\left[\left\|x_t^{(i)} - x_t^{(j)}\right\|^2\right] \leq 20 \cdot \frac{1 + 3\beta_{TGL}}{(1 - \beta_{TGL})^2} \cdot \beta_{TGL} \cdot \gamma^2(\sigma^2 + \mathcal{H}^2).$$

*(b) Gradient Variance Bound:*

$$\frac{1}{n_l^2} \sum_{i,j \in [n_l]} \mathbb{E}\left[\left\|g_t^{(i)} - g_t^{(j)}\right\|^2\right] \leq 15(\sigma^2 + \mathcal{H}^2).$$

**Lemma A.5.** *The expected gradient norm of the global objective satisfies the following upper bound:*

$$\mathbb{E}\left[\left\|\nabla F(\bar{x}_t)\right\|^2\right] \leq \frac{2}{\gamma}\mathbb{E}\left[F(\bar{x}_t) - F(\bar{x}_{t+1})\right] + \frac{L}{2n_l^2} \sum_{i,j \in [n_l]} \mathbb{E}\left[\left\|x_t^{(i)} - x_t^{(j)}\right\|^2\right]$$

$$+ \frac{4L\gamma\sigma^2}{n_l} + \frac{4L}{\gamma}\mathbb{E}\left[\left\|\bar{x}_{t+1} - \bar{x}_{t+\frac{3}{4}}\right\|^2 + \left\|\bar{x}_{t+\frac{3}{4}} - \bar{x}_{t+\frac{2}{4}}\right\|^2 + \left\|\bar{x}_{t+\frac{2}{4}} - \bar{x}_{t+\frac{1}{4}}\right\|^2\right].$$

*Remark* A.6 (Variance Decomposition). For any set of vectors $\{x_t^{(i)}\}_{i \in [n]}$, the variance around the mean equals the average pairwise squared deviation:

$$\frac{1}{n_l} \sum_{i \in [n]} \left\|x_t^{(i)} - \bar{x}_t\right\|^2 = \frac{1}{2n^2} \sum_{\substack{i,j \in [n] \\ i \neq j}} \left\|x_t^{(i)} - x_t^{(j)}\right\|^2.$$

*Remark* A.7. Note that $\beta_{rr}$, computed in A.2 is decreasing in $b_{rr}$ and increasing in $n_r$, therefore, for any $b_{rr} \geq 1$, and $n_r \geq 2$ we have

$$\beta_{rr}\Big|_{n_r < \infty} \leq \lim_{n_r \to \infty} \beta_1$$

$$= \lim_{n_r \to \infty} \left(1 - \left(1 - \frac{1}{n_r - 1}\right)_r^n - \frac{1}{n_r - 1}\right)$$

$$= 1 - \frac{1}{e},$$

where $\beta_1$ is the $\beta$ at $b_{rr} = 1$. $e$ is Euler's Number and we used the fact that $\lim_{n_r \to \infty}\left(1 - \frac{1}{n_r}\right)_r^n = \frac{1}{e}$.

We also have $\beta_{lr} \leq 1$ and $\beta_{rl} \leq 1$. Multiplying these, we get $\beta_{TGL} \leq 1 - \frac{1}{e}$

# B  ADDITIONAL EXPERIMENTAL DETAILS AND RESULTS

## B.1  SPECTRAL GAP ANALYSIS

To compare the mixing properties of TGL, ELL, and Erdos–Renyi graphs, we simulate their respective mixing matrices across a range of communication budgets. For each configuration, we generate random graphs according to the protocol of the corresponding method and compute the spectral gap. This process is repeated over 1000 independent trials, and the average spectral gap is reported.

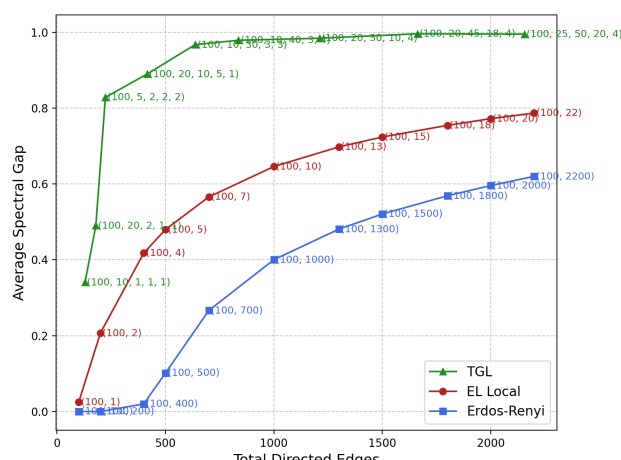

Figure 6: Variation in average spectral gap for TGL, ELL, and Erdos–Renyi as a function of total directed edges.

Figure 6 shows the results. Across all comparable budgets (up to approximately 2200 edges), TGL consistently achieves a substantially higher spectral gap than ELL. As the budget increases, the spectral gaps of all methods begin to converge, however, TGL converges to a higher asymptotic value than both ELL and Erdos–Renyi. These findings directly explain TGL's superior performance in practice, as faster mixing leads to more efficient consensus and optimization. This result strongly supports the theoretical and empirical advantages of TGL. BaseGraph generates near-zero spectral gap for all $k$ when $n_l = 100$ and is therefore omitted from the plot.

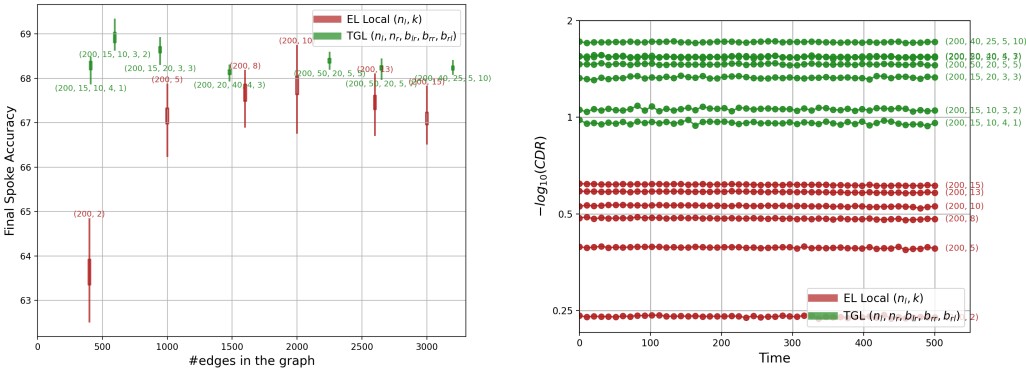

Figure 7: **TGL (green) vs. ELL (red) on CIFAR-10 ($n_l = 200$).** These results complement Figure 5, which reports performance for $n_l = 100$.

### B.2 OTHER RESULTS

Due to space constraints, only representative results were included in the main paper. Here, we provide the complete set of experimental results for a thorough evaluation of TGL.

Figure 3 presents results on FEMNIST for $n_l = 175$ and 350, where TGL is compared against ELL and BaseGraph. TGL consistently achieves the highest test accuracy and lowest consensus distance ratio (CDR). Figure 5 shows results on CIFAR-10 with $n_l = 100$. BaseGraph is excluded from this plot due to its significantly lower accuracy, which compresses the vertical scale and obscures the improvements of TGL over ELL. We also omit results for $n_l = 200$ and AG News from the main paper for brevity, and exclude Erdos–Renyi in some figures due to its near-overlap with ELL.

For completeness, we now present the full results across all datasets. Figure 7 shows the CIFAR-10 results with $n_l = 200$. Figure 8 reports performance on AG News for both $n_l = 100$ and 200.

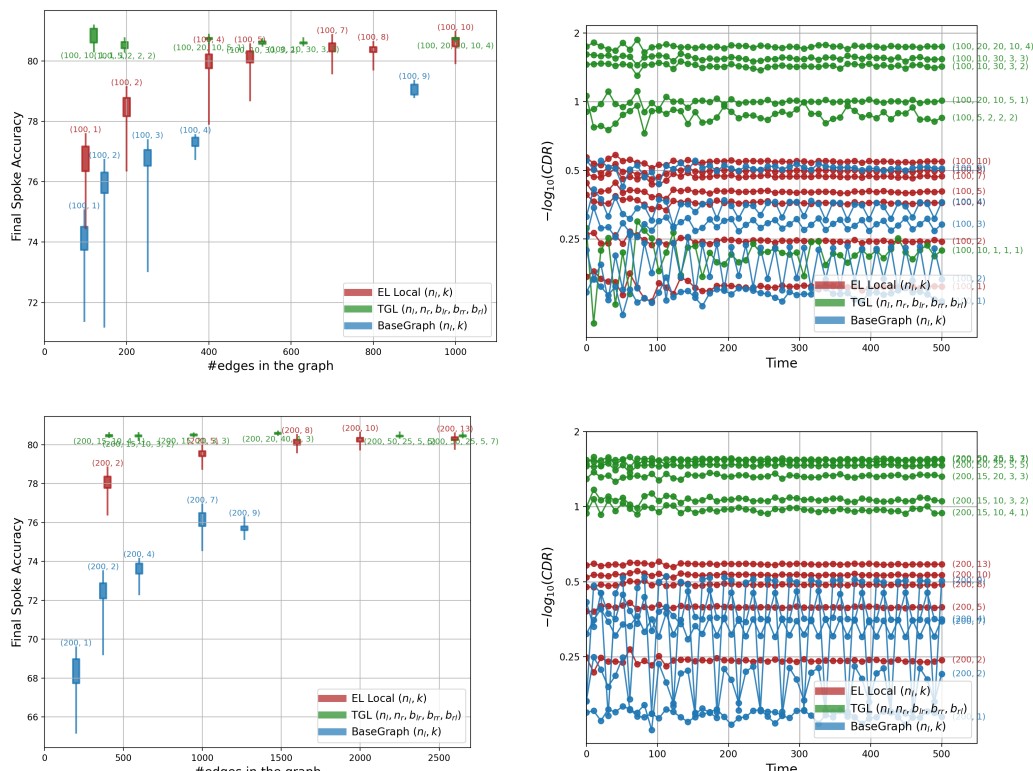

Figure 8: **TGL (green) vs. ELL (red) and BaseGraph (blue) on AG News ($n_l = 100$ (top),** 200 **(bottom)). TGL achieves comparable or superior accuracy relative to ELL even at the lowest communication budgets. Improved accuracy is attributed to enhanced model mixing efficiency. These results align consistently with findings from FEMNIST and CIFAR-10 datasets.**

Erdos–Renyi results on FEMNIST, CIFAR-10, and AG News are shown in Figures 9, 10, and 11, respectively.

Across all datasets and settings, the performance trends remain consistent. TGL configurations using budgets comparable to the lowest ELL setting (e.g., $k = 2$) still achieve strong test accuracy and effective mixing. As a result, TGL shows minimal sensitivity to budget size, in contrast to ELL, BaseGraph, and Erdos–Renyi, which degrade notably under tighter budget constraints. BaseGraph results are only shown at the fixed edge counts allowed by its design and consistently perform the worst across all comparisons.

### B.3 MODEL ARCHITECTURES AND TRAINING SETUP.

We use dataset-specific models optimized for each task. For **CIFAR-10**, we adopt a ResNet-20 with three stages of residual blocks (16, 32, 64 channels), each with 3 basic blocks using 3×3 convolutions, batch normalization, and ReLU. Downsampling is done via strided convolutions, followed by global average pooling and a linear classification head.

For **FEMNIST**, we use a CNN matching the LEAF benchmark: two convolutional layers with 32 and 64 filters (5×5, padding=2), each followed by 2×2 max-pooling; a 2048-unit fully connected ReLU layer; and a final linear classifier over 62 classes.

For **AG News**, we use a lightweight Transformer: two encoder layers with 4 attention heads, 512-dimensional feedforward sublayers, and 128-dimensional token embeddings with learnable positional encodings. Encoded sequences are averaged across tokens and passed through a linear classifier.

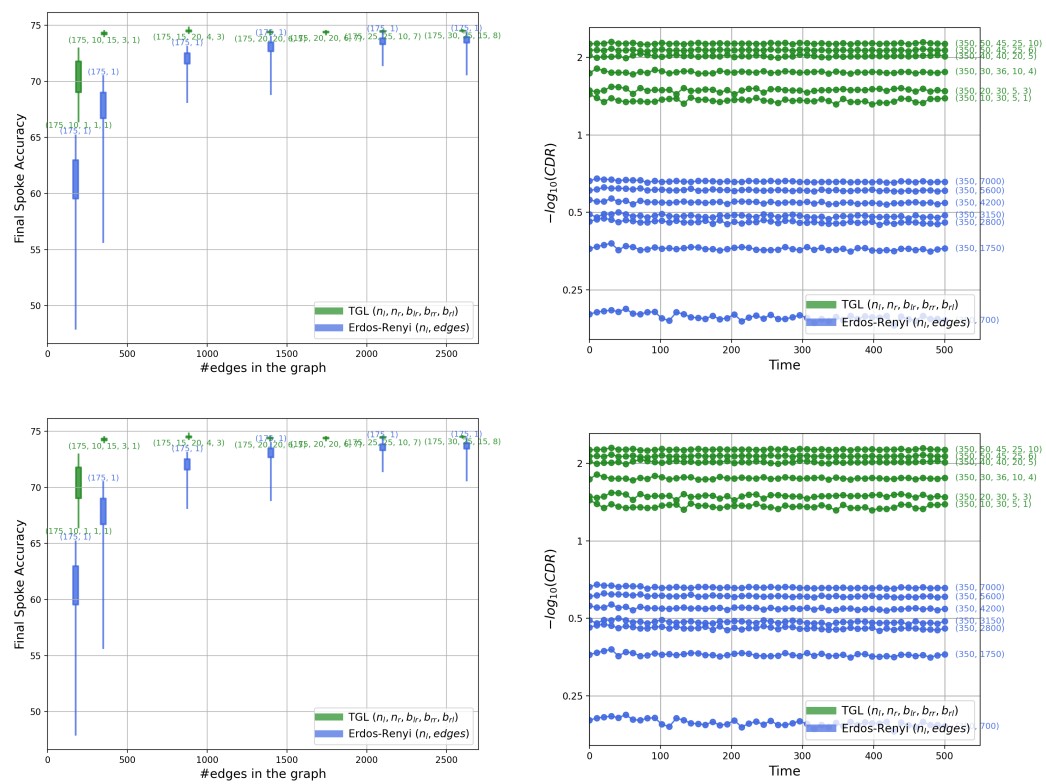

Figure 9: **TGL (green) vs. Erdos-Renyi (blue) on FEMNIST ($n_l = 175$ (top), $350$ (bottom)).**
While Erdos-Renyi performs comparably to ELL, TGL consistently outperforms it across all budgets, maintaining the same advantage observed in earlier comparisons.

All models are trained using cross-entropy loss and vanilla SGD, without momentum or adaptive updates.

### B.4 DATA PREPROCESSING.

For **FEMNIST**, input images are normalized using a mean of 0.1307 and standard deviation of 0.3081. For **CIFAR-10**, we apply standard data augmentation: random cropping (32×32 with padding=4), random horizontal flips, followed by normalization using channel-wise means (0.4914, 0.4822, 0.4465) and standard deviations (0.2023, 0.1994, 0.2010).

For **AG News**, we tokenize text using a basic English tokenizer and construct a vocabulary by scanning the training set. Each token is mapped to a unique ID, with padding (ID 0) and unknown (ID 1) tokens reserved. Inputs are then converted into fixed-length padded sequences based on the longest training sample. Labels are remapped from {1,2,3,4} to {0,1,2,3}. The test set is processed similarly, using the same maximum sequence length as the training set.

All experiments were conducted on a single NVIDIA A100 GPU with 80GB memory. We used PyTorch v2.3.1+cu121, along with torchvision v0.18.1 and torchtext v0.18.0. Node-level parallelism is simulated via sequential local training on a single GPU.

### B.5 DIRECTED EDGE COUNTS ACROSS CONFIGURATIONS.

Table 2 shows the number of directed edges for the different configurations of TGL, ELL, and BaseGraph used. The tuple in TGL represents $(n_r, b_{lr}, b_{rr}, b_{rl})$.

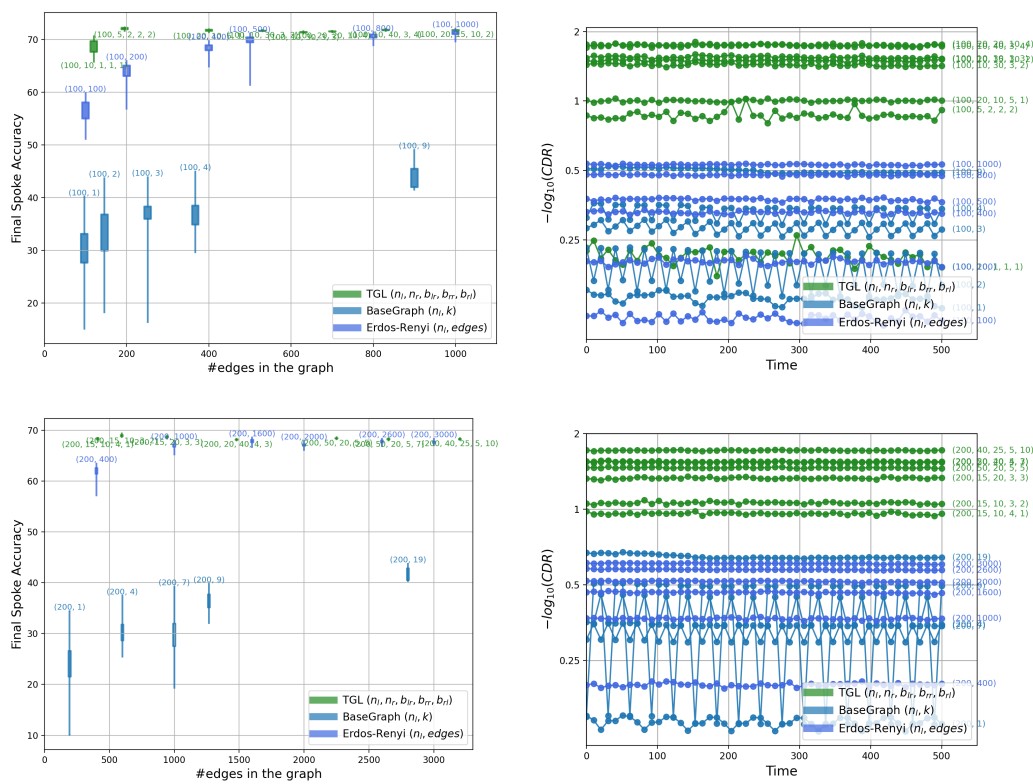

Figure 10: **TGL (green) vs. Erdos-Renyi (blue) on CIFAR-10 ($n_l = 100$ (top), $200$ (bottom)).**

## C  ADDITIONAL NOTES

### C.1  EXTENSION TO ASYNCHRONY

While our implementation is synchronous for clarity of analysis, TGL naturally extends to asynchronous operation using standard techniques. The push and pull stages can employ local buffers with non-blocking requests and staleness-weighted aggregation, similar to asynchronous FL methods such as FedBuff (Nguyen et al., 2022). The relay gossip stage can adopt event-driven, pairwise communication with staleness-aware weighting, as in asynchronous P2P learning (Assran et al., 2019). These adaptations allow TGL to tolerate stragglers gracefully without requiring centralized coordination, preserving its decentralization and scalability.

### C.2  FAULT TOLERANCE

We evaluate TGL under random node failures by dropping a fixed fraction of relays at each round and measuring the resulting test accuracy distributions. Figure 12 summarizes the results. As the drop rate increases, the mean accuracy shows a gradual, graceful decline rather than a sharp collapse. At the same time, the variance across nodes widens, reflecting greater heterogeneity in model quality under high churn. This behavior is consistent with prior observations in decentralized training: random failures primarily increase variance while the system maintains a stable average. These results highlight TGL's robustness to node unavailability, owing to its decentralized relay layer and randomized sampling.

### C.3  PARAMETER SENSITIVITY.

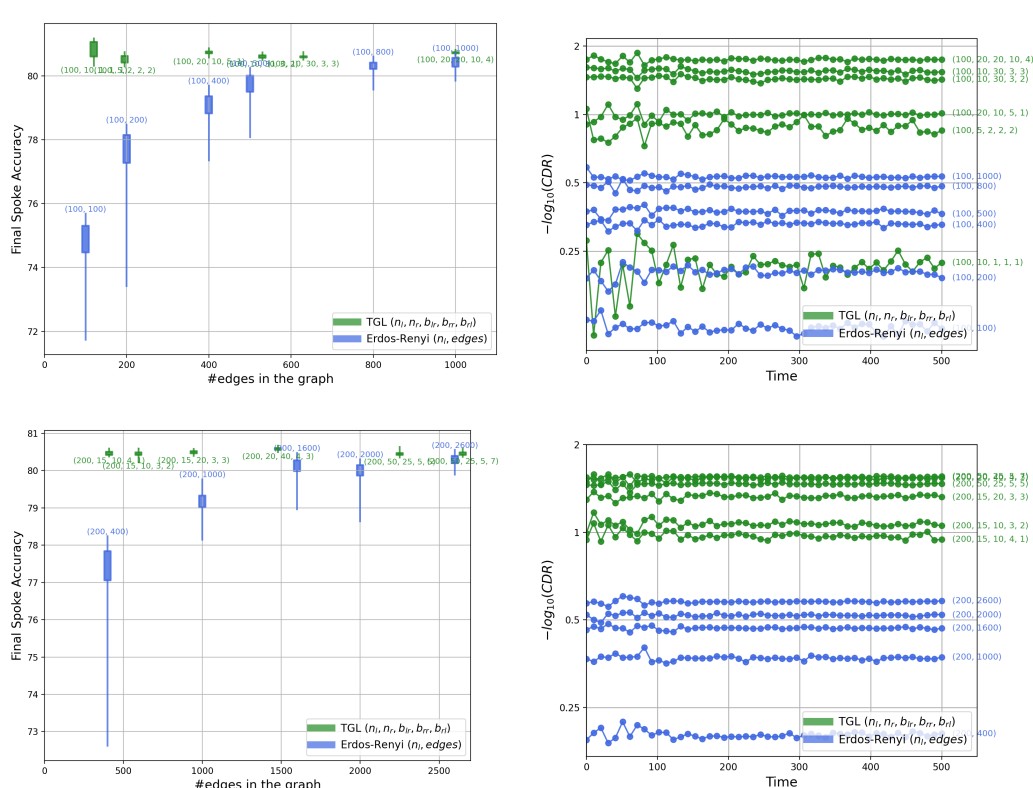

Figure 11: **TGL (green) vs. Erdos-Renyi (blue) on AG News ($n_l = 100$ (top), $200$ (bottom)).**

TGL's performance is not overly sensitive to the precise choice of $(b_{lr}, b_{rr}, b_{rl})$. Across a wide range of configurations, the effective spectral gap of the induced mixing matrix remains very high, ensuring strong consensus. As a result, small changes in these parameters lead to only marginal differences in accuracy, which explains the robustness we observe across experiments. As a general rule, increasing the budgets improves performance, though with diminishing returns once the spectral gap is already large.

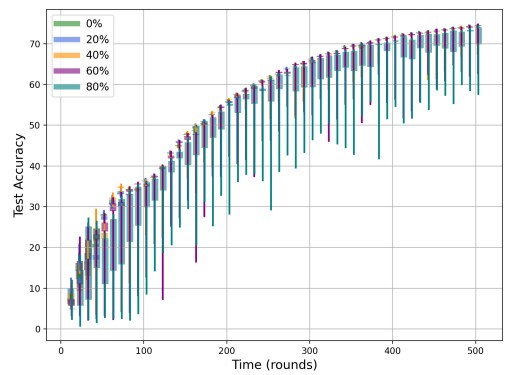

Figure 12: **Fault tolerance of TGL under random node drops.** Increasing drop rates lead to a graceful decline in mean accuracy with rising variance across nodes, demonstrating robustness to failures.

## D  COMPLETE PROOF

Our analysis builds upon the framework introduced in the Epidemic Learning paper De Vos et al. (2023). Specifically, their convergence analysis directly applies to our relay gossip stage (Stage 2). We extend this by developing new analyses for the leaf-to-relay (Stage 1) and relay-to-leaf (Stage 3) communication steps. By combining these components, we establish a complete convergence result for our proposed TGL system.

Table 2: Total number of directed edges for each protocol configuration across different values of $n_l$.

| $n_l$ | TGL | ELL | Base-$(k+1)$ |
|---|---|---|---|
| 100 | (10,1,1,1)→120 | $(k=1)$→100 | $(k=1)$→97 |
| | (5,2,2,2)→195 | $(k=2)$→200 | $(k=2)$→146 |
| | (20,10,5,1)→400 | $(k=4)$→400 | $(k=3)$→251 |
| | (10,30,3,2)→530 | $(k=5)$→500 | $(k=4)$→367 |
| | (10,30,3,3)→630 | $(k=7)$→700 | $(k=9)$→900 |
| | (20,20,10,4)→700 | $(k=8)$→800 | |
| | (10,40,3,4)→830 | $(k=10)$→1000 | |
| | (20,20,10,4)→1000 | | |
| 175 | (10,1,1,1)→195 | $(k=1)$→175 | $(k=1)$→174 |
| | (10,15,3,1)→355 | $(k=2)$→350 | $(k=2)$→328 |
| | (15,20,4,3)→885 | $(k=5)$→875 | $(k=3)$→430 |
| | (20,20,6,5)→1395 | $(k=8)$→1400 | $(k=4)$→550 |
| | (20,20,6,7)→1745 | $(k=10)$→1750 | $(k=5)$→690 |
| | (25,25,10,7)→2100 | $(k=12)$→2100 | $(k=6)$→817 |
| | (30,25,15,8)→2600 | $(k=15)$→2625 | $(k=24)$→2625 |
| 200 | (15,10,4,1)→410 | $(k=2)$→400 | $(k=1)$→194 |
| | (15,10,3,2)→595 | $(k=5)$→1000 | $(k=2)$→282 |
| | (15,20,3,3)→945 | $(k=8)$→1600 | $(k=3)$→459 |
| | (20,40,4,3)→1480 | $(k=10)$→2000 | $(k=4)$→600 |
| | (50,20,5,5)→2250 | $(k=14)$→2600 | $(k=7)$→1000 |
| | (50,20,5,7)→2650 | | $(k=9)$→1267 |
| | (40,25,5,10)→3200 | | $(k=19)$→2800 |
| 350 | (10,30,5,1)→700 | $(k=2)$→700 | $(k=3)$→803 |
| | (20,30,5,3)→1750 | $(k=5)$→1750 | $(k=4)$→975 |
| | (30,36,10,4)→2780 | $(k=8)$→2800 | $(k=6)$→1312 |
| | (40,40,20,5)→4150 | $(k=9)$→3150 | $(k=9)$→2216 |
| | (50,45,25,6)→5600 | $(k=12)$→4200 | $(k=13)$→2450 |
| | (50,45,25,10)→7000 | $(k=16)$→5600 | $(k=24)$→6475 |
| | | $k=20$→7000 | |

## D.1 Proof of Theorem 4.4

*Proof.* Recall that for any vectors $\mathbf{a}, \mathbf{b} \in \mathbb{R}^d$, Jensen's inequality (for the $\ell_2$-norm) states:

$$\|\mathbf{a} + \mathbf{b}\|^2 \leq 2\|\mathbf{a}\|^2 + 2\|\mathbf{b}\|^2.$$

We apply this inequality with $\mathbf{a} = \nabla F(\bar{x}_t)$ and $\mathbf{b} = \nabla F(x_t^{(i)}) - \nabla F(\bar{x}_t)$. For any $i \in [n_l]$, we obtain

$$\mathbb{E}\left[\left\|\nabla F(x_t^{(i)})\right\|^2\right] = \mathbb{E}\left[\left\|\nabla F(\bar{x}_t) + \left(\nabla F(x_t^{(i)}) - \nabla F(\bar{x}_t)\right)\right\|^2\right]$$

$$\leq 2\,\mathbb{E}\left[\|\nabla F(\bar{x}_t)\|^2\right] + 2\,\mathbb{E}\left[\left\|\nabla F(x_t^{(i)}) - \nabla F(\bar{x}_t)\right\|^2\right].$$

Using Assumption 4.1 (*Smoothness*), which implies $\|\nabla F(x) - \nabla F(y)\| \leq L\|x - y\|$, we further bound the second term to obtain:

$$\mathbb{E}\left[\left\|\nabla F(x_t^{(i)})\right\|^2\right] \leq 2\,\mathbb{E}\left[\|\nabla F(\bar{x}_t)\|^2\right] + 2L^2\mathbb{E}\left[\left\|x_t^{(i)} - \bar{x}_t\right\|^2\right].$$

Next, we average over all $i \in [n_l]$:

$$\frac{1}{n_l}\sum_{i=1}^{n_l}\mathbb{E}\left[\left\|\nabla F(x_t^{(i)})\right\|^2\right] \leq 2\,\mathbb{E}\left[\|\nabla F(\bar{x}_t)\|^2\right] + \frac{2L^2}{n_l}\sum_{i=1}^{n_l}\mathbb{E}\left[\left\|x_t^{(i)} - \bar{x}_t\right\|^2\right].$$

Finally, making use of Remark A.6, which states

$$\frac{1}{n_l} \sum_{i=1}^{n_l} \left\| x_t^{(i)} - \bar{x}_t \right\|^2 = \frac{1}{2n_l^2} \sum_{i,j \in [n_l]} \left\| x_t^{(i)} - x_t^{(j)} \right\|^2 ,$$

we get

$$\frac{1}{n_l} \sum_{i=1}^{n_l} \mathbb{E}\left[ \left\| \nabla F(x_t^{(i)}) \right\|^2 \right] \le 2\, \mathbb{E}\left[ \left\| \nabla F(\bar{x}_t) \right\|^2 \right] \;+\; \frac{L^2}{n_l^2} \sum_{i,j \in [n_l]} \mathbb{E}\left[ \left\| x_t^{(i)} - x_t^{(j)} \right\|^2 \right] .$$

Bounding the first term on the RHS using Lemma A.5, we further obtain:

$$\frac{1}{n_l} \sum_{i=1}^{n_l} \mathbb{E}\left[ \left\| \nabla F(x_t^{(i)}) \right\|^2 \right] \le \frac{4}{\gamma} \mathbb{E}\left[ F(\bar{x}_t) - F(\bar{x}_{t+1}) \right] + \frac{2L^2}{n_l^2} \sum_{i,j \in [n_l]} \mathbb{E}\left[ \left\| x_t^{(i)} - x_t^{(j)} \right\|^2 \right]$$

$$+ \frac{8L\gamma\sigma^2}{n_l} + \frac{8L}{\gamma} \mathbb{E}\left[ \left\| \bar{x}_{t+1} - \bar{x}_{t+\frac{3}{4}} \right\|^2 + \left\| \bar{x}_{t+\frac{3}{4}} - \bar{x}_{t+\frac{2}{4}} \right\|^2 + \left\| \bar{x}_{t+\frac{2}{4}} - \bar{x}_{t+\frac{1}{4}} \right\|^2 \right].$$

(7)

Using Lemma A.3 with Remark A.6, we also have:

$$\mathbb{E}\left[ \left\| \bar{x}_{t+\frac{2}{4}} - \bar{x}_{t+\frac{1}{4}} \right\|^2 \right] \le \frac{\beta_{lr}}{2n_r n_l^2} \sum_{i,j \in [n_l]} \mathbb{E}\left[ \left\| x_{t+\frac{1}{4}}^{(i)} - x_{t+\frac{1}{4}}^{(j)} \right\|^2 \right]$$

$$\mathbb{E}\left[ \left\| \bar{x}_{t+\frac{3}{4}} - \bar{x}_{t+\frac{2}{4}} \right\|^2 \right] \le \frac{\beta_{lr}\beta_{rr}}{2n_r n_l^2} \sum_{i,j \in [n_l]} \mathbb{E}\left[ \left\| x_{t+\frac{1}{4}}^{(i)} - x_{t+\frac{1}{4}}^{(j)} \right\|^2 \right]$$

$$\mathbb{E}\left[ \left\| \bar{x}_{t+1} - \bar{x}_{t+\frac{3}{4}} \right\|^2 \right] \le \frac{\beta_{lr}\beta_{rr}\beta_{rl}}{2n_l^3} \sum_{i,j \in [n_l]} \mathbb{E}\left[ \left\| x_{t+\frac{1}{4}}^{(i)} - x_{t+\frac{1}{4}}^{(j)} \right\|^2 \right]$$

Adding the above inequalites,

$$\mathbb{E}\left[ \left\| \bar{x}_{t+1} - \bar{x}_{t+\frac{3}{4}} \right\|^2 \right] + \mathbb{E}\left[ \left\| \bar{x}_{t+\frac{3}{4}} - \bar{x}_{t+\frac{2}{4}} \right\|^2 \right] + \mathbb{E}\left[ \left\| \bar{x}_{t+\frac{2}{4}} - \bar{x}_{t+\frac{1}{4}} \right\|^2 \right] \le \frac{\beta'}{n_l} \frac{1}{n_l^2} \sum_{i,j \in [n_l]} \mathbb{E}\left[ \left\| x_{t+\frac{1}{4}}^{(i)} - x_{t+\frac{1}{4}}^{(j)} \right\|^2 \right]$$

(8)

where

$$\frac{\beta'}{n_l} = \frac{\beta_{lr}}{2n_r} + \frac{\beta_{lr}\beta_{rr}}{2n_r} + \frac{\beta_{lr}\beta_{rr}\beta_{rl}}{2n_l}$$

Remember the partial update step $x_{t+\frac{1}{4}}^{(i)} \triangleq x_t^{(i)} - \gamma\, g_t^{(i)}$. Thus,

$$\mathbb{E}\left[ \left\| x_{t+\frac{1}{4}}^{(i)} - x_{t+\frac{1}{4}}^{(j)} \right\|^2 \right] = \mathbb{E}\left[ \left\| x_t^{(i)} - \gamma\, g_t^{(i)} - x_t^{(j)} + \gamma\, g_t^{(j)} \right\|^2 \right]$$

$$\le 2\, \mathbb{E}\left[ \left\| x_t^{(i)} - x_t^{(j)} \right\|^2 \right] + 2\gamma^2\, \mathbb{E}\left[ \left\| g_t^{(i)} - g_t^{(j)} \right\|^2 \right]. \qquad (9)$$

where we make use of Young's inequality.

Substituting 9 and 8 into 7, we get:

$$\frac{1}{n_l} \sum_{i=1}^{n_l} \mathbb{E}\left[ \left\| \nabla F(x_t^{(i)}) \right\|^2 \right] \le \frac{4}{\gamma} \mathbb{E}\left[ F(\bar{x}_t) - F(\bar{x}_{t+1}) \right] + \frac{8L\gamma\sigma^2}{n_l}$$

$$+ \left( 2L^2 + \frac{16L\beta'}{\gamma n_l} \right) \frac{1}{n_l^2} \sum_{i=1}^{n_l} \mathbb{E}\left[ \left\| x_t^{(i)} - x_t^{(j)} \right\|^2 \right] + \frac{16L\gamma\beta'}{n_l} \frac{1}{n_l^2} \sum_{i=1}^{n_l} \mathbb{E}\left[ \left\| g_t^{(i)} - g_t^{(j)} \right\|^2 \right]$$

(10)

From Remark A.7, we have: $\beta_{TGL} \leq 1 - \frac{1}{e}$ Therefore,

$$20\frac{1 + 3\beta_{TGL}}{(1 - \beta_{TGL})^2} \leq 500$$

We substitute this in Lemma A.4 to get

$$\frac{1}{n_l^2}\sum_{i=1}^{n_l}\mathbb{E}\left[\left\|x_t^{(i)} - x_t^{(j)}\right\|^2\right] \leq 500\beta_{TGL}\gamma^2(\sigma^2 + \mathcal{H}^2)$$

From Lemma A.4, we also have,

$$\frac{1}{n_l^2}\sum_{i=1}^{n_l}\mathbb{E}\left[\left\|g_t^{(i)} - g_t^{(j)}\right\|^2\right] \leq 15(\sigma^2 + \mathcal{H}^2)$$

Substituting this in 10, we obtain:

$$\frac{1}{n_l}\sum_{i=1}^{n_l}\mathbb{E}\left[\left\|\nabla F(x_t^{(i)})\right\|^2\right] \leq \frac{4}{\gamma}\mathbb{E}\left[F(\bar{x}_t) - F(\bar{x}_{t+1})\right] + \left(2L^2 + \frac{16L\beta'}{\gamma n_l}\right)500\beta_{TGL}\gamma^2(\sigma^2 + \mathcal{H}^2)$$

$$+ \frac{8L\gamma\sigma^2}{n_l} + \frac{240}{n_l}L\gamma\beta_{TGL}(\sigma^2 + \mathcal{H}^2)$$

Taking the average over $t \in 0, ..., T - 1$, we obtain:

$$\frac{1}{n_l T}\sum_{t=0}^{T-1}\sum_{i=1}^{n_l}\mathbb{E}\left[\left\|\nabla F(x_t^{(i)})\right\|^2\right] \leq \frac{4}{T\gamma}\Delta_0 + \frac{\gamma}{n_l}\left(16L\beta'500\beta_{TGL}(\sigma^2 + \mathcal{H}^2) + 8L\sigma^2 + 240L\beta_{TGL}(\sigma^2 + \mathcal{H}^2)\right)$$

$$+ \gamma^2\left(2L^2500\beta_{TGL}(\sigma^2 + \mathcal{H}^2)\right)$$

$$\leq \frac{4}{T\gamma}\Delta_0 + \frac{8L\gamma}{n_l}\left((1 + 663\beta')\sigma^2 + 663\beta'\mathcal{H}^2\right) + \gamma^2\left(1000L^2\beta_{TGL}(\sigma^2 + \mathcal{H}^2)\right)$$

$$\tag{11}$$

Here, we make use of the fact that since $\beta_{TGL} \leq 1 - \frac{1}{e}$, $1000\beta_{TGL} \leq 663$ Now, setting

$$\gamma = min\left\{\sqrt{\frac{n_l\Delta_0}{2TL((1 + 663\beta')\sigma^2 + 663\beta'\mathcal{H}^2)}}, \sqrt[3]{\frac{\Delta_0}{250TL^2\beta_{TGL}(\sigma^2 + \mathcal{H}^2)}}, \frac{1}{20L}\right\} \tag{12}$$

we have

$$\frac{1}{\gamma} = max\left\{\sqrt{\frac{2TL((1 + 663\beta')\sigma^2 + 663\beta'\mathcal{H}^2)}{n_l\Delta_0}}, \sqrt[3]{\frac{250TL^2\beta_{TGL}(\sigma^2 + \mathcal{H}^2)}{\Delta_0}}, 20L\right\}$$

$$\leq \sqrt{\frac{2TL((1 + 663\beta')\sigma^2 + 663\beta'\mathcal{H}^2)}{n_l\Delta_0}} + \sqrt[3]{\frac{250TL^2\beta_{TGL}(\sigma^2 + \mathcal{H}^2)}{\Delta_0}} + 20L \tag{13}$$

Plugging equation 13 and equation 12 in equation 11 we obtain

$$\frac{1}{n_l T}\sum_{t=0}^{T-1}\sum_{i=1}^{n_l}\mathbb{E}\left[\left\|\nabla F(x_t^{(i)})\right\|^2\right] \leq 8\sqrt{\frac{2L\Delta_0}{Tn_l}(1 + 663\beta')\sigma^2 + 663\beta'\mathcal{H}^2} + 51\sqrt[3]{\frac{L^2\beta_{TGL}\Delta_0^2(\sigma^2 + \mathcal{H}^2)}{T^2}} + \frac{80L\Delta_0}{T}$$

$$\in \mathcal{O}\left(\sqrt{\frac{L\Delta_0}{Tn_l}((1 + \beta')\sigma^2 + \beta'\mathcal{H}^2)} + \sqrt[3]{\frac{L^2\beta_{TGL}\Delta_0^2(\sigma^2 + \mathcal{H}^2)}{T^2}} + \frac{L\Delta_0}{T}\right)$$

Here, we use the simplification that $\frac{2000}{250^{\frac{2}{3}}} < 51$.

This completes the derivation of the stated bound. $\qquad\square$

## D.2 PROOF OF LEMMA A.1

Here, we prove the *average preservation in expectation* property of TGL.

$$\mathbb{E}\big[\overline{x}_{t+1}\big] = \mathbb{E}\big[\overline{x}_{t+\frac{3}{4}}\big] = \mathbb{E}\big[\overline{x}_{t+\frac{2}{4}}\big] = \mathbb{E}\big[\overline{x}_{t+\frac{1}{4}}\big].$$

*Proof.* From the system dynamics, we have:

$$X_{t+\frac{2}{4}} = W_{lr} X_{t+\frac{1}{4}},$$

where $W_{lr}$ is independent of $X_{t+\frac{1}{4}}$. Taking expectations:

$$\mathbb{E}\big[X_{t+\frac{2}{4}}\big] = \mathbb{E}\big[W_{lr}\big]\mathbb{E}\big[X_{t+\frac{1}{4}}\big].$$

By construction, $W_{lr}$ is row-stochastic, so we have:

$$\sum_{j=1}^{n_l} W_{lr}^{(i,j)} = 1 \quad \text{for all relays } i.$$

Taking expectations and using symmetry (equal probability for all elements), let $c = \mathbb{E}\big[W_{lr}^{(i,j)}\big]$. Then:

$$\sum_{j=1}^{n_l} \mathbb{E}\big[W_{lr}^{(i,j)}\big] = 1 \implies n_l c = 1 \implies c = \frac{1}{n_l}.$$

Thus:

$$\mathbb{E}[W_{lr}] = \frac{1}{n_l}\mathbf{1}_{n_r}\mathbf{1}_{n_l}^T,$$

where $\mathbf{1}_{n_r}$ and $\mathbf{1}_{n_l}$ are column vectors of ones of dimension $n_r$ and $n_l$, respectively.

Substituting, we get:

$$\mathbb{E}\big[X_{t+\frac{2}{4}}\big] = \frac{1}{n_l}\mathbf{1}_{n_r}\mathbf{1}_{n_l}^T\mathbf{1}_{n_l}\overline{x}_{t+\frac{1}{4}}$$

$$= \mathbf{1}_{n_r}\overline{x}_{t+\frac{1}{4}}$$

$$= \mathbb{E}\big[X_{t+\frac{1}{4}}\big]$$

Here we use the fact that $\mathbf{1}_{n_l}^T\mathbf{1}_{n_l} = n_l$.

Thus, $\mathbb{E}\big[\overline{x}_{t+\frac{2}{4}}\big] = \mathbb{E}\big[\overline{x}_{t+\frac{1}{4}}\big]$

Now, consider the **second stage of aggregation** post relay gossip.

$$X_{t+\frac{3}{4}} = W_r X_{t+\frac{2}{4}},$$

Let $n_r$ be the total number of relays, and let $A^{(i)} = |\mathcal{A}_k|$ denote the in-degree of the $i$-th relay, where the outdegree of every relay is fixed to $b_{hs}$. For any relay $i \in [n_r]$, define $I_j^{(i)}$ as the indicator function denoting whether the $j$-th relay is connected to relay $i$. Then we claim:

$$\mathbb{E}\big[x_{t+\frac{3}{4}}^{(i)}\big] = \mathbb{E}\left[\frac{1}{A^{(i)}+1}\left(x_{t+\frac{2}{4}}^{(i)} + \sum_{j\in[n_r]\setminus\{i\}} \mathcal{I}_j^{(i)} x_{t+\frac{2}{4}}^{(j)}\right)\right].$$

First, we take a conditional expectation on $A^{(i)}$:

$$\mathbb{E}\big[x_{t+\frac{3}{4}}^{(i)}\big] = \mathbb{E}\left[\mathbb{E}\left[\frac{1}{A^{(i)}+1}\left(x_{t+\frac{2}{4}}^{(i)} + \sum_{j\in[n_r]\setminus\{i\}} \mathcal{I}_j^{(i)} x_{t+\frac{2}{4}}^{(j)}\right) \ \middle| \ A^{(i)}\right]\right]$$

$$= \mathbb{E}\left[\frac{1}{A^{(i)}+1}\left(x_{t+\frac{2}{4}}^{(i)} + \sum_{j\in[n_r]\setminus\{i\}} \mathbb{E}[\mathcal{I}_j^{(i)} \mid A^{(i)}] x_{t+\frac{2}{4}}^{(j)}\right)\right].$$

Since each of the other $n_r - 1$ relays has the same probability of sending its value to relay $i$, we have

$$\mathbb{E}\big[I_j^{(i)} \mid A^{(i)}\big] = \frac{A^{(i)}}{n_r - 1}.$$

Thus,

$$\mathbb{E}\big[x_{t+\frac{3}{4}}^{(i)}\big] = \mathbb{E}\Big[\frac{1}{A^{(i)} + 1}\Big(x_{t+\frac{2}{4}}^{(i)} + \frac{A^{(i)}}{n_r - 1} \sum_{j \in [n_r] \setminus \{i\}} x_{t+\frac{2}{4}}^{(j)}\Big)\Big]$$

$$= \mathbb{E}\Big[\frac{1}{A^{(i)} + 1}\Big(x_{t+\frac{2}{4}}^{(i)} + \frac{A^{(i)}}{n_r - 1}\big(n_r\, \bar{x}_{t+\frac{2}{4}} - x_{t+\frac{2}{4}}^{(i)}\big)\Big)\Big],$$

where $\bar{x}_{t+\frac{2}{4}} = \frac{1}{n_r} \sum_{j=1}^{n_r} x_{t+\frac{2}{4}}^{(j)}$. Let

$$p = \mathbb{E}\Big[\frac{A^{(i)}}{A^{(i)} + 1}\Big].$$

Collecting terms, it follows that

$$\mathbb{E}\big[x_{t+\frac{3}{4}}^{(i)}\big] = \frac{p\, n_r}{n_r - 1}\, \bar{x}_{t+\frac{2}{4}} + \Big(1 - \frac{p\, n_r}{n_r - 1}\Big) x_{t+\frac{2}{4}}^{(i)}.$$

Averaging over all $i \in [n_r]$ gives

$$\mathbb{E}\big[\bar{x}_{t+\frac{3}{4}}\big] = \mathbb{E}\big[\bar{x}_{t+\frac{2}{4}}\big].$$

Now, we consider the **last step of aggregation** where the leaves aggregate models received from the relays.

From the system dynamics, we have:

$$X_{t+1} = W_{rl}\, X_{t+\frac{3}{4}},$$

where $W_{rl}$ is independent of $X_{t+\frac{3}{4}}$. Taking expectations:

$$\mathbb{E}\big[X_{t+1}\big] = \mathbb{E}\big[W_{rl}\big]\, \mathbb{E}\big[X_{t+\frac{3}{4}}\big].$$

By the construction of $W_{rl}$ as row-stochastic, we have:

$$\sum_{j=1}^{n_r} W_{rl}^{(i,j)} = 1 \quad \text{for all leaves } i.$$

Taking expectations and using symmetry (equal probability for all elements), let

$$c = \mathbb{E}\big[W_{rl}^{(i,j)}\big].$$

Then,

$$\sum_{j=1}^{n_r} \mathbb{E}\big[W_{rl}^{(i,j)}\big] = 1 \quad \implies \quad n_r\, c = 1 \quad \implies \quad c = \frac{1}{n_r}.$$

Thus,

$$\mathbb{E}[W_{rl}] = \frac{1}{n_r}\, \mathbf{1}_{n_l}\, \mathbf{1}_{n_r}^T,$$

where $\mathbf{1}_{n_l}$ and $\mathbf{1}_{n_r}$ are column vectors of ones of dimension $n_l$ and $n_r$, respectively.

Substituting into the expectation, we get:

$$\mathbb{E}\big[X_{t+1}\big] = \frac{1}{n_r}\, \mathbf{1}_{n_l}\, \mathbf{1}_{n_r}^T\, \mathbf{1}_{n_r}\, \bar{x}_{t+\frac{3}{4}} = \mathbf{1}_{n_l}\, \bar{x}_{t+\frac{3}{4}},$$

since $\mathbf{1}_{n_r}^T\, \mathbf{1}_{n_r} = n_r$.

Therefore,

$$\mathbb{E}\big[\bar{x}_{t+1}\big] = \mathbb{E}\big[\bar{x}_{t+\frac{3}{4}}\big].$$

Thus, we conclude that

$$\mathbb{E}\big[\bar{x}_{t+1}\big] = \mathbb{E}\big[\bar{x}_{t+\frac{3}{4}}\big] = \mathbb{E}\big[\bar{x}_{t+\frac{2}{4}}\big] = \mathbb{E}\big[\bar{x}_{t+\frac{1}{4}}\big].$$

$\square$

### D.3 PROOF OF LEMMA A.2

**Stage One mixing: Leaf-to-Relay Push**  The models at the leaves after local training are denoted by $x_{t+\frac{1}{4}}$. Each relay randomly samples $b_{lr}$ leaves, and the model transfer from leaf $j$ to relay $i$ is represented by the indicator function $I_j^i$. relay $i$ aggregates the $b_{lr}$ collected models to produce $x_{t+\frac{2}{4}}$. This proof bounds the consensus distance after mixing to that before it, that is:

$$\frac{1}{n_r^2} \sum_{\substack{i,j\in[n_r] \\ i\neq j}} \mathbb{E}\left[\left\|x_{t+\frac{2}{4}}^{(i)} - x_{t+\frac{2}{4}}^{(j)}\right\|^2\right] \leq \frac{\beta_{lr}}{n_l^2} \sum_{\substack{i,j\in[n_l] \\ i\neq j}} \mathbb{E}\left[\left\|x_{t+\frac{1}{4}}^{(i)} - x_{t+\frac{1}{4}}^{(j)}\right\|^2\right].$$

where

$$\beta_{lr} = \frac{1}{b_{lr}}\left[1 - \frac{b_{lr}-1}{n_l-1}\right].$$

*Proof.*

$$\frac{1}{n_r}\sum_{i\in[n_r]}\mathbb{E}\left[\left\|x_{t+\frac{2}{4}}^{(i)} - \bar{x}_{t+\frac{1}{4}}\right\|^2\right] = \frac{1}{n_r}\sum_i \mathbb{E}\left[\left\|\frac{1}{b_{lr}}\sum_{j\in[n_l]}\mathcal{I}_j^{(i)}x_{t+\frac{1}{4}}^{(j)} - \bar{x}_{t+\frac{1}{4}}\right\|^2\right]$$

$$= \frac{1}{n_r b_{lr}^2}\sum_i \mathbb{E}\left[\left\|\sum_j \left(\mathcal{I}_j^{(i)}x_{t+\frac{1}{4}}^{(j)} - \bar{x}_{t+\frac{1}{4}}\right)\right\|^2\right]$$

$$= \frac{1}{n_r b_{lr}^2}\sum_i \mathbb{E}\left[\sum_j \mathcal{I}_j^{(i)}\left\|x_{t+\frac{1}{4}}^{(j)} - \bar{x}_{t+\frac{1}{4}}\right\|^2\right]$$

$$+ \frac{1}{n_r b_{lr}^2}\sum_i \mathbb{E}\left[\sum_j\sum_{k\neq j}\mathcal{I}_j^{(i)}\mathcal{I}_k^{(i)}\left\langle x_{t+\frac{1}{4}}^{(j)} - \bar{x}_{t+\frac{1}{4}}, \quad x_{t+\frac{1}{4}}^{(k)} - \bar{x}_{t+\frac{1}{4}}\right\rangle\right]$$

$$= \frac{1}{n_r b_{lr}}\frac{1}{n_l}\sum_i \mathbb{E}\left[\sum_j \left\|x_{t+\frac{1}{4}}^{(j)} - \bar{x}_{t+\frac{1}{4}}\right\|^2\right]$$

$$+ \frac{1}{n_r b_{lr}}\frac{b_{lr}-1}{n_l(n_l-1)}(-1)\sum_i \mathbb{E}\left[\sum_j \left\|x_{t+\frac{1}{4}}^{(j)} - \bar{x}_{t+\frac{1}{4}}\right\|^2\right]$$

where we utilize the fact that $\mathbb{E}[\mathcal{I}_j^{(i)}] = \frac{b_{lr}}{n_l}$ and $\mathbb{E}[\mathcal{I}_j^{(i)}\mathcal{I}_k^{(i)}] = \frac{b_{lr}}{n_l}\frac{b_{lr}-1}{n_l-1}$,

Observe that

$$\mathbb{E}\left[\sum_j \left\|x_{t+\frac{1}{4}}^{(j)} - \bar{x}_{t+\frac{1}{4}}\right\|^2\right]$$

is independent of $i$, therefore summing over all $i \in [n_r]$ scales the entire expression by a factor of $n_r$. Thus,

$$\frac{1}{n_r}\sum_j \mathbb{E}\left[\left\|x_{t+\frac{2}{4}}^{(j)} - \bar{x}_{t+\frac{1}{4}}\right\|^2\right] = \frac{1}{n_l b_{lr}}\left(1 - \frac{b_{lr}-1}{n_l-1}\right)\mathbb{E}\left[\sum_j \left\|x_{t+\frac{1}{4}}^{(j)} - \bar{x}_{t+\frac{1}{4}}\right\|^2\right]$$

$$= \frac{\beta_{lr}}{n_l}\mathbb{E}\left[\sum_j \left\|x_{t+\frac{1}{4}}^{(j)} - \bar{x}_{t+\frac{1}{4}}\right\|^2\right]. \tag{14}$$

where

$$\beta_{lr} = \frac{1}{b_{lr}} \left( 1 - \frac{b_{lr} - 1}{n_l - 1} \right)$$

Noting that as $\bar{y}$ is the minimizer of $g(z) := \frac{1}{n} \sum_{i \in [n]} \mathbb{E}\left[ \left\| y^{(i)} - z \right\|^2 \right]$, we have

$$\frac{1}{n} \sum_{i \in [n]} \mathbb{E}\left[ \left\| y^{(i)} - \bar{y} \right\|^2 \right] \leq \frac{1}{n} \sum_{i \in [n]} \mathbb{E}\left[ \left\| y^{(i)} - \bar{x} \right\|^2 \right]. \tag{15}$$

Substituting $x$ to $x_{t+\frac{1}{4}}$ and $y$ to $x_{t+\frac{2}{4}}$, and $n$ to $n_r$, and using (14), we obtain

$$\frac{1}{n_r} \sum_{i=1}^{n_r} \mathbb{E}\left[ \left\| x_{t+\frac{2}{4}}^{(i)} - \bar{x}_{t+\frac{2}{4}} \right\|^2 \right] \leq \frac{1}{n_r} \sum_{i=1}^{n_r} \mathbb{E}\left[ \left\| x_{t+\frac{2}{4}}^{(i)} - \bar{x}_{t+\frac{1}{4}} \right\|^2 \right] = \frac{\beta_{lr}}{n_l} \sum_{i=1}^{n_l} \mathbb{E}\left[ \left\| x_{t+\frac{1}{4}}^{(i)} - \bar{x}_{t+\frac{1}{4}} \right\|^2 \right]. \tag{16}$$

Using Remark A.6, we obtain,

$$\frac{1}{n_r^2} \sum_{\substack{i,j \in [n_r] \\ i \neq j}} \mathbb{E}\left[ \left\| x_{t+\frac{2}{4}}^{(i)} - x_{t+\frac{2}{4}}^{(j)} \right\|^2 \right] \leq \frac{\beta_{lr}}{n_l^2} \sum_{\substack{i,j \in [n_l] \\ i \neq j}} \mathbb{E}\left[ \left\| x_{t+\frac{1}{4}}^{(i)} - x_{t+\frac{1}{4}}^{(j)} \right\|^2 \right]. \tag{17}$$

$\square$

**Stage Two mixing: Relay Gossip** During the relay-gossip stage, every relay shares its models with $b_{rr}$ other relays, all having a constant outdegree. However, the indegree of the relays is a variable $A^{(i)}$. This is exactly how nodes communicate in ELL. Then the consensus distance among the relay models after then gossip stage is bound by:

$$\frac{1}{n_r^2} \sum_{\substack{i,j \\ i \neq j}} \mathbb{E}\left[ \left\| x_{t+\frac{3}{4}}^{(i)} - x_{t+\frac{3}{4}}^{(j)} \right\|^2 \right] \leq \beta_{rr} \cdot \frac{1}{n_r^2} \sum_{\substack{i,j \\ i \neq j}} \mathbb{E}\left[ \left\| x_{t+\frac{2}{4}}^{(i)} - x_{t+\frac{2}{4}}^{(j)} \right\|^2 \right],$$

where

$$\beta_{rr} = \frac{1}{b_{rr}} \left( 1 - \left( 1 - \frac{b_{rr}}{n_r - 1} \right)^{n_r} \right) - \frac{1}{n_r - 1}.$$

*Proof.*

$$\frac{1}{n_r} \sum_{i \in [n_r]} \mathbb{E}\left[\left\| x_{t+\frac{3}{4}}^{(i)} - \bar{x}_{t+\frac{2}{4}} \right\|^2\right] = \frac{1}{n_r} \sum_{i \in [n_r]} \mathbb{E}\left[\left\| \frac{1}{A^{(i)} + 1} \left( x_{t+\frac{2}{4}}^{(i)} + \sum_{j \in [n_r] \setminus \{i\}} \mathcal{I}_j^{(i)} x_{t+\frac{2}{4}}^{(j)} \right) - \bar{x}_{t+\frac{2}{4}} \right\|^2\right]$$

$$= \frac{1}{n_r} \sum_{i \in [n_r]} \mathbb{E}\left[ \mathbb{E}\left[\left\| \frac{1}{A^{(i)} + 1} \left( x_{t+\frac{2}{4}}^{(i)} + \sum_j \mathcal{I}_j^{(i)} x_{t+\frac{2}{4}}^{(j)} \right) - \bar{x}_{t+\frac{2}{4}} \right\|^2 \Big| A^{(i)} \right]\right]$$

$$= \frac{1}{n_r} \sum_{i \in [n_r]} \mathbb{E}\left[ \mathbb{E}\left[\left\| \frac{1}{A^{(i)} + 1} \left( (x_{t+\frac{2}{4}}^{(i)} - x_{t+\frac{2}{4}}^{(i)}) + \sum_j \mathcal{I}_j (x_{t+\frac{2}{4}}^{(j)} - \bar{x}_{t+\frac{2}{4}}) \right) \right\|^2 \Big| A^{(i)} \right]\right]$$

$$= \frac{1}{n_r} \sum_{i \in [n_r]} \mathbb{E}\left[ \frac{1}{A^{(i)} + 1} \mathbb{E}\left[\left\| x_{t+\frac{2}{4}}^{(i)} - x_{t+\frac{2}{4}}^{(i)} \right\|^2 + \sum_j \mathcal{I}_j \left\| (x_{t+\frac{2}{4}}^{(j)} - \bar{x}_{t+\frac{2}{4}}) \right\|^2 \right] \Big| A^{(i)} \right]$$

$$= \frac{1}{n_r} \sum_{i \in [n_r]} \mathbb{E}\left[ \frac{1}{(A^{(i)} + 1)^2} \mathbb{E}\left[ 2 \sum_{j \neq i} \mathcal{I}_j^{(i)} \left\langle x_{t+\frac{2}{4}}^{(i)} - \bar{x}_{t+\frac{2}{4}}, \, x_{t+\frac{2}{4}}^{(j)} - \bar{x}_{t+\frac{2}{4}} \right\rangle \right.\right.$$

$$\left.\left. + \sum_{j \neq i} \sum_{\substack{k \neq i \\ k \neq j}} \mathcal{I}_j^{(i)} \mathcal{I}_k^{(i)} \left\langle x_{t+\frac{2}{4}}^{(j)} - \bar{x}_{t+\frac{2}{4}}, \, x_{t+\frac{2}{4}}^{(k)} - \bar{x}_{t+\frac{2}{4}} \right\rangle \, \Big| A^{(i)} \right]\right]$$

Taking the expectation inside, we obtain

$$\frac{1}{n_r} \sum_{i \in [n_r]} \mathbb{E}[| x_{t+\frac{3}{4}}^{(i)} - \bar{x}_{t+\frac{2}{4}} |^2] = \frac{1}{n_r} \sum_{i \in [n_r]} \mathbb{E}\left[ \frac{1}{(A^{(i)} + 1)^2} \left( \left\| x_{t+\frac{2}{4}}^{(i)} - \bar{x}_{t+\frac{2}{4}} \right\|^2 + \sum_{j \neq i} \mathbb{E}[\mathcal{I}_j^{(i)} | A^{(i)}] \left\| x_{t+\frac{2}{4}}^{(j)} - \bar{x}_{t+\frac{2}{4}} \right\|^2 \right) \right]$$

$$+ \frac{1}{n_r} \sum_{i \in [n_r]} \mathbb{E}\left[ \frac{1}{(A^{(i)} + 1)^2} \left( 2 \sum_{j \neq i} \mathbb{E}[\mathcal{I}_j^{(i)} | A^{(i)}] \langle x_{t+\frac{2}{4}}^{(i)} - \bar{x}_{t+\frac{2}{4}}, x_{t+\frac{2}{4}}^{(j)} - \bar{x}_{t+\frac{2}{4}} \rangle \right) \right]$$

$$+ \frac{1}{n_r} \sum_{i \in [n_r]} \mathbb{E}\left[ \frac{1}{(A^{(i)} + 1)^2} \left( \sum_{j \neq i, k \neq i, k \neq j} \mathbb{E}[\mathcal{I}_j^{(i)} \mathcal{I}_k^{(i)} | A^{(i)}] \langle x_{t+\frac{2}{4}}^{(j)} - \bar{x}_{t+\frac{2}{4}}, x_{t+\frac{2}{4}}^{(k)} - \bar{x}_{t+\frac{2}{4}} \rangle \right) \right] .$$

Observe that $\mathbb{E}[\mathcal{I}_j^{(i)} | A^{(i)}]$ represents the probability of node $j$ selecting node $i$, given that a total of $A^{(i)}$ nodes select $i$. Thus,

$$\mathbb{E}[\mathcal{I}_j^{(i)} | A^{(i)}] = \frac{A^{(i)}}{n_r - 1}$$

Similarly, $\mathcal{I}_j^{(i)} \mathcal{I}_k^{(i)}$ equals 1 only when both $j$ and $k$ choose $i$, hence

$$\mathbb{E}\left[ \mathcal{I}_j^{(i)} \mathcal{I}_k^{(i)} | A^{(i)} \right] = \frac{A^{(i)}(A^{(i)} - 1)}{(n_r - 1)(n_r - 2)}.$$

Also, note that

$$\sum_{j \neq i} \langle x_{t+\frac{2}{4}}^{(i)} - \bar{x}_{t+\frac{2}{4}}, x_{t+\frac{2}{4}}^{(j)} - \bar{x}_{t+\frac{2}{4}} \rangle = \langle x_{t+\frac{2}{4}}^{(i)} - \bar{x}_{t+\frac{2}{4}}, \sum_{j \neq i} (x_{t+\frac{2}{4}}^{(j)} - \bar{x}_{t+\frac{2}{4}}) \rangle = - \left\| x_{t+\frac{2}{4}}^{(i)} - \bar{x}_{t+\frac{2}{4}} \right\|^2$$

and

$$\sum_{j\neq i}\sum_{k\neq i,k\neq j}\langle x_{t+\frac{2}{4}}^{(j)} - \bar{x}_{t+\frac{2}{4}}, x_{t+\frac{2}{4}}^{(k)} - \bar{x}_{t+\frac{2}{4}}\rangle = \sum_{j\neq i}\left\langle x_{t+\frac{2}{4}}^{(j)} - \bar{x}_{t+\frac{2}{4}}, \sum_{k\neq i,k\neq j}(x_{t+\frac{2}{4}}^{(k)} - \bar{x}_{t+\frac{2}{4}})\right\rangle$$

$$= \sum_{j\neq i}\left\langle x_{t+\frac{2}{4}}^{(j)} - \bar{x}_{t+\frac{2}{4}}, (x_{t+\frac{2}{4}}^{(i)} - \bar{x}_{t+\frac{2}{4}}) + (x_{t+\frac{2}{4}}^{(j)} - \bar{x}_{t+\frac{2}{4}})\right\rangle$$

$$= \|x_{t+\frac{2}{4}}^{(i)} - \bar{x}_{t+\frac{2}{4}}\|^2 - \sum_{j\neq i}\|x_{t+\frac{2}{4}}^{(j)} - \bar{x}_{t+\frac{2}{4}}\|^2$$

Bringing everything together, we obtain

$$\frac{1}{n_r}\sum_{i\in[n_r]}\mathbb{E}\left[\left\|x_{t+\frac{3}{4}}^{(i)} - \bar{x}_{t+\frac{2}{4}}\right\|^2\right] = \frac{1}{n_r}\sum_{i\in[n_r]}\mathbb{E}\left[\frac{1}{(A^{(i)}+1)^2}\left(\left\|x_{t+\frac{2}{4}}^{(i)} - \bar{x}_{t+\frac{2}{4}}\right\|^2 + \frac{A^{(i)}}{n_r-1}\sum_{j\neq i}\left\|x_{t+\frac{2}{4}}^{(j)} - \bar{x}_{t+\frac{2}{4}}\right\|^2\right)\right]$$

$$+ \frac{1}{n_r}\sum_{i\in[n_r]}\mathbb{E}\left[\frac{1}{(A^{(i)}+1)^2}\left(-\frac{2A^{(i)}}{n_r-1}\left\|x_{t+\frac{2}{4}}^{(i)} - \bar{x}_{t+\frac{2}{4}}\right\|^2\right)\right]$$

$$+ \frac{1}{n_r}\sum_{i\in[n_r]}\mathbb{E}\left[\frac{1}{(A^{(i)}+1)^2}\left(\frac{A^{(i)}(A^{(i)}-1)}{(n_r-1)(n_r-2)}\left(\left\|x_{t+\frac{2}{4}}^{(i)} - \bar{x}_{t+\frac{2}{4}}\right\|^2\right.\right.\right.$$

$$\left.\left.\left.- \sum_{j\neq i}\left\|x_{t+\frac{2}{4}}^{(j)} - \bar{x}_{t+\frac{2}{4}}\right\|^2\right)\right)\right]$$

$$= \frac{1}{n_r}\sum_{i\in[n_r]}\left\|x_{t+\frac{2}{4}}^{(i)} - \bar{x}_{t+\frac{2}{4}}\right\|^2 \mathbb{E}\left[\frac{1}{(A^{(i)}+1)^2}\left(1 - \frac{2A^{(i)}}{n_r-1} + \frac{A^{(i)}(A^{(i)}-1)}{(n_r-1)(n_r-2)}\right)\right]$$

$$+ \frac{1}{n_r}\sum_{i\in[n_r]}\mathbb{E}\left[\frac{1}{(A^{(i)}+1)^2}\left(\frac{A^{(i)}}{n_r-1} - \frac{A^{(i)}(A^{(i)}-1)}{(n_r-1)(n_r-2)}\right)\right]\sum_{j\neq i}\left\|x_{t+\frac{2}{4}}^{(j)} - \bar{x}_{t+\frac{2}{4}}\right\|^2$$

Observe that, due to symmetry, the distribution of $A^{(i)}$ is identical to that of $A^{(j)}$ for any $i, j \in [n_r]$. Hence,

$$\frac{1}{n_r}\sum_{i\in[n_r]}\mathbb{E}\left[\left\|x_{t+\frac{3}{4}}^{(i)} - \bar{x}_{t+\frac{2}{4}}\right\|^2\right]$$

$$= \frac{1}{n_r}\sum_{i\in[n_r]}\left\|x_{t+\frac{2}{4}}^{(i)} - \bar{x}_{t+\frac{2}{4}}\right\|^2 \mathbb{E}\left[\frac{1}{(A^{(1)}+1)^2}\left(1 - \frac{2A^{(1)}}{n_r-1} + \frac{A^{(1)}(A^{(1)}-1)}{(n_r-1)(n_r-2)} + \frac{A^{(1)}}{n_r-1} - \frac{A^{(1)}(A^{(1)}-1)}{(n_r-2)}\right)\right]$$

Now note that

$$1 - \frac{2A^{(1)}}{n_r-1} + \frac{A^{(1)}(A^{(1)}-1)}{(n_r-1)(n_r-2)} + A^{(1)} - \frac{A^{(1)}(A^{(1)}-1)}{n_r-2} = 1 + A^{(1)} - \frac{{A^{(1)}}^2 + A^{(1)}}{n_r-1} = (1+A^{(1)})\left(1 - \frac{A^{(1)}}{n_r-1}\right)$$

Thus

$$\frac{1}{n_r}\sum_{i\in[n_r]}\mathbb{E}\left[\left\|x_{t+\frac{3}{4}}^{(i)} - \bar{x}_{t+\frac{2}{4}}\right\|^2\right] = \frac{1}{n_r}\sum_{i\in[n_r]}||x_{t+\frac{2}{4}}^{(i)} - \bar{x}_{t+\frac{2}{4}}||^2\left[\mathbb{E}\left[\frac{1}{A^{(1)}+1}\right] - \frac{1}{n_r-1}\cdot\mathbb{E}\left[\frac{A^{(1)}}{A^{(1)}+1}\right]\right]$$

Observe that since each node $j \neq 1$ independently and uniformly selects a set of $b_{rr}$ nodes, $A^{(1)}$ follows a binomial distribution with parameters $n_r - 1$ and $\frac{b_{rr}}{n_r-1}$. Thus, for $b_{rr} > 0$, we have

$$\mathbb{E}\left[\frac{1}{A^{(1)}+1}\right] = \sum_{k=0}^{n_r-1} \frac{1}{k+1}\binom{n_r-1}{k}\left(\frac{b_{rr}}{n_r-1}\right)^k\left(1-\frac{b_{rr}}{n_r-1}\right)^{n_r-1-k}$$

$$= \frac{n_r-1}{b_{rr}n_r}\sum_{k=0}^{n_r-1}\binom{n_r}{k+1}\left(\frac{b_{rr}}{n_r-1}\right)^{k+1}\left(1-\frac{b_{rr}}{n_r-1}\right)^{n_r-1-k}$$

$$= \frac{n_r-1}{b_{rr}n_r}\left(1-\left(1-\frac{b_{rr}}{n_r-1}\right)^{n_r}\right)$$

Also noting that

$$\mathbb{E}\left[\frac{A^{(1)}}{A^{(1)}+1}\right] = 1 - \mathbb{E}\left[\frac{1}{A^{(1)}+1}\right],$$

we obtain

$$\frac{1}{n_r}\sum_{i\in[n_r]}\mathbb{E}\left[\left\|x_{t+\frac{3}{4}}^{(i)}-\bar{x}_{t+\frac{2}{4}}\right\|^2\right] = \left[\frac{1}{b_{rr}}\left(1-\left(1-\frac{b_{rr}}{n_r-1}\right)^{n_r}\right)-\frac{1}{n_r-1}\right]\frac{1}{n_r}\sum_{i\in[n_r]}\left\|x_{t+\frac{2}{4}}^{(i)}-\bar{x}_{t+\frac{2}{4}}\right\|^2$$

$$\tag{18}$$

we obtain that

Noting that as $\bar{y}$ is the minimizer of $g(z) := \frac{1}{n}\sum_{i\in[n]}\mathbb{E}\left[\left\|y^{(i)}-z\right\|^2\right]$, following the logic in (15), and using Reamrk A.6, we convert the equality in (18) to the following inequality:

$$\frac{1}{n_r^2}\sum_{i,j\in[n_r]}\mathbb{E}\left[\left\|x_{t+\frac{3}{4}}^{(i)}-x_{t+\frac{3}{4}}^{(j)}\right\|^2\right] \leq \left[\frac{1}{b_{rr}}\left(1-\left(1-\frac{b_{rr}}{n_r-1}\right)^{n_r}\right)-\frac{1}{n_r-1}\right]\frac{1}{n_r^2}\sum_{i,j\in[n_r]}\mathbb{E}\left[\left\|x_{t+\frac{2}{4}}^{(i)}-x_{t+\frac{2}{4}}^{(j)}\right\|^2\right]$$

$$\tag{19}$$

which is the desired result. $\qquad\square$

**Stage Three Mixing: Relay-to-Leaf Pull**  The models at the relays after relay gossip are represented as $x_{t+\frac{3}{4}}$. Each leaf independently selects $b_{rl}$ relays at random, with the model transfer from relay $j$ to leaf $i$ indicated by the function $\mathcal{I}_j^i$. Leaf $i$ then aggregates the $b_{rl}$ received models to obtain $x_{t+1}$. This proof establishes an upper bound on the consensus distance among the leaf models after the final aggregation stage relative to its value before aggregation, that is,

$$\frac{1}{n_l^2}\sum_{\substack{i,j\in[n_l]\\i\neq j}}\mathbb{E}\left[\left\|x_{t+1}^{(i)}-x_{t+1}^{(j)}\right\|^2\right] \leq \frac{\beta_{rl}}{n_r^2}\sum_{\substack{i,j\in[n_r]\\i\neq j}}\mathbb{E}\left[\left\|x_{t+\frac{3}{4}}^{(i)}-x_{t+\frac{3}{4}}^{(j)}\right\|^2\right].$$

where

$$\beta_{rl} = \frac{1}{b_{rl}}\left[1-\frac{b_{rl}-1}{n_r-1}\right].$$

*Proof.*

$$\frac{1}{n_l} \sum_{i \in [n_l]} \mathbb{E}\left[\left\|x_{t+1}^{(i)} - \bar{x}_{t+\frac{3}{4}}\right\|^2\right] = \frac{1}{n_l} \sum_i \mathbb{E}\left[\left\|\frac{1}{b_{rl}} \sum_j \mathcal{I}_j^{(i)} x_{t+\frac{3}{4}}^{(j)} - \bar{x}_{t+\frac{3}{4}}\right\|^2\right]$$

$$= \frac{1}{n_l\, b_{rl}^2} \sum_i \mathbb{E}\left[\left\|\sum_j \left(\mathcal{I}_j^{(i)} x_{t+\frac{3}{4}}^{(j)} - \bar{x}_{t+\frac{3}{4}}\right)\right\|^2\right]$$

$$= \frac{1}{n_l\, b_{rl}^2} \sum_i \mathbb{E}\left[\sum_j \mathcal{I}_j^{(i)} \left\|x_{t+\frac{3}{4}}^{(j)} - \bar{x}_{t+\frac{3}{4}}\right\|^2\right]$$

$$+ \frac{1}{n_l\, b_{rl}^2} \sum_i \mathbb{E}\left[\sum_j \sum_{k \neq j} \mathcal{I}_j^{(i)} \mathcal{I}_k^{(i)} \left\langle x_{t+\frac{3}{4}}^{(j)} - \bar{x}_{t+\frac{3}{4}},\ x_{t+\frac{3}{4}}^{(k)} - \bar{x}_{t+\frac{3}{4}}\right\rangle\right]$$

$$= \frac{1}{n_l\, b_{rl}} \frac{1}{n_r} \sum_i \mathbb{E}\left[\sum_j \left\|x_{t+\frac{3}{4}}^{(j)} - \bar{x}_{t+\frac{3}{4}}\right\|^2\right]$$

$$+ \frac{1}{n_l\, b_{rl}} \frac{b_{rl}-1}{n_r(n_r-1)}(-1) \sum_i \mathbb{E}\left[\sum_j \left\|x_{t+\frac{3}{4}}^{(j)} - \bar{x}_{t+\frac{3}{4}}\right\|^2\right]$$

where we utilize the fact that $\mathbb{E}[\mathcal{I}_j^{(i)}] = \frac{b_{rl}}{n_r}$ and $\mathbb{E}[\mathcal{I}_j^{(i)} \mathcal{I}_k^{(i)}] = \frac{b_{rl}}{n_r} \frac{b_{rl}-1}{n_r-1}$,

Just like for stage 1 aggregation, observe that

$$\mathbb{E}\left[\sum_j \left\|x_{t+\frac{3}{4}}^{(j)} - \bar{x}_{t+\frac{3}{4}}\right\|^2\right]$$

is independent of $i$, therefore summing over all $i \in [n_l]$ scales the entire expression by a factor of $n_l$. Thus,

$$\frac{1}{n_l} \sum_{i \in [n_l]} \mathbb{E}\left[\left\|x_{t+1}^{(i)} - \bar{x}_{t+\frac{3}{4}}\right\|^2\right] = \frac{1}{n_r\, b_{rl}} \left(1 - \frac{b_{rl}-1}{n_r-1}\right) \mathbb{E}\left[\sum_j \left\|x_{t+\frac{3}{4}}^{(j)} - \bar{x}_{t+\frac{3}{4}}\right\|^2\right]$$

$$= \frac{\beta_{lr}}{n_r} \mathbb{E}\left[\sum_j \left\|x_{t+\frac{3}{4}}^{(j)} - \bar{x}_{t+\frac{3}{4}}\right\|^2\right]. \tag{20}$$

where $\beta_{rl}$:

$$\beta_{rl} = \frac{1}{b_{rl}} \left(1 - \frac{b_{rl}-1}{n_r-1}\right).$$

Using the minimizer property described in (15), and applying Remark A.6, we finally obtain:

$$\frac{1}{n_l^2} \sum_{\substack{i,j \in [n_l] \\ i \neq j}} \mathbb{E}\left[\left\|x_{t+1}^{(i)} - x_{t+1}^{(j)}\right\|^2\right] \leq \frac{\beta_{rl}}{n_r^2} \sum_{\substack{i,j \in [n_r] \\ i \neq j}} \mathbb{E}\left[\left\|x_{t+\frac{3}{4}}^{(i)} - x_{t+\frac{3}{4}}^{(j)}\right\|^2\right]. \tag{21}$$

$\square$

## D.4   Proof of Lemma A.3

**Stage 1 Leaf-to-Relay Push**   We have:

$$\mathbb{E}\left[\left\|\bar{x}_{t+\frac{2}{4}} - \bar{x}_{t+\frac{1}{4}}\right\|^2\right] = \frac{\beta_{lr}}{n_l n_r} \sum_i \mathbb{E}\left[\left\|x_{t+\frac{1}{4}}^{(i)} - \bar{x}_{t+\frac{1}{4}}\right\|^2\right]$$

*Proof.* Note that we can expand the norm as follows:

$$\mathbb{E}\left[\|\bar{y} - \bar{x}\|^2\right] = \mathbb{E}\left[\left\|\frac{1}{n}\sum_i y^{(i)} - \bar{x}\right\|^2\right]$$

$$= \frac{1}{n^2}\sum_i \mathbb{E}\left[\left\|y^{(i)} - \bar{x}\right\|^2\right] + \frac{1}{n^2}\sum_{i \neq j}\mathbb{E}\left[\left\langle y^{(i)} - \bar{x}, y^{(j)} - \bar{x}\right\rangle\right] \quad (22)$$

For the first stage of communication from leaves to relays, we denote $x_{t+\frac{2}{4}}$ as $y$ and $x_{t+\frac{1}{4}}$ as $x$, replacing $n$ with $n_r$. For the $i$-th relay, we have:

$$x_{t+\frac{2}{4}}^{(i)} - \bar{x}_{t+\frac{1}{4}} = \frac{1}{b_{lr}}\sum_k \mathcal{I}_k^{(i)}(x_{t+\frac{1}{4}}^{(k)} - \bar{x}_{t+\frac{1}{4}})$$

where $\mathcal{I}_k^{(i)}$ is an indicator function that represents the connectivity between relay $i$ and leaf $k$. We can thus write the second term in (22) as

$$\frac{1}{n^2}\sum_{i \neq j}\mathbb{E}\left[\langle y^{(i)} - \bar{x}, y^{(j)} - \bar{x}\rangle\right] = \frac{1}{n_r^2}\sum_{\substack{i \in [n_r] \\ i \neq j}}\mathbb{E}\left[\langle x_{t+\frac{2}{4}}^{(i)} - \bar{x}_{t+\frac{1}{4}}, x_{t+\frac{2}{4}}^{(j)} - \bar{x}_{t+\frac{1}{4}}\rangle\right]$$

$$= \frac{2}{n_r^2}\sum_{i \neq j}\sum_{k \in [n_l]}\sum_{l \in [n_l]}\mathbb{E}\left[\frac{\mathcal{I}_k^{(i)}\mathcal{I}_l^{(j)}}{b_{lr}^2}\langle x_{t+\frac{1}{4}}^{(k)} - \bar{x}_{t+\frac{1}{4}}, x_{t+\frac{1}{4}}^{(l)} - \bar{x}_{t+\frac{1}{4}}\rangle\right]$$

$$(23)$$

Now note that by symmetry, for any $i, j \in [n_r]$, we have,

$$\mathbb{E}\left[\mathcal{I}_k^{(i)}\mathcal{I}_l^{(j)}\right] = \mathbb{E}\left[\mathcal{I}_3^{(1)}\mathcal{I}_4^{(2)}\right]$$

This implies that all three terms in (23) can be written as,

$$c \cdot \mathbb{E}\left[\sum_{k \in [n_l]}\sum_{l \in [n_l]}\left\langle x_{t+\frac{1}{4}}^{(k)} - \bar{x}_{t+\frac{1}{4}}, x_{t+\frac{1}{4}}^{(l)} - \bar{x}_{t+\frac{1}{4}}\right\rangle\right]$$

$$= c \cdot \mathbb{E}\left[\sum_l \left\|x_{t+\frac{1}{4}}^{(l)} - \bar{x}_{t+\frac{1}{4}}\right\|^2 + \sum_{k \neq l}\left\langle x_{t+\frac{1}{4}}^{(k)} - \bar{x}_{t+\frac{1}{4}}, x_{t+\frac{1}{4}}^{(l)} - \bar{x}_{t+\frac{1}{4}}\right\rangle\right]$$

$$= c \cdot \mathbb{E}\left[\sum_l \left\|x_{t+\frac{1}{4}}^{(l)} - \bar{x}_{t+\frac{1}{4}}\right\|^2 - \sum_l \left\|x_{t+\frac{1}{4}}^{(l)} - \bar{x}_{t+\frac{1}{4}}\right\|^2\right]$$

$$= 0$$

Therefore, from equation (22), we obtain

$$\mathbb{E}\left[\left\|\bar{x}_{t+\frac{2}{4}} - \bar{x}_{t+\frac{1}{4}}\right\|^2\right] = \frac{1}{n_r^2}\sum_i \mathbb{E}\left[\left\|x_{t+\frac{2}{4}}^{(i)} - \bar{x}_{t+\frac{1}{4}}\right\|^2\right]$$

$$= \frac{\beta_{lr}}{n_l n_r}\sum_i \mathbb{E}\left[\left\|x_{t+\frac{1}{4}}^{(i)} - \bar{x}_{t+\frac{1}{4}}\right\|^2\right] \quad (24)$$

where we make use of (14).

$$\square$$

**Stage 2 mixing: Relay Gossip** Here we prove that

$$\mathbb{E}\left[\left\|x_{t+\frac{3}{4}} - \bar{x}_{t+\frac{2}{4}}\right\|^2\right] \le \frac{\beta_{rr}}{n_r^2} \sum_{i\in[n_r]} \left\|x_{t+\frac{2}{4}}^{(i)} - \bar{x}_{t+\frac{2}{4}}\right\|^2$$

*Proof.* For the second stage of aggregation, we have:

$$\mathbb{E}\left[\left\|x_{t+\frac{3}{4}} - \bar{x}_{t+\frac{2}{4}}\right\|^2\right] = \mathbb{E}\left[\left\|\frac{1}{n_r} \sum_{i\in[n_r]} \left(x_{t+\frac{3}{4}}^{(i)} - \bar{x}_{t+\frac{2}{4}}\right)\right\|^2\right]$$

$$= \frac{1}{n_r^2} \sum_{i\in[n_r]} \mathbb{E}\left[\left\|x_{t+\frac{3}{4}}^{(i)} - \bar{x}_{t+\frac{2}{4}}\right\|^2\right] + \frac{1}{n_r^2} \sum_{i\neq j} \mathbb{E}\left[\left\langle x_{t+\frac{3}{4}}^{(i)} - \bar{x}_{t+\frac{2}{4}}, x_{t+\frac{3}{4}}^{(j)} - \bar{x}_{t+\frac{2}{4}}\right\rangle\right]$$

$$\tag{25}$$

Now recall that

$$x_{t+\frac{3}{4}}^{(i)} - \bar{x}_{t+\frac{2}{4}} = \frac{1}{A^{(i)}+1}\left((x_{t+\frac{2}{4}}^{(i)} - \bar{x}_{t+\frac{2}{4}}) + \sum_{k\in[n_r]\setminus\{i\}} \mathcal{I}_k^{(i)}(x_{t+\frac{2}{4}}^{(k)} - \bar{x}_{t+\frac{2}{4}})\right)$$

This implies that

$$\frac{1}{n_r^2} \sum_{i\neq j} \mathbb{E}\left[\left\langle x_{t+\frac{3}{4}}^{(i)} - \bar{x}_{t+\frac{2}{4}}, x_{t+\frac{3}{4}}^{(j)} - \bar{x}_{t+\frac{2}{4}}\right\rangle\right] \tag{26}$$

$$= \frac{1}{n_r^2} \sum_{i\in[n_r]} \sum_{j\neq i} \mathbb{E}\left[\frac{1}{(A^{(i)}+1)(A^{(j)}+1)} \left\langle x_{t+\frac{2}{4}}^{(i)} - \bar{x}_{t+\frac{2}{4}}, x_{t+\frac{2}{4}}^{(j)} - \bar{x}_{t+\frac{2}{4}}\right\rangle\right]$$

$$+ \frac{2}{n_r^2} \sum_{i\in[n_r]} \sum_{j\neq i} \sum_{k\neq i, k\neq j} \mathbb{E}\left[\frac{\mathcal{I}_k^{(i)}}{(A^{(i)}+1)(A^{(j)}+1)} \left\langle x_{t+\frac{2}{4}}^{(k)} - \bar{x}_{t+\frac{2}{4}}, x_{t+\frac{2}{4}}^{(j)} - \bar{x}_{t+\frac{2}{4}}\right\rangle\right]$$

$$+ \frac{2}{n_r^2} \sum_{i\in[n_r]} \sum_{j\neq i} \sum_{k\neq i, k\neq j} \sum_{l\neq i, l\neq j, l\neq k} \mathbb{E}\left[\frac{\mathcal{I}_k^{(i)}\mathcal{I}_l^{(j)}}{(A^{(i)}+1)(A^{(j)}+1)} \left\langle x_{t+\frac{2}{4}}^{(k)} - \bar{x}_{t+\frac{2}{4}}, x_{t+\frac{2}{4}}^{(l)} - \bar{x}_{t+\frac{2}{4}}\right\rangle\right]$$

$$\tag{27}$$

Now note that by symmetry, for any $i, j \in [n_r]$, we have

$$\mathbb{E}\left[\frac{1}{(A^{(i)}+1)(A^{(j)}+1)}\right] = \mathbb{E}\left[\frac{1}{(A^{(1)}+1)(A^{(2)}+1)}\right]$$

Similarly

$$\mathbb{E}\left[\frac{\mathcal{I}_k^{(i)}}{(A^{(i)}+1)(A^{(j)}+1)}\right] = \mathbb{E}\left[\frac{\mathcal{I}_3^{(1)}}{(A^{(1)}+1)(A^{(2)}+1)}\right]$$

and

$$\mathbb{E}\left[\frac{\mathcal{I}_k^{(i)}\mathcal{I}_l^{(j)}}{(A^{(i)}+1)(A^{(j)}+1)}\right] = \mathbb{E}\left[\frac{\mathcal{I}_3^{(1)}\mathcal{I}_4^{(2)}}{(A^{(1)}+1)(A^{(2)}+1)}\right]$$

This implies that all three terms in (27) can be written as

$$c \sum_{i \in [n]} \sum_{j \neq i} \langle x_{t+\frac{2}{4}}^{(i)} - \bar{x}_{t+\frac{2}{4}}, x_{t+\frac{2}{4}}^{(j)} - \bar{x}_{t+\frac{2}{4}} \rangle$$

where c is a positive constant. We also have

$$\sum_{i \in [n_r]} \sum_{j \neq i} \langle x_{t+\frac{2}{4}}^{(i)} - \bar{x}_{t+\frac{2}{4}}, x_{t+\frac{2}{4}}^{(j)} - \bar{x}_{t+\frac{2}{4}} \rangle = \sum_{i \in [n_r]} \left\langle x_{t+\frac{2}{4}}^{(i)} - \bar{x}_{t+\frac{2}{4}}, \sum_{j \neq i} (x_{t+\frac{2}{4}}^{(j)} - \bar{x}_{t+\frac{2}{4}}) \right\rangle$$

$$= - \sum_{i \in [n_r]} \left\| x_{t+\frac{2}{4}}^{(i)} - \bar{x}_{t+\frac{2}{4}} \right\|^2.$$

Therefore all the terms in (27) are non-positive. Combining this with (25), we obtain that

$$\mathbb{E}\left[ \left\| x_{t+\frac{3}{4}} - \bar{x}_{t+\frac{2}{4}} \right\|^2 \right] \leq \frac{1}{n_r^2} \sum_{i \in [n_r]} \mathbb{E}\left[ \left\| x_{t+\frac{3}{4}}^{(i)} - \bar{x}_{t+\frac{2}{4}} \right\|^2 \right]$$

$$\leq \frac{\beta_{rr}}{n_r} \cdot \frac{1}{n_r} \sum_{i \in [n_r]} \left\| x_{t+\frac{2}{4}}^{(i)} - \bar{x}_{t+\frac{2}{4}} \right\|^2$$

where the second inequality uses (19). Combining this with Lemma A.6 then concludes the proof. □

**Stage 3 mixing: Relay-to-Leaf Pull**   Here we have:

$$\mathbb{E}\left[ \left\| \bar{x}_{t+1} - \bar{x}_{t+\frac{3}{4}} \right\|^2 \right] = \frac{\beta_{rl}}{n_l n_r} \sum_i \mathbb{E}\left[ \left\| x_{t+\frac{3}{4}}^{(i)} - \bar{x}_{t+\frac{3}{4}} \right\|^2 \right]$$

*Proof.* Note that we can expand the norm as follows:

$$\mathbb{E}\left[ \| \bar{y} - \bar{x} \|^2 \right] = \mathbb{E}\left[ \left\| \frac{1}{n} \sum_i y^{(i)} - \bar{x} \right\|^2 \right]$$

$$= \frac{1}{n^2} \sum_i \mathbb{E}\left[ \left\| y^{(i)} - \bar{x} \right\|^2 \right] + \frac{1}{n^2} \sum_{i \neq j} \mathbb{E}\left[ \left\langle y^{(i)} - \bar{x}, y^{(j)} - \bar{x} \right\rangle \right] \quad (28)$$

For the third stage of communication from relays to leaves, we denote $x_{t+1}$ as $y$ and $x_{t+\frac{3}{4}}$ as $x$, replacing $n$ with $n_l$. For the $i$-th leaf, we have:

$$x_{t+1}^{(i)} - \bar{x}_{t+\frac{3}{4}} = \frac{1}{b_{rl}} \sum_k \mathcal{I}_k^{(i)} (x_{t+\frac{3}{4}}^{(k)} - \bar{x}_{t+\frac{3}{4}})$$

where $\mathcal{I}_k^{(i)}$ is an indicator function that represents the connectivity between leaf $i$ and relay $k$. We can thus write the second term in (28) as

$$\frac{1}{n^2} \sum_{i \neq j} \mathbb{E}\left[ \left\langle y^{(i)} - \bar{x}, y^{(j)} - \bar{x} \right\rangle \right] = \frac{1}{n_l^2} \sum_{\substack{i \in [n_l] \\ i \neq j}} \mathbb{E}\left[ \left\langle x_{t+1}^{(i)} - \bar{x}_{t+\frac{3}{4}}, x_{t+1}^{(j)} - \bar{x}_{t+\frac{3}{4}} \right\rangle \right]$$

$$= \frac{2}{n_l^2} \sum_{i \neq j} \sum_{k \in [n_r]} \sum_{l \in [n_r]} \mathbb{E}\left[ \frac{\mathcal{I}_k^{(i)} \mathcal{I}_l^{(j)}}{b_{rl}^2} \langle x_{t+\frac{3}{4}}^{(k)} - \bar{x}_{t+\frac{3}{4}}, x_{t+\frac{3}{4}}^{(l)} - \bar{x}_{t+\frac{3}{4}} \rangle \right]$$

$$(29)$$

Now note that by symmetry, for any $i, j \in [n_l]$, we have,

$$\mathbb{E}\left[\mathcal{I}_k^{(i)} \mathcal{I}_l^{(j)}\right] = \mathbb{E}\left[\mathcal{I}_3^{(1)} \mathcal{I}_4^{(2)}\right]$$

This implies that all three terms in (29) can be written as,

$$c \cdot \mathbb{E}\left[\sum_{k\in[n_r]}\sum_{l\in[n_r]} \left\langle x_{t+\frac{3}{4}}^{(k)} - \bar{x}_{t+\frac{3}{4}}, x_{t+\frac{3}{4}}^{(l)} - \bar{x}_{t+\frac{3}{4}} \right\rangle\right]$$

$$= c \cdot \mathbb{E}\left[\sum_l \left\|x_{t+\frac{3}{4}}^{(l)} - \bar{x}_{t+\frac{3}{4}}\right\|^2 + \sum_{k\neq l} \left\langle x_{t+\frac{3}{4}}^{(k)} - \bar{x}_{t+\frac{3}{4}}, x_{t+\frac{3}{4}}^{(l)} - \bar{x}_{t+\frac{3}{4}} \right\rangle\right]$$

$$= c \cdot \mathbb{E}\left[\sum_l \left\|x_{t+\frac{3}{4}}^{(l)} - \bar{x}_{t+\frac{3}{4}}\right\|^2 - \sum_l \left\|x_{t+\frac{3}{4}}^{(l)} - \bar{x}_{t+\frac{3}{4}}\right\|^2\right]$$

$$= 0$$

Therefore, from equation (28), we obtain

$$\mathbb{E}\left[\left\|\bar{x}_{t+1} - \bar{x}_{t+\frac{3}{4}}\right\|^2\right] = \frac{1}{n_l^2} \sum_i \mathbb{E}\left[\left\|x_{t+1}^{(i)} - \bar{x}_{t+\frac{3}{4}}\right\|^2\right]$$

$$= \frac{\beta_{rl}}{n_l n_r} \sum_i \mathbb{E}\left[\left\|x_{t+\frac{3}{4}}^{(i)} - \bar{x}_{t+\frac{3}{4}}\right\|^2\right] \qquad (30)$$

where we make use of (20). $\qquad \square$

### D.5 PROOF OF LEMMA A.4

The expected consensus distance and gradient variance across leaves are bounded as follows:

1. **Consensus Distance Bound:**
$$\frac{1}{n_l^2} \sum_{i,j\in[n_l]} \mathbb{E}\left[\left\|x_t^{(i)} - x_t^{(j)}\right\|^2\right] \leq 20\frac{1+3\beta_{TGL}}{(1-\beta_{TGL})^2}\beta_{TGL}\gamma^2(\sigma^2 + \mathcal{H}^2).$$

2. **Gradient Variance Bound:**
$$\frac{1}{n_l^2} \sum_{i,j\in[n_l]} \mathbb{E}\left[\left\|g_t^{(i)} - g_t^{(j)}\right\|^2\right] \leq 15(\sigma^2 + \mathcal{H}^2).$$

*Proof.* For any $i \in [n_l]$, we have

$$g_t^{(i)} - g_t^{(j)} = g_t^{(i)} - \nabla f^{(i)}\left(x_t^{(i)}\right) + \nabla f^{(i)}\left(x_t^{(i)}\right) - \nabla f^{(i)}(\bar{x}_t) + \nabla f^{(j)}(\bar{x}_t)$$

$$- \nabla f^{(j)}(\bar{x}_t) + \nabla f^{(j)}(\bar{x}_t) - \nabla f^{(j)}\left(x_t^{(j)}\right) + \nabla f^{(j)}\left(x_t^{(j)}\right) - g_t^{(j)}$$

where $g_t$ is the stochastic version of $\nabla f^{(i)}\left(x_t^{(i)}\right)$. Thus, using Jensens's inequality, we have,

$$\left\|g_t^{(i)} - g_t^{(j)}\right\|^2 \leq 5\left\|g_t^{(i)} - \nabla f^{(i)}\left(x_t^{(i)}\right)\right\|^2 + 5\left\|\nabla f^{(i)}\left(x_t^{(i)}\right) - \nabla f^{(i)}(\bar{x}_t)\right\|^2$$

$$+ 5\left\|\nabla f^{(j)}\left(x_t^{(j)}\right) - \nabla f^{(j)}(\bar{x}_t)\right\|^2 + 5\left\|g_t^{(j)} - \nabla f^{(j)}\left(x_t^{(j)}\right)\right\|^2$$

$$+ 5\left\|\nabla f^{(i)}(\bar{x}_t) - \nabla f^{(j)}(\bar{x}_t)\right\|^2$$

Taking the conditional expectation, we have,

$$
\mathbb{E}_t \left[ \left\| g_t^{(i)} - g_t^{(j)} \right\|^2 \right] \leq 5\mathbb{E}_t \left[ \left\| g_t^{(i)} - \nabla f^{(i)} \left( x_t^{(i)} \right) \right\|^2 \right] + 5\mathbb{E}_t \left[ \left\| \nabla f^{(i)} \left( x_t^{(i)} \right) - \nabla f^{(i)} \left( \bar{x}_t \right) \right\|^2 \right]
$$
$$
+ 5\mathbb{E}_t \left[ \left\| \nabla f^{(j)} \left( x_t^{(j)} \right) - \nabla f^{(j)} \left( \bar{x}_t \right) \right\|^2 \right] + 5\mathbb{E}_t \left[ \left\| g_t^{(j)} - \nabla f^{(j)} \left( x_t^{(j)} \right) \right\|^2 \right]
$$
$$
+ 5\mathbb{E}_t \left[ \left\| \nabla f^{(i)} \left( \bar{x}_t \right) - \nabla f^{(j)} \left( \bar{x}_t \right) \right\|^2 \right] \tag{31}
$$

Now by Assumption 4.2, we have,

$$
\mathbb{E}_t \left[ \left\| g_t^{(i)} - \nabla f^{(i)} \left( x_t^{(i)} \right) \right\|^2 \right] \leq \sigma^2 \tag{32}
$$

By Assumption 4.1, we have,

$$
\mathbb{E}_t \left[ \left\| \nabla f^{(i)} \left( x_t^{(i)} \right) - \nabla f^{(i)} \left( \bar{x}_t \right) \right\|^2 \right] \leq L^2 \mathbb{E}_t \left[ \left\| x_t^{(i)} - \bar{x}_t \right\|^2 \right] \tag{33}
$$

Thus, by Assumption 4.3, and Remark A.6, we obtain that,

$$
\frac{1}{n_l^2} \sum_{i,j \in [n_l]} \mathbb{E}_t \left[ \left\| \nabla f^{(i)}(\bar{x}_t) - \nabla f^{(j)}(\bar{x}_t) \right\|^2 \right] \leq 2\mathcal{H}^2 \tag{34}
$$

Combining (31), (32), (33), and (34), and taking total expectation from both sides, we obtain that,

$$
\frac{1}{n_l^2} \sum_{i,j \in [n_r]} \mathbb{E} \left[ \left\| g_t^{(i)} - g_t^{(j)} \right\|^2 \right] \leq \frac{10L^2}{n_l} \sum_{i \in [n_l]} \mathbb{E} \left[ \left\| x_t^{(i)} - \bar{x}_t \right\|^2 \right] + 10\sigma^2 + 10\mathcal{H}^2 \tag{35}
$$

Now Remark A.6 yields

$$
\frac{1}{n_l^2} \sum_{i,j \in [n_l]} \mathbb{E} \left[ \left\| g_t^{(i)} - g_t^{(j)} \right\|^2 \right] \leq \frac{5L^2}{n_l^2} \sum_{i \in [n_l]} \mathbb{E} \left[ \left\| x_t^{(i)} - x_t^{(j)} \right\|^2 \right] + 10\sigma^2 + 10\mathcal{H}^2 \tag{36}
$$

We now analyze $\frac{1}{n_l^2} \sum_{i,j \in [n_l]} \mathbb{E} \left[ \left\| x_t^{(i)} - x_t^{(j)} \right\|^2 \right]$. From Algorithm 1, recall that for all $i \in [n_l]$, we have $x_{t+1/4}^{(i)} = x_t^{(i)} - \gamma g_t^{(i)}$. We obtain for all $i, j \in [n_l]$, that

$$
\mathbb{E} \left[ \left\| x_{t+\frac{1}{4}}^{(i)} - x_{t+\frac{1}{4}}^{(j)} \right\|^2 \right] \leq \mathbb{E} \left[ \left\| x_t^{(i)} - x_t^{(j)} - \gamma(g_t^{(i)} - g_t^{(j)}) \right\|^2 \right]
$$
$$
\leq (1+c)\mathbb{E} \left[ \left\| x_t^{(i)} - x_t^{(j)} \right\|^2 \right] + \left( 1 + \frac{1}{c} \right) \gamma^2 \mathbb{E} \left[ \left\| g_t^{(i)} - g_t^{(j)} \right\|^2 \right] \tag{37}
$$

From (17), we have

$$
\frac{1}{n_r^2} \sum_{i,j} \mathbb{E} \left[ \left\| x_{t+\frac{2}{4}}^{(i)} - x_{t+\frac{2}{4}}^{(j)} \right\|^2 \right] \leq \frac{\beta_{lr}}{n_l^2} \sum_{i,j} \mathbb{E} \left[ \left\| x_{t+\frac{1}{4}}^{(i)} - x_{t+\frac{1}{4}}^{(j)} \right\|^2 \right]
$$

Combining with (37), we get

$$\frac{1}{n_r^2} \sum_{i,j} \mathbb{E}\left[\left\|x_{t+\frac{2}{4}}^{(i)} - x_{t+\frac{2}{4}}^{(j)}\right\|^2\right] \leq (1+c)\frac{\beta_{lr}}{n_l^2} \sum_{i,j} \mathbb{E}\left[\left\|x_t^{(i)} - x_t^{(j)}\right\|^2\right]$$

$$+ \left(1+\frac{1}{c}\right)\gamma^2\frac{\beta_{lr}}{n_l^2} \sum_{i,j} \mathbb{E}\left[\left\|g_t^{(i)} - g_t^{(j)}\right\|^2\right] \quad (38)$$

From (19), we also have

$$\frac{1}{n_r^2} \sum_{i,j} \mathbb{E}\left[\left\|x_{t+\frac{3}{4}}^{(i)} - x_{t+\frac{3}{4}}^{(j)}\right\|^2\right] \leq \frac{\beta_{rr}}{n_r^2} \sum_{i,j} \mathbb{E}\left[\left\|x_{t+\frac{2}{4}}^{(i)} - x_{t+\frac{2}{4}}^{(j)}\right\|^2\right]$$

We substitute (38) here to get:

$$\frac{1}{n_r^2} \sum_{i,j} \mathbb{E}\left[\left\|x_{t+\frac{3}{4}}^{(i)} - x_{t+\frac{3}{4}}^{(j)}\right\|^2\right] \leq (1+c)\frac{\beta_{lr}\beta_{rr}}{n_l^2} \sum_{i,j} \mathbb{E}\left[\left\|x_t^{(i)} - x_t^{(j)}\right\|^2\right] \quad (39)$$

$$+ \left(1+\frac{1}{c}\right)\gamma^2\frac{\beta_{lr}\beta_{rr}}{n_l^2} \sum_{i,j} \mathbb{E}\left[\left\|g_t^{(i)} - g_t^{(j)}\right\|^2\right] \quad (40)$$

Also, from (21), we obtain

$$\frac{1}{n_l^2} \sum_{i,j} \mathbb{E}\left[\left\|x_{t+1}^{(i)} - x_{t+1}^{(j)}\right\|^2\right] \leq \frac{\beta_{lr}}{n_l^2} \sum_{i,j} \mathbb{E}\left[\left\|x_{t+\frac{3}{4}}^{(i)} - x_{t+\frac{3}{4}}^{(j)}\right\|^2\right]$$

We combine this with (40) to get

$$\frac{1}{n_l^2} \sum_{i,j} \mathbb{E}\left[\left\|x_{t+1}^{(i)} - x_{t+1}\right\|^2\right] \leq (1+c)\frac{\beta_{TGL}}{n_l^2} \sum_{i,j} \mathbb{E}\left[\left\|x_t^{(i)} - x_t^{(j)}\right\|^2\right]$$

$$+ \left(1+\frac{1}{c}\right)\gamma^2\frac{\beta_{TGL}}{n_l^2} \sum_{i,j} \mathbb{E}\left[\left\|g_t^{(i)} - g_t^{(j)}\right\|^2\right]$$

For $c = \frac{1-\beta_{TGL}}{4\beta_{TGL}}$, we obtain that,

$$\frac{1}{n_l^2} \sum_{i,j \in [n_l]} \mathbb{E}\left[\left\|x_{t+1}^{(i)} - x_{t+1}^{(j)}\right\|^2\right] \leq \frac{1+3\beta_{TGL}}{4}\frac{1}{n_l^2} \sum_{i,j \in [n_l]} \mathbb{E}\left[\left\|x_t^{(i)} - x_t^{(j)}\right\|^2\right]$$

$$+ \frac{1+3\beta_{TGL}}{1-\beta_{TGL}}\beta_{TGL}\gamma^2\frac{1}{n_l^2} \sum_{i,j \in [n_l]} \mathbb{E}\left[\left\|g_t^{(i)} - g_t^{(j)}\right\|^2\right]$$

Combining this with (36), we obtain that

$$\frac{1}{n_l^2} \sum_{i,j \in [n_l]} \mathbb{E}\left[\left\|x_{t+1}^{(i)} - x_{t+1}^{(j)}\right\|^2\right] \leq \frac{1+3\beta_{TGL}}{4}\frac{1}{n_l^2} \sum_{i,j \in [n_l]} \mathbb{E}\left[\left\|x_t^{(i)} - x_t^{(j)}\right\|^2\right]$$

$$+ \frac{1+3\beta_{TGL}}{1-\beta_{TGL}}\beta_{TGL}\gamma^2\left(\frac{5L^2}{n_l^2} \sum_{i,j \in [n_l]} \mathbb{E}\left[\left\|x_t^{(i)} - x_t^{(j)}\right\|^2\right] + 10\sigma^2 + 10\mathcal{H}^2\right)$$

$$= \left(\frac{1+3\beta_{TGL}}{4} + 5\frac{1+3\beta_{TGL}}{1-\beta_{TGL}}\beta_{TGL}\gamma^2 L^2\right)\frac{1}{n_l^2} \sum_{i,j \in [n_l]} \mathbb{E}\left[\left\|x_t^{(i)} - x_t^{(j)}\right\|^2\right]$$

$$+ \frac{1+3\beta_{TGL}}{1-\beta_{TGL}}\beta_{TGL}\gamma^2(10\sigma^2 + 10\mathcal{H}^2).$$

Now note that from Remark A.7 we have $\beta_{TGL} \leq 1 - \frac{1}{e}$ which implies that

$$\gamma^2 \leq \frac{1}{(20L)^2} \leq \frac{(1 - \beta_{TGL})^2}{20\beta_{TGL}(1 + 3\beta_{TGL})L^2}$$

Therefore,

$$\frac{1}{n_l^2} \sum_{i,j \in [n_l]} \mathbb{E}\left[\left\|x_{t+1}^{(i)} - x_{t+1}^{(j)}\right\|^2\right] \leq \frac{1 + \beta_{TGL}}{2} \frac{1}{n_l^2} \sum_{i,j \in [n_l]} \mathbb{E}\left[\left\|x_t^{(i)} - x_t^{(j)}\right\|^2\right] + \frac{1 + 3\beta_{TGL}}{1 - \beta_{TGL}}\beta_{TGL}\gamma^2(10\sigma^2 + 10\mathcal{H}^2).$$

Unrolling the recursion, we obtain that,

$$\frac{1}{n_l^2} \sum_{i,j \in [n_l]} \mathbb{E}\left[\left\|x_t^{(i)} - x_t^{(j)}\right\|^2\right] \leq 20\frac{1 + 3\beta_{TGL}}{(1 - \beta_{TGL})^2}\beta_{TGL}\gamma^2(\sigma^2 + \mathcal{H}^2).$$

Combining this with (35), we obtain that,

$$\frac{1}{n_l^2} \sum_{i,j \in [n_l]} \mathbb{E}\left[\left\|g_t^{(i)} - g_t^{(j)}\right\|^2\right] \leq 15(\sigma^2 + \mathcal{H}^2).$$

$\square$

### D.5.1 PROOF OF LEMMA A.5

The expected gradient norm of the global objective satisfies the following upper bound:

$$\mathbb{E}\left[\|\nabla F(\bar{x}_t)\|^2\right] \leq \frac{2}{\gamma}\mathbb{E}\left[F(\bar{x}_t) - F(\bar{x}_{t+1})\right] + \frac{L}{2n_l^2}\sum_{i,j}\mathbb{E}\left[\left\|x_t^{(i)} - x_t^{(j)}\right\|^2\right] + \frac{4L\gamma\sigma^2}{n_l}$$
$$+ \frac{4L}{\gamma}\mathbb{E}\left[\left\|\bar{x}_{t+1} - \bar{x}_{t+\frac{3}{4}}\right\|^2 + \left\|\bar{x}_{t+\frac{3}{4}} - \bar{x}_{t+\frac{2}{4}}\right\|^2 + \left\|\bar{x}_{t+\frac{2}{4}} - \bar{x}_{t+\frac{1}{4}}\right\|^2\right].$$

*Proof.* Consider an arbitrary $t \in [T]$. Then from the smoothness property, we have:

$$F(\bar{x}_{t+1}) - F(\bar{x}_t) \leq \langle \bar{x}_{t+1} - \bar{x}_t, \nabla F(\bar{x}_t)\rangle + \frac{L}{2}\|\bar{x}_{t+1} - \bar{x}_t\|^2$$
$$= \langle \bar{x}_{t+1} - \bar{x}_{t+\frac{3}{4}} + \bar{x}_{t+\frac{3}{4}} - \bar{x}_{t+\frac{2}{4}} + \bar{x}_{t+\frac{2}{4}} - \bar{x}_{t+\frac{1}{4}} + \bar{x}_{t+\frac{1}{4}} - \bar{x}_t, \nabla F(\bar{x}_t)\rangle$$
$$+ \frac{L}{2}\|\bar{x}_{t+1} - \bar{x}_{t+\frac{3}{4}} + \bar{x}_{t+\frac{3}{4}} - \bar{x}_{t+\frac{2}{4}} + \bar{x}_{t+\frac{2}{4}} - \bar{x}_{t+\frac{1}{4}} + \bar{x}_{t+\frac{1}{4}} - \bar{x}_t\|^2. \quad (41)$$

From Lemma A.1, we have

$$\mathbb{E}\left[\bar{x}_{t+1}\right] = \mathbb{E}\left[\bar{x}_{t+\frac{3}{4}}\right] = \mathbb{E}\left[\bar{x}_{t+\frac{2}{4}}\right] = \mathbb{E}\left[\bar{x}_{t+\frac{1}{4}}\right].$$

Now, we take conditional expectation on (41) and use Lemma A.1 to get

$$\mathbb{E}_t\left[F(\bar{x}_{t+1}) - F(\bar{x}_t)\right] \leq \langle \mathbb{E}_t\left[\bar{x}_{t+1} - \bar{x}_t\right], \nabla F(\bar{x}_t)\rangle$$
$$+ \frac{L}{2}\mathbb{E}_t\left[\|\bar{x}_{t+1} - \bar{x}_{t+\frac{3}{4}} + \bar{x}_{t+\frac{3}{4}} - \bar{x}_{t+\frac{2}{4}} + \bar{x}_{t+\frac{2}{4}} - \bar{x}_{t+\frac{1}{4}} + \bar{x}_{t+\frac{1}{4}} - \bar{x}_t\|^2\right]$$
$$\leq -\gamma\langle \overline{\nabla F}_t, \nabla F(\bar{x}_t)\rangle + 2L\gamma^2\mathbb{E}_t\left[\|\bar{g}_t\|^2\right] + 2L\mathbb{E}_t\left[\left\|\bar{x}_{t+1} - \bar{x}_{t+\frac{3}{4}}\right\|^2\right]$$
$$+ 2L\mathbb{E}_t\left[\left\|\bar{x}_{t+\frac{3}{4}} - \bar{x}_{t+\frac{2}{4}}\right\|^2\right] + 2L\mathbb{E}_t\left[\left\|\bar{x}_{t+\frac{2}{4}} - \bar{x}_{t+\frac{1}{4}}\right\|^2\right] \quad (42)$$

where $\overline{\nabla F}_t = \frac{1}{n_l}\sum_{i\in[n_l]}\nabla f^{(i)}(x_t^{(i)})$, $\bar{g}_t = \frac{1}{n_l}\sum_{i\in[n_l]}g_t^{(i)}$, and we make use of Jensen's inequality. Then we use the $\mathbb{E}_t[\bar{g}_t] = \overline{\nabla F}_t$, and $\mathbb{E}_t[\|\bar{g}_t - \overline{\nabla F}_t\|^2] \le \frac{\sigma^2}{n_l}$

$$
\mathbb{E}_t\left[F(\bar{x}_{t+1}) - F(\bar{x}_t)\right] \le -\gamma\langle\overline{\nabla F}_t, \nabla F(\bar{x}_t)\rangle + 2L\gamma^2\mathbb{E}\left[\|\nabla F(\bar{x}_t)\|^2\right] + 2L\frac{\gamma^2\sigma^2}{n_l} + 2L\,\mathbb{E}_t\left[\left\|\bar{x}_{t+1} - \bar{x}_{t+\frac{3}{4}}\right\|^2\right]
$$
$$
+ 2L\,\mathbb{E}_t\left[\left\|\bar{x}_{t+\frac{3}{4}} - \bar{x}_{t+\frac{2}{4}}\right\|^2\right] + 2L\,\mathbb{E}_t\left[\left\|\bar{x}_{t+\frac{2}{4}} - \bar{x}_{t+\frac{1}{4}}\right\|^2\right] \tag{43}
$$

Then we use $\gamma \le \frac{1}{4L}$ to get

$$
-\gamma\langle\overline{\nabla F}_t, \nabla F(\bar{x}_t)\rangle + 2L\gamma^2\|\overline{\nabla F}_t\|^2 \le \frac{\gamma}{2}(-2\langle\overline{\nabla F}_t, \nabla F(\bar{x}_t)\rangle + \|\overline{\nabla F}_t\|^2)
$$
$$
= \frac{\gamma}{2}(-\|\nabla F(\bar{x}_t)\|^2 + \|\nabla F(\bar{x}_t) - \overline{\nabla F}_t\|^2). \tag{44}
$$

Combining (43) and (44), we obtain

$$
\mathbb{E}_t\left[F(\bar{x}_{t+1}) - F(\bar{x}_t)\right] \le -\frac{\gamma}{2}\|\nabla F(\bar{x}_t)\|^2 + \frac{\gamma}{2}\|\nabla F(\bar{x}_t) - \bar{\nabla} F_t\|^2 + 2L\frac{\gamma^2\sigma^2}{n_l}
$$
$$
+ 2L\mathbb{E}_t\left[\left\|\bar{x}_{t+1} - \bar{x}_{t+\frac{3}{4}}\right\|^2\right] + 2L\mathbb{E}_t\left[\left\|\bar{x}_{t+\frac{3}{4}} - \bar{x}_{t+\frac{2}{4}}\right\|^2\right] + 2L\mathbb{E}_t\left[\left\|\bar{x}_{t+\frac{2}{4}} - \bar{x}_{t+\frac{1}{4}}\right\|^2\right]
$$

Taking total expectation, we obtain

$$
\mathbb{E}_t\left[F(\bar{x}_{t+1}) - F(\bar{x}_t)\right] \le -\frac{\gamma}{2}\mathbb{E}_t\left[\|\nabla F(\bar{x}_t)\|^2\right] + \frac{\gamma}{2}\mathbb{E}_t\left[\|\nabla F(\bar{x}_t) - \bar{\nabla} F_t\|^2\right] + 2L\frac{\gamma^2\sigma^2}{n_l}
$$
$$
+ 2L\mathbb{E}_t\left[\left\|\bar{x}_{t+1} - \bar{x}_{t+\frac{3}{4}}\right\|^2\right] + 2L\mathbb{E}_t\left[\left\|\bar{x}_{t+\frac{3}{4}} - \bar{x}_{t+\frac{2}{4}}\right\|^2\right] + 2L\mathbb{E}_t\left[\left\|\bar{x}_{t+\frac{2}{4}} - \bar{x}_{t+\frac{1}{4}}\right\|^2\right] \tag{45}
$$

Now, note that

$$
\mathbb{E}\left[\|\overline{\nabla F}_t - \nabla F(\bar{x}_t)\|^2\right] = \mathbb{E}\left[\left\|\frac{1}{n_l}\sum_{i\in[n_l]}\nabla f^{(i)}(x_t^{(i)}) - \frac{1}{n_l}\sum_{i\in[n_l]}\nabla f^{(i)}(\bar{x}_t)\right\|^2\right]
$$
$$
= \mathbb{E}\left[\left\|\frac{1}{n_l}\sum_{i\in[n_l]}\left(\nabla f^{(i)}(x_t^{(i)}) - \nabla f^{(i)}(\bar{x}_t)\right)\right\|^2\right]
$$
$$
\le \frac{1}{n_l}\sum_{i\in[n_l]}\mathbb{E}\left[\left\|\nabla f^{(i)}(x_t^{(i)}) - \nabla f^{(i)}(\bar{x}_t)\right\|^2\right]
$$
$$
\le \frac{L^2}{n_l}\sum_{i\in[n_l]}\mathbb{E}\left[\left\|x_t^{(i)} - \bar{x}_t\right\|^2\right].
$$
$$
\le \frac{L^2}{2n_l^2}\sum_{i,j\in[n_l]}\mathbb{E}\left[\left\|x_t^{(i)} - x_t^{(j)}\right\|^2\right],
$$

where we use Assumption 4.1 and Remark A.6 in the above two inequalities. Now, substituting this back in (45).

$$
\mathbb{E}_t\left[F(\bar{x}_{t+1}) - F(\bar{x}_t)\right] \le -\frac{\gamma}{2}\mathbb{E}_t\left[\|\nabla F(\bar{x}_t)\|^2\right] + \frac{\gamma}{2}\frac{L^2}{2n_l^2}\sum_{i,j\in[n_l]}\mathbb{E}\left[\left\|x_t^{(i)} - x_t^{(j)}\right\|^2\right] + 2L\frac{\gamma^2\sigma^2}{n_l}
$$
$$
+ 2L\mathbb{E}_t\left[\left\|\bar{x}_{t+1} - \bar{x}_{t+\frac{3}{4}}\right\|^2\right] + 2L\mathbb{E}_t\left[\left\|\bar{x}_{t+\frac{3}{4}} - \bar{x}_{t+\frac{2}{4}}\right\|^2\right] + 2L\mathbb{E}_t\left[\left\|\bar{x}_{t+\frac{2}{4}} - \bar{x}_{t+\frac{1}{4}}\right\|^2\right]
$$

Rearranging these terms, we get

$$\mathbb{E}\left[\|\nabla F(\bar{x}_t)\|^2\right] \le \frac{2}{\gamma}\mathbb{E}\left[F(\bar{x}_t) - F(\bar{x}_{t+1})\right] + \frac{L}{2n_l^2}\sum_{i,j}\mathbb{E}\left[\left\|x_t^{(i)} - x_t^{(j)}\right\|^2\right] + \frac{4L\gamma\sigma^2}{n_l}$$

$$+ \frac{4L}{\gamma}\mathbb{E}\left[\left\|\bar{x}_{t+1} - \bar{x}_{t+\frac{3}{4}}\right\|^2 + \left\|\bar{x}_{t+\frac{3}{4}} - \bar{x}_{t+\frac{2}{4}}\right\|^2 + \left\|\bar{x}_{t+\frac{2}{4}} - \bar{x}_{t+\frac{1}{4}}\right\|^2\right]$$

$\square$

### D.6 PROOF OF REMARK A.6

For any set $\{x_t^{(i)}\}_{i\in[n]}$ of $n$ vectors, we have

$$\frac{1}{n}\sum_{i\in[n]}\|x_t^{(i)} - \bar{x}_t\|^2 = \frac{1}{2n^2}\sum_{i,j\in[n]}\|x_t^{(i)} - x_t^{(j)}\|^2,$$

where $\bar{x}_t = \frac{1}{n}\sum_{i\in[n]}x_t^{(i)}$.

**Proof**

$$\frac{1}{n^2}\sum_{i,j\in[n]}\|x_t^{(i)} - x_t^{(j)}\|^2 = \frac{1}{n^2}\sum_{i,j\in[n]}\|(x_t^{(i)} - \bar{x}_t) - (x_t^{(j)} - \bar{x}_t)\|^2$$

$$= \frac{1}{n^2}\sum_{i,j\in[n]}\left[\|x_t^{(i)} - \bar{x}_t\|^2 + \|x_t^{(j)} - \bar{x}_t\|^2 - 2\langle x_t^{(i)} - \bar{x}_t, x_t^{(j)} - \bar{x}_t\rangle\right]$$

$$= \frac{2}{n}\sum_{i\in[n]}\|x_t^{(i)} - \bar{x}_t\|^2 - \frac{2}{n^2}\sum_{i\in[n]}\left\langle x_t^{(i)} - \bar{x}_t, \sum_{j\in[n]}(x_t^{(j)} - \bar{x}_t)\right\rangle.$$

Noting that $\sum_{j\in[n]}(x_t^{(j)} - \bar{x}_t) = 0$, yields the desired result.

## E ADDITIONAL THEORETICAL ANALYSIS

We analyze how the condition $b_{rl}n_l \le b_{lr}n_r$ leads to a tighter upper bound on $\beta_{\text{TGL}}$, thereby ensuring stronger consensus among the leaf models. This inequality implies that the total indegree of relays during the leaf-to-relay (Stage 1) mixing exceeds the total indegree of leaves during the relay-to-leaf (Stage 3) mixing. Intuitively, this allocates a higher communication load to the relays, which they are structurally designed for. Under this condition, the resulting lower $\beta_{TGL}$ translates into faster convergence.

*Proof.*

$$\beta_{lr} = \frac{1}{b_{lr}}\left(1 - \frac{b_{lr} - 1}{n_l - 1}\right) \tag{46}$$

$$\beta_{rr} \le 1 - \frac{1}{e}$$

$$\beta_{rl} \le \frac{1}{b_{rl}}\left(1 - \frac{b_{rl} - 1}{n_r - 1}\right) \tag{47}$$

By combining (46) and (47),

$$\beta_{lr}\beta_{rl} \le \frac{1}{b_{lr}b_{rl}}\frac{n_l - b_{lr}}{n_l - 1}\frac{n_r - b_{rl}}{n_r - 1}$$

$$= \frac{1}{b_{lr}b_{rl}}\frac{(1 - \frac{b_{lr}}{n_l})}{(1 - \frac{1}{n_l})}\frac{(1 - \frac{b_{rl}}{n_r})}{(1 - \frac{1}{n_r})}$$

If we have $b_{rl}n_l \leq b_{lr}n_r$, then we have $\frac{b_{rl}}{n_r} \leq \frac{b_{lr}}{n_l}$. Let $a = \frac{b_{rl}}{n_r}$ and $b = \frac{b_{lr}}{n_l}$. Since we have $a \leq b$, we have $\frac{1-b}{b} \leq \frac{1-a}{a}$. Thus,

$$\beta_{lr}\beta_{rl} \leq \frac{1}{n_r n_l} \frac{(1-a)^2}{a^2} \frac{1}{(1-\frac{1}{n_l})(1-\frac{1}{n_r})}$$

This is decreasing in $a$ and hence in $b_{rl}$. Thus we substitute the lowest leaf budget, $b_{rl} = 1$ to obtain

$$\beta_{lr}\beta_{rl} \leq \frac{1}{n_r n_l} \frac{(1-\frac{1}{n_r})^2}{(\frac{1}{n_r})^2} \frac{1}{(1-\frac{1}{n_l})(1-\frac{1}{n_r})}$$

$$= \frac{n_r - 1}{n_l - 1} < \frac{n_r}{n_l}$$

$$\beta_{TGL} \leq \frac{n_r}{n_l}\left(1 - \frac{1}{e}\right)$$

$\square$

Hence, when the load-balancing condition $n_r b_{lr} \geq n_l b_{rl}$ is satisfied, TGL attains a strictly tighter (smaller) upper bound on $\beta$ than Epidemic Learning, whose $\beta$ is bounded only by the looser estimate $\beta_{\text{EL}} \leq 1 - \frac{1}{e}$.

## F  LLM USAGE

We used large language models solely to assist with rephrasing and improving the clarity of our writing. No LLMs were used for ideation, experimental design, algorithm development, or code creation.

