# OpenReview forum: "Tiered Gossip Learning: Communication-Frugal and Scalable Collaborative Learning"
_ICLR.cc/2026/Conference — Submitted to ICLR 2026_

### Official Review · Reviewer_cy9C · 2025-10-17

**Soundness:** 4
**Presentation:** 3
**Contribution:** 3
**Rating:** 6
**Confidence:** 5

**Summary:**

The paper proposes **Tiered Gossip Learning (TGL)**. It is a two-tier, coordinator-free protocol. Each round has three stages: push (leaf→relay), gossip (relay↔relay), and pull (relay→leaf). The goal is strong mixing with small leaf degrees. The theory shows a contraction with product form \( \beta_{\text{TGL}} = \beta_{lr}\beta_{rr}\beta_{rl} \) and gives convergence under standard assumptions. They also derive a general upper bound \( \beta_{\text{TGL}} \le 1 - 1/e \). Under a relay-heavy “load-balancing’’ condition, the bound is tighter than the Epidemic Learning baseline. Experiments on image, text, and character datasets show better accuracy-per-communication than several decentralized baselines, and often approach FL with similar communication.

**Strengths:**

- The problem is interesting and practical:
- Solution is simple, easy to implement..
- Clean intuition: multiply three sparse mixing steps to get effective dense mixing.
- Theory links budgets to one multiplicative factor \( \beta_{\text{TGL}} \). This gives clear knobs to trade accuracy vs. communication.
- Experiments are broad and mostly convincing; Spectral-gap simulation supports the mixing part.

**Weaknesses:**

- The method likely introduces **significantly higher latency**, since each communication round includes three sequential stages (push, relay gossip, and pull). This means roughly **three times the latency** compared to direct node-to-node exchanges in fully decentralized protocols.

- The **name “Tiered Gossip Learning” (TGL)** can be somewhat confusing. Traditionally, *Gossip Learning* refers to an asynchronous decentralized protocol where nodes train and exchange models independently, as in [A]. In contrast, this work **extends the Epidemic Learning** framework, which already includes synchronized rounds and coordinated model exchanges. The paper should clarify this connection more explicitly and perhaps modify the naming to reflect this.

  [A] Róbert Ormándi, István Hegedüs, and Márk Jelasity. *Gossip learning with linear models on fully distributed data.* Concurrency and Computation: Practice and Experience, 25(4):556–571, 2013.

- The **theoretical contribution mainly analyzes the consensus rate** of the proposed scheme. The rest of the convergence proof closely follows from the Epidemic Learning paper, with minor adaptations. This is not necessarily a weakness, but it should be clearly stated in the main body so readers understand the level of novelty in the analysis.

- The method **assumes synchronous rounds and independent random sampling**. The paper does not discuss how the protocol behaves under **network delays**, **node churn**, or **partial participation**. Including an analysis or at least a discussion of these factors would make the paper more complete.

**Questions:**

Why is EL-Oracle not included in the experiments?

---

> ### Author Response · Authors · 2025-11-18
>
> We sincerely thank the reviewer for their excellent and thorough review. We are especially grateful that you recognized the core intuition of TGL, multiplying three sparse matrices to achieve effective dense mixing, and the practical value of our design’s tunable knobs for scalability. We also deeply appreciate the effort taken to review Appendix E and validate our tighter consensus bounds under load balancing. Your insightful feedback is highly valued, and we address your concerns below.
>
> **W1: On Latency Trade-offs**
>
> We agree that TGL prioritizes communication-frugality and faster convergence (in fewer total rounds) by trading-off the per-round-wall-clock latency as noted in our Discussion (Line 474). This is a deliberate trade-off, but we believe the 3-stage latency is not as high as it might first appear, for two key reasons -
> - The 3-hop process is not a “3x cost” because the relay gossip (stage 2) happens over a small and resource-capable layer of $n_r$ relays, which is much faster than gossiping across the entire flat P2P graph of $n_l$ nodes that can suffer from a large network diameter. Further, the relay layer acts as a shortcut, reducing the network diameter and hence the total latency for equivalent mixing.
> - The reviewer's premise compares 1 TGL round to 1 P2P round. However, our core contribution is mixing efficiency, as shown in all our experiments and CDR plots. To match TGL's mixing quality, a sparse P2P protocol (like ELL) would require multiple gossip rounds per update, which would compound its own latency and negate any single-round advantage.
>
> **W2: On the Terminology "Gossip Learning"**
>
> We respectfully view “Gossip Learning” as a broader descriptor for the paradigm of server-less peer-to-peer learning, rather than being restricted to asynchronous pairwise protocols. Several prior works [1,2,3,4] use the term to describe this paradigm as we have. [1] explicitly states that gossip can be synchronous or asynchronous where [2] uses this term for synchronous protocols. [3] and [4] are foundational works that describe gossip learning as a protocol where multiple nodes (not just a pair of nodes) gossip synchronously to aggregate models.
>
> [1]: Nadiradze et al Asynchronous Decentralized SGD with Quantized and Local Updates [NIPS 2021]
>
> [2]: Vogels et al RelaySum for Decentralized Deep Learning on Heterogeneous Data [NIPS 2021]
>
> [3]: Koloskova et al Decentralized Stochastic Optimization and Gossip Algorithms with Compressed Communication" [ICML 2019]
>
> [4]: Koloskova et al Decentralized deep learning with arbitrary communication compression [ICLR 2020]
>
> **W3: On the mathematic framework**
>
> We apologize for not making it clear. Indeed, we say it explicitly in Appendix D that our analysis builds on the framework introduced in the EL paper. Our proof contributes novel analyses for the Stages 1and 3, which we then combine with the EL analysis to prove TGL’s overall convergence. We chose EL as our foundation because it is one of the few frameworks that accommodates non-doubly-stochastic mixing matrices. We will ensure this distinction is clearly stated in the main body of the final version.
>
> **W4: On Practical Assumptions (Synchrony, Churn, Participation)**
>
> We appreciate the reviewer’s concerns over these practical matters. We assume synchronous rounds (like EL, our primary baseline) for analytical clarity. However, TGL's receiver-driven design is naturally suited for asynchrony. We provide a discussion in Appendix C.1 on how TGL can be extended to asynchronous operation using standard techniques (eg local buffers), to gracefully handle network delays. While simulating network delays is a systems-level challenge beyond the scope of this algorithmic work, it is a clear direction for future work.
>
> We do discuss node churn (new node joining, and node failures) in Section 3 (Lines 298-303). TGL’s use of dynamic, random sampling provides inherent advantages in handling this. We also provide an empirical study of fault tolerance in Appendix C.2 / Figure 12 , which shows TGL's graceful performance degradation as the relay drop rate increases.
>
> TGL is fully compatible with partial participation. However, we highlight our ability to support large-scale participation (Line 40) as a strength. While some related works rely on partial participation to save cost (Lines 150-157), TGL's core design is already so communication-frugal that it achieves strong performance even with large-scale or full participation.
>
> **Q1: On the EL-Oracle baseline**
>
> We did not include ELO because it is not a decentralized protocol. ELO requires a central coordinator (oracle) to construct a doubly-stochastic mixing matrix. This is noted in the EL paper itself, where ELO serves as an idealistic baseline for their practical, decentralized ELL protocol. Our work's central motivation is to design a fully coordinator-free protocol, making a comparison to the coordinator-dependent ELO inappropriate.

---

### Official Review · Reviewer_dcRb · 2025-10-31

**Soundness:** 2
**Presentation:** 2
**Contribution:** 1
**Rating:** 2
**Confidence:** 3

**Summary:**

As decentralized training of machine learning models often suffer from fault tolerance and communication overhead, this paper propose a novel hierarchical federated learning algorithm termed "Tiered Gossip Learning" (TGL). The TGL consists of two-layer push-gossip-pull communication layer so that computational nodes do not communication with each other directly, but rather through intermediate relay nodes. By randomly selecting computation nodes and relaying nodes, TGL is able to have some robustness against failure nodes. Theoretical convergence rates are given and some empirical experiments are conducted to support the claims.

**Strengths:**

- Clarity: The paper is written in a clear way.
- Originality: Combining relay nodes and random sampling into a two layer communication topology may be original.
- Quality: Theoretical rates look correct, empirical evaluations are given. Overall this seems a complete work.

**Weaknesses:**

- TGL increases latencies due to the overhead of inter-relay communication.
- TGL increases bubble in training pipelines as Line 258-259 says "Only sampled leaves perform local training in that round" which is not ideal for distributed training in data centers.
- This paper does not give a clear threat model for faulty nodes. It seems this paper assumes as soon as a node fails, servers/relay nodes are automatically aware and move on to the next step. If this paper targets cross-device federated learning, then the system heterogenity of devices leads to some clients being more likely to be straggler and one may not simply exclude them. If this paper only consider simple homogeneous devices, then the novelty in terms of fault tolerance seem limited.

**Questions:**

I am not sure which distributed training scenario does this paper target? Is it cross-device FL / cross-silo FL/ distributed training inside a data center? The setups of TGL does not seem to fall into these common scenarios.

---

> ### Author Response · Authors · 2025-11-19
>
> We thank you for your valuable feedback, and for calling the work original, complete, and clear. We would like to address the concerns raised, particularly clarifying the training setting where TGL may be deployed.
>
> **W1: On Latency Trade-offs**
>
> This is a deliberate and necessary trade-off, which we note in our Discussion. TGL prioritizes communication-frugality and faster convergence (in fewer total rounds) over the per-round wall-clock latency. As we argued in our response to Reviewer cy9C, this 3-stage latency is not a 3x "cost" because:
> - The relay-gossip stage is fast, running on a small, well-connected $n_r$-node resource-capable backbone.
> - The relays act as network shortcuts, effectively reducing the network diameter and thereby the per-round latency in huge networks.
> - Sparse P2P baselines (like ELL) would require multiple gossip rounds per update to match TGL's superior mixing efficiency (shown in our CDR plots), which would compound their own latency.
>
> **W2: On Resource Conservation (Lazy Local Training)**
>
> The reviewer is correct that idling nodes ("bubbles") would be inefficient in a data-center setting. However, TGL is not designed for data-center training. As stated in the introduction (first paragraph) our protocol targets edge deployments and cross-device settings, where data is decentralized and ownership is local.
>
> In this context, the strategy to only perform local training on leaves sampled for the "Push" stage is an intentional design choice for resource conservation. Because we rely on independent random sampling without a central coordinator in Stage 1, it is expected that some leaves may not be selected by any relay in a given round. We prevent unnecessary computation (training) on these resource-constrained devices. This only occurs when the relay budget is extremely small. Crucially, all leaves (whether they trained or not) still participate in Stage 3 (Pull), ensuring all nodes receive the updated, mixed models and stay synchronized. Furthermore, due to the nature of independent random sampling, each node has an equal probability of being selected over the rounds, preventing starvation and ensuring overall fairness.
>
> **W3: On Threat Model and Fault Tolerance**
>
> We assume fail-stop (crashed/unavailable) nodes, not a Byzantine threat model. The "fault tolerance" we refer to is TGL's structural robustness to node failure, which is a key advantage of a decentralized design over the centralized topologies in FL and HFL. Fault tolerance is not a novelty of this work, but a foundational principle for our design to use a decentralized top layer. We validate this structural robustness in Appendix C.2 / Figure 12, which shows TGL's graceful performance degradation under increasing relay drop rates.
>
> The reviewer's query about a node instantly becoming aware of failure addresses a systems-level challenge. Standard distributed system solutions (e.g., using time bounds to infer failure or implementing a separate liveness consensus mechanism) are compatible with TGL's design. Our work does not propose novelty in these system-level detection mechanisms, as our focus is on the core algorithmic framework, the two-tier architecture providing high mixing at low cost with no central coordinator.
>
> The reviewer is correct that stragglers are the main challenge in cross-device FL. As with our baselines, our main analysis assumes synchrony for analytical clarity. However, TGL's design is naturally suited for this setting. We provide a detailed discussion in Appendix C.1 on how TGL can be extended to an asynchronous protocol using standard techniques (e.g., local buffering) to gracefully handle stragglers. This is a primary direction for future systems-level work.
>
> **Q1: On the Training Scenario**
>
> Our protocol targets the cross-device FL scenario. Our core motivation, stated in the introduction (first paragraph), is for "edge deployments," "sensor networks," and "distributed organizations" where data is decentralized and must remain at the source.
>
> This scenario is defined by resource-constrained "leaf" nodes (devices) that hold private data and run local training, do not communicate with each other, and are resource-constrained. This is precisely the setting where TGL's coordinator-free, low-communication, and fault-tolerant design is most beneficial for a practical scalable system.

---

### Official Review · Reviewer_gbto · 2025-11-01

**Soundness:** 3
**Presentation:** 2
**Contribution:** 2
**Rating:** 2
**Confidence:** 5

**Summary:**

This paper proposes a new fully decentralized learning framework called tiered gossip learning, which includes a hierarchical aggregation protocol. Both theoretical and numerical analysis is  provided.

**Strengths:**

* The research topic is quite practical.
* It is always important to robustify decentralized algorithms with an optimized communication budget.

**Weaknesses:**

* Some of the claims are incorrect:
  * Lines 19: "TGL" is fully coordinator-free. The relays are aggregators of the leaf nodes.
  * In line 147, the authors claim that "but generally require orchestration by a central scheduler, and thus remain structurally closer to FL." However, the authors do not specify the benefits of their learning framework as compared to classical FL. From a communication-complexity perspective, a network coordinated by a central server is more efficient and stable than a gossiping framework.
* Some of the statements are unclear and need clarification.
  * A concrete definition (using math and notations) of leaf nodes and relay nodes is never given.
  * The reason why the authors use $\text{CDR}$ to measure the consensus mixing efficiency is unclear. In fact, we can track $\text{CD}$ itself.
  * In Step 0 Local Training, the SGD protocol is not written out. Do we have 1-step local optimization or multiple steps?
  * One of the motivations is to make the training framework fault-tolerant. Yet, Theorem 4.4 in the convergence analysis does not have any characterizations on this part. In fact, the relays may come and go, and their communication graph might not be a complete graph, which the authors have experimented with in their numerical experiments. In fact, they talk about the spectral norm in lines 200 - 203. The reviewer cannot find $\lambda_2$ in Theorem 4.4.
  * It is quite interesting that expectations appear in both the denominators and the numerators in the lower bounds for different $\beta$'s.
  * In assumption 4.2, the authors assume $\mathbb{E}_{\xi \sim D^{(i)}}[\|\|\nabla f(x, \xi) - \nabla f^{(i)}(x)\|\|^2] \le \sigma^2$. The reviewer is quite confused by this. We don't have a global stochastic gradient.
  * The experiment results are barely readable, in particular, for $-\log (\text{CDR})$.
 * Numerical results:
  * It is quite limited compared only to FL. The authors may also wish to compare with state-of-the-art hierarchical FL methods.

**Questions:**

* Can the authors justify their aggregator-free approach in the context of relays? I agreed that there is not a fixed central coordinator, but aggregator-free is an overclaim.
* Can the authors discuss in detail the traditional FL, hierarchical FL, and their approach?
* Can the authors provide more intermediate results with discussions in the main text? Rushing to Theorem 4.4 is too fast.
* Can the authors explain why we have expectations appear both in denominators and numerators?
* Can the authors conduct another round of proofreading and fix the notations and typos?
* Can the authors discuss the technical challenges in their proofs?
* Can the authors provide theoretical results for the fault-tolerant part of the proposed methods, which is their main contribution as compared to traditional FL with only one central server?
* What does $\in {\mathcal{O}}(\cdot)$ mean in line 326 in Theorem 4.4?
* Can the authors illustrate how the spectral norm differs with different graphs?
* Can the authors provide more numerical experiment results as compared to the traditional hierarchical FL?

---

> ### Author Response · Authors · 2025-11-19
> **Response 1/2**
>
> We thank you for your time and feedback. We would like to address the numerous points raised, as we believe many of them stem from a misunderstanding or overlooking of our paper's core claims and contents. We will answer each point directly with citations to the relevant sections of our paper.
>
> **W1.1, Q1: On TGL being aggregator-free or coordinator-free**
>
> We believe the reviewer has misread our claim. We never state TGL is "aggregator-free"; the relays are, of course, aggregators.
>
> Our claim, as stated in the Abstract and Introduction, is that TGL is "coordinator-free". This is a standard term signifying that TGL has no central node (like the server in FL or the root server in HFL) that orchestrates the protocol. This distinction is critical because the central coordinator is the primary bottleneck and single-point-of-failure in centralized systems.
>
> In contrast, the relays in TGL operate independently in a fully decentralized, peer-to-peer fashion. This design is possible because TGL does not require a doubly-stochastic mixing matrix for convergence, unlike other protocols. EL-Oracle, for example, requires a central coordinator to construct a doubly-stochastic matrix (where both rows and columns sum to 1). This removes the nodes' freedom to sample peers independently. Hierarchical FL (HFL) relies on a single root server to coordinate the edge servers.
>
> TGL, similar to EL-Local, only requires row-stochastic matrices. This allows each relay to independently and randomly sample leaves (Stage 1) and other relays (Stage 2)  without any central oversight.
>
> **W1.2, Q2: Benefits of TGL over FL and HFL**
>
> We respectfully point the reviewer to the first two paragraphs of our Introduction, where we discussed these benefits. In the text, we acknowledge that "centralized coordination offers simpler orchestration" but note that this comes "at the cost of fault tolerance and long-term scalability" (Lines 46-47). TGL is explicitly designed to solve this specific scalability issue: that the central server gets congested as clients increase, slows down training , acts as a single point of failure, and struggles to scale. This is the foundational motivation of our work and is detailed in the Introduction (paragraphs 1 & 2).
>
> While a central server may be efficient in small, stable networks, the claim that it is "more efficient and stable" is precisely what our work (and the entire field of P2PL) contests for large-scale, fault-prone edge deployments, where decentralized architectures provide superior robustness and avoid bottlenecks.
>
> Regarding the comparison requested in Q2, FL relies on one server that all clients communicate with. In HFL, the clients talk to edge servers which in turn communicate with the root server. TGL eliminates the root server entirely and uses a decentralized relay layer where aggregation load is distributed and connections are dynamic. This ensures that no single node failure can disconnect the network or halt training.
>
> **W2.1: On the Definitions of Leaf and Relay Nodes**
>
> We explicitly define these roles in the first paragraph of Section 3 (Design of TGL). We describe relays as "server-like intermediaries" and leaves as nodes that "hold private data and perform local training". Formally, we define the model states at the leaves ($x^{(i)}$) and relays ($x^{(k)}$) in Algorithm 1 and provide the mathematical expressions for their updates in each stage of the protocol in the same design section (Lines 240–280).
>
> **W2.2: On using CDR instead of CD**
>
> As clarified in Section 2 (Lines 185-200) and Section 5 (lines 374-377), the absolute value of CD is not informative in isolation as it depends on the model used and changes dynamically before and after aggregation. After local training, the individual models drift, increasing CD, and are brought closer in the mixing step, reducing CD. The ratio by which it contracts, captured by CDR is a measure of the mixing.
>
> **W2.3: On the number of local steps**
>
> We use multiple steps in our experiments. This parameter is explicitly defined as $T_{loc}$ in Algorithm 1 and explained in Section 3 (Lines 236-238). The specific values of $T_{loc}$ in each experiment are listed in Table 1.
>
> **Q3, Q6: On Intermediate Theoretical Results and Technical Challenges**
>
> We provide the proof sketch with all intermediate results as Lemmas A.1-A.5 in Appendix A due to space constraints.
>
> The primary technical challenge was analyzing the novel three-stage mixing process. Unlike standard P2PL (single-step gossip), TGL involves three distinct stages. While Stage 2 (relay-gossip) resembles standard P2PL, the Stage 1 (leaf-to-relay push) and Stage 3 (relay-to-leaf pull) steps introduced unique analytical challenges. We had to derive new, non-trivial bounds for these stages and then combine all three to establish the final, multiplicative contraction factor ($\beta_{TGL}$), as detailed in Appendix D.3 .

---

> ### Author Response · Authors · 2025-11-19
> **Response 2/2**
>
> **W2.4, Q7: On Theoretical Results with Fault Tolerance and Spectral Gap**
>
> "One of the motivations is to make the training framework fault-tolerant. Yet, Theorem 4.4 in the convergence analysis does not have any characterizations on this part."
>
> This is a misunderstanding of how fault tolerance is analyzed. Convergence theorems (like Theorem 4.4) analyze the protocol under its assumed, failure-free model. Fault tolerance is a structural property of the system's design, not a variable in the convergence rate. TGL's fault tolerance stems from its decentralized relay layer, which eliminates the single points of failure inherent in FL and HFL. This is a standard distinction: the vast majority of decentralized learning works derive convergence rates under failure-free assumptions while motivating the architecture via structural robustness.
>
> Regarding the spectral gap: The reviewer appears to be conflating two alternative analytical methodologies. Convergence analysis is typically conducted either by analyzing the spectral gap directly or by recursively bounding the Consensus Distance (CD). While earlier works relied on spectral gap analysis, modern literature (e.g., Epidemic Learning) predominantly utilizes the CD approach. Consequently, the spectral gap term ($\lambda _ 2$) does not appear in convergence results derived via CD contraction. We have selected the CD approach for our theoretical analysis in Theorem 4.4, while providing an empirical spectral gap analysis in Appendix B.1 to independently validate the mixing quality.
>
> **W2.5: On the Mathematical Expression for Lower Bounds of $\beta's$**
>
> This is a correct and intended result of the mathematical derivation. The $\beta$ terms are stage-wise contraction factors on the expected consensus distance. The full derivations are in Appendix D.3 (Proof of Lemma A.2).
> Because the network topology in TGL is random and dynamic, the mixing weights in any given round depend on random variables (for example, the in-degree $A^{(i)}$ in Stage 2). To derive the convergence bound, we must take the expectation over these random graph realizations. Consequently, the resulting contraction coefficient ($\beta_{rr}$ here) naturally contains expectation terms both in the numerator and the denominator.
>
> **W2.6: On Assumption 4.2**
>
> Thank you for pointing out this typo. We mean to use $\nabla f^{(i)}(x, \xi)$ to denote the local stochastic gradient.
>
> **W2.7: Readability of CDR plots**
>
> We apologize if the plots were difficult to read. We will de-clutter the plots and highlight only the important curves. The key takeaway, as described in the captions (e.g., Figure 3 ), is the relative performance: TGL (green) consistently achieves stronger mixing (higher on the y-axis) than the baselines (red, blue) at all comparable practical communication budgets.
>
> **W4: Comparison with SOTA HFL**
>
> We do not only compare to FL. We compare TGL to four important decentralized baselines: ELL, Erdos-Renyi, Exponential, and BaseGraph. We selected these specific baselines because they share TGL's core algorithmic goal: maximizing mixing efficiency at minimum communication cost. These are TGL's direct algorithmic peers.
> We do not compare to Hierarchical FL (HFL) because, as we have stated, HFL operates under a fundamentally different, centralized paradigm that relies on edge and root servers. FL is included solely as a "performance upper bound"  to benchmark the ideal accuracy achievable without communication constraints. HFL is expected to perform as good as FL due to exact consensus while it also inherits the flaws of FL.
>
> **Q8: On Big-O notation**
>
> The notation $\in \mathcal{O}(\cdot)$ refers to standard Big-O notation, used universally in computer science and optimization theory to describe the asymptotic upper bound of the complexity or convergence rate. In Equation 5 (Line 326), it characterizes how the average squared gradient norm scales with key parameters like $T$ (rounds), $n_l$ (nodes) and other design parameters.
>
> Since $\mathcal{O}(g)$ formally denotes a set of functions bounded by $g$, we use the set membership symbol ($\in$) to indicate that our convergence rate belongs to this family, adhering to rigorous mathematical convention rather than the equality sign ($=$) often used loosely in computer science.
>
> **Q9: On Spectral Gap Comparison for Different Graphs**
>
> We illustrate exactly this in two places in the paper. Figure 1 (Page 2) plots the spectral gap against the number of nodes ($n$) for k-regular and Exponential graphs, illustrating how node degree must scale to maintain mixing quality. Figure 6 (Appendix B.1) explicitly compares the spectral gap evolution of TGL against EL-Local and Erdos-Renyi graphs across varying communication budgets (edge counts).

---

### Official Review · Reviewer_Fjaj · 2025-11-04

**Soundness:** 3
**Presentation:** 2
**Contribution:** 2
**Rating:** 4
**Confidence:** 3

**Summary:**

This paper proposes Tiered Gossip Learning (TGL) for large-scale collaborative learning over edge devices. TGL addresses the limitations of Federated Learning (FL), which suffers from a single point-of-failure (the central server) issue, and peer-to-peer learning (P2PL), which requires high communication degrees to maintain good mixing quality as networks scale. The primary motivation behind the design of TGL is to combine the fault-tolerance and decentralization of P2P systems with the efficiency and scalability of hierarchical aggregation.

TGL introduces a two-tier push-gossip-pull protocol with relay nodes in the upper tier and leaf nodes in the lower tier. Leaves push model updates to a small random subset of relays, relays gossip among themselves, and leaves pull updates from random relays. This coordinator-free architecture spreads aggregation across relays, avoids centralized failure risks, and significantly reduces communication overhead.

The paper proves convergence under standard smoothness, bounded variance, and data heterogeneity assumptions , offering explicit bounds on consensus distance and showing how TGL enables independent control over mixing at each layer. Empirical results on CIFAR-10, FEMNIST, and AG-news show that TGL achieves equal or better accuracy that decentralized baselines while using fewer model exchanges.

**Strengths:**

The paper tackles a real limitation of both FL and decentralized FL by using hierarchical aggregation. The proposed tiered gossip approach is clean and intuitive -- using relays to offload communication but without introducing a single coordinator.

The authors provide strong theoretical grounding under standard assumptions, along with explicit bounds to consensus contraction across stages. The spectral gap and consensus distance analysis seem sound. Empirical results also show non-negligible gains. The proposed algorithm is also relevantly compared with prior works such as ELL.

**Weaknesses:**

My major concern is regarding the novelty of the idea of hierarchical aggregation in general. There are several works that have tried addressing this problem, and a more detailed discussion is required over the pros and cons of the proposed TGL with respect to prior works. For example,

1. Client-Edge-Cloud Hierarchical Federated Learning (https://arxiv.org/abs/1905.06641)
2. ColRel: Collaborative Relaying for Federated Learning over Intermittently Connected Networks (https://openreview.net/forum?id=8b0RHdh2Xd0)
3. Multi-Level Local SGD for Heterogeneous Hierarchical Networks (https://arxiv.org/abs/2007.13819)

How do the proposed method compare with prior related works such as the above (and maybe some more).

Some other minor concerns include:

1. In practice, the 3-stage aggregation procedure may introduce additional latency overhead. The authors discuss this asynchronous behavior  only briefly and is not benchmarked experimentally. Some benchmarks in this regard or a better analytical model that takes into account this relaying latency (depending on the quality of communication channels) will be appreciated.

**Questions:**

See above.

---

> ### Author Response · Authors · 2025-11-19
>
> We thank the reviewer for their valuable and constructive feedback. We are encouraged that you found our TGL protocol "clean and intuitive" and our theoretical grounding "strong", and how we propose to solve a "real limitation" of FL and P2PL. Your two main (and very fair) concerns are novelty relative to hierarchical aggregation and latency. We believe we can fully resolve these.
>
> **W1: On the Novelty of the Idea of Hierarchical Aggregation in General**
>
> While TGL adopts a hierarchical structure which sounds like an intuitive solution, this is just one of several core design principles that, when combined, make our work fundamentally different from the cited HFL papers. As stated in our introduction (Lines 95-97), TGL's design is a synthesis of: 1) hierarchy, 2) asymmetric load sharing, 3) a decentralized top layer, and 4) random, dynamic connections.
>
> The HFL works suggested (Client-Edge-Cloud [1], ColRel [2], Multi-Level Local SGD [3]) are all centralized. They employ only the "hierarchy" principle and completely lack the others, most notably the decentralized top layer. This makes them all vulnerable to single points-of-failure (both benign and malicious). They rely on centralized edge servers, which are managed by a single root server. If an edge server fails, all its connected clients are disconnected. If the root server fails, the entire training process halts. As we note in our paper , this HFL design in [1] "inherits, and often amplifies the limitations of FL" by adding new points of failure. [3] further exacerbates this by using static edges in disjoint subnetworks.
>
> TGL's novelty is the specific solution to this. The relay layer is not a collection of static edge servers; it is a fully decentralized, coordinator-free, peer-to-peer gossip network with no root server. Because leaves use random, dynamic connections to sample from this relay pool, no single relay failure can isolate any leaf.
>
> TGL thus uniquely synthesizes the efficiency of a hierarchy with the robustness of a fully decentralized P2P design.
>
> **W2: On Latency Trade-offs**
>
> The reviewer raises a very fair point about the per-round latency, but this is a deliberate trade-off that TGL has. TGL trades a modest increase in per-round wall-clock latency (from 3 stages) for a massive reduction in communication-frugality (total model exchanges) and faster overall convergence (fewer total rounds).
>
> The 3-hop process is not a 3x "cost." The relay gossip stage involves a small, resource-capable backbone of $n_r$ relays which is faster than gossiping across the entire $n_l$-node flat P2P graph. Moreover, relays act as shortcuts in the graph and reduce the network diameter, without which, the communication across spokes could result in a much higher latency across the graph diameter. Further, to match TGL's superior mixing (see our CDR plots), a sparse P2P protocol like ELL would require multiple gossip rounds per update, which would compound its own latency and negate any single-round latency advantage. We have noted these in the Discussion (Lines 471-477).
>
> **W3: On Asynchrony**
>
> We agree that an evaluation in an asynchronous setting with latency simulations (varying the quality of communication channels) is an important direction. However, we consider this a systems-level analysis, whereas the scope of this paper is primarily algorithmic. Our core contribution is to demonstrate that superior mixing can be obtained with significantly reduced communication volume and without a central coordinator by incorporating specific design principles, specifically, the synthesis of hierarchy with decentralized mixing.
>
> To rigorously validate this algorithmic improvement, we selected baselines that share this exact optimization goal of maximizing mixing at minimum cost, namely Epidemic Learning, BaseGraphs, Exponential Graphs, and Erdos-Renyi . While we discuss the straightforward extension to asynchrony in Appendix C.1  to show TGL's design is naturally compatible with such systems, we believe a dedicated systems-level benchmark falls outside the current scope and remains valuable future work.

---

### Author Response · Authors · 2025-11-25
**Request for Feedback**

Dear Area Chair and Reviewers,

We posted our detailed responses to all the reviews last week. We would greatly appreciate your feedback on whether our rebuttals have adequately addressed your concerns, and we remain available for any further discussion.

Best regards,
The Authors

---

### Comment · Area_Chair_ogqq · 2025-11-27
**Request for Timely Response to Authors’ Rebuttal and Discussion**

Dear Reviewers,

I hope you are doing well. The authors have now submitted their rebuttal for the paper under your review. At this stage, your timely response is essential for ensuring a smooth discussion phase.

Could you please review the rebuttal at your earliest convenience and share your updated thoughts? If there are points that require further discussion among the reviewers, please feel free to initiate or join the conversation on the discussion thread.

Your prompt input will greatly help us maintain the review timeline. Thank you very much for your efforts and valuable contributions.

Best regards,

AC

---

### Author Response · Authors · 2025-12-01
**Summary of Rebuttal Key Points**

Dear Area Chair,

Here we summarize our rebuttal discussions regarding our submission to assist in the final assessment.

Regarding the recurring themes of latency and asynchrony, we clarified that the 3-stage protocol does not necessarily increase effective latency; the relay layer acts as a network shortcut, and TGL requires fewer total rounds than P2P baselines to achieve equivalent mixing, as detailed in our responses. Our contribution is primarily algorithmic, achieving superior mixing at lower communication costs. Our evaluation intentionally matches standard baselines (like Epidemic Learning) in a synchronous framework to isolate these algorithmic gains, though we outline straightforward extensions to asynchrony.

Addressing the novelty concern raised by Reviewer Fjaj, we emphasized that TGL uniquely synthesizes four foundational design principles: hierarchy, asymmetric load sharing, a decentralized top layer, and random dynamic connections, needed for a practical and scalable collaborative learning system. We clarified that no existing method combines all four. Specifically, unlike centralized HFL which retains single points of failure, TGL achieves a practical, scalable, and fault-tolerant system through its fully coordinator-free design.

The remainder of our responses focuses on clarifying specific misunderstandings and highlighting technical details that were overlooked in the initial reviews.

Best regards,

The Authors

---

### Meta-Review · Area_Chair_ahDd · 2025-12-26

**Summary:**

This paper studies collaborative training when many devices keep data locally and must communicate over a network without relying on a single central server. The paper introduces a relay-based communication pattern in which devices exchange models through a smaller relay layer that also mixes information among relays, with the goal of keeping per-device communication small while maintaining good global mixing. The submission includes a convergence analysis under standard assumptions and experiments suggesting improved accuracy per communication compared to several decentralized baselines.

The reviews agree that the direction is relevant and potentially useful, but they raise two clusters of concerns. From a systems perspective, multiple reviews emphasize that a multi-stage communication round raises wall-clock latency and robustness questions (delays, stragglers, churn, partial participation) that are not empirically investigated. From a more theory-oriented perspective, the core contribution appears closely connected to the well-studied problem of designing randomized gossip topologies and analyzing their mixing, and the convergence results rely on standard but somewhat strong conditions, such as bounded heterogeneity. Without a tighter integration into decentralized optimization theory and a clearer explanation of how the analysis meaningfully extends prior frameworks, the contribution risks feeling underdeveloped relative to the maturity of the area.

**Reviewer Concerns:**

The rebuttal and discussion provide clarity on several key points, including terminology related to coordinator-free operation, where definitions and update rules are presented, and how the convergence argument relates to prior epidemic-learning style analyses, while also adding bounds for the additional stages. These responses plausibly address some misunderstandings and would likely improve the perceived readability of a revised version.

The main decision-driving concerns remain open. On the systems side, cy9C and Fjaj in particular focus on latency and on the lack of evidence under realistic delays or asynchronous behavior; dcRb echoes related issues about participation dynamics and practical deployment assumptions. A single targeted evaluation that measures wall-clock time under simple delay models or tests an asynchronous variant would directly address this. On the theory and positioning side, the paper would be much stronger if it explicitly connected to established decentralized optimization and gossip averaging frameworks, clearly stated which parts are standard (topology-driven mixing and contraction-style arguments) and which parts are new, and justified the assumption set, especially around heterogeneity, in relation to common conditions in the literature. This deeper contextualization is also important for the community value of the work, since otherwise the contribution can read primarily as a new topology choice without a sufficiently grounded relationship to prior results.

**Reviewer Scores:**

It seems, Fjaj might possibly increase their score to an accept, mainly due to improved framing and clarifications in the rebuttal.

cy9C already recommends acceptance, and it is unlikely they would move to strong acceptance (under the current state of the discussion).

In contrast, dcRb and gbto both recommend rejecting the paper. While the rebuttal helps with misunderstandings it does so far not supply deeper comparative grounding that would materially change the view of these reviewers.

---

### Decision · Program_Chairs · 2026-01-26

Reject